

# The EC-Earth3 Earth System Model for the Climate Model Intercomparison Project 6

Ralf Döscher[1], Mario Acosta[2], Andrea Alessandri[3], Peter Anthoni[4], Almut Arneth[4], Thomas Arsouze[2],
Tommi Bergman[5], Raffaele Bernardello[2], Souhail Bousetta[6], Louis-Philippe Caron[2], Glenn Carver[6],
Miguel Castrillo[2], Franco Catalano[7], Ivana Cvijanovic[2], Paolo Davini[8], Evelien Dekker[1], Francisco J. Doblas-Reyes[2], David Docquier[1], Pablo Echevarria[2], Uwe Fladrich[1], Ramon Fuentes-Franco[1], Matthias Gröger[1], Jost v. Hardenberg[9,8], Jenny Hieronymus[1], M. Pasha Karami[1], Jukka-Pekka Keskinen[10], Torben Koenigk[1], Risto Makkonen[11], Francois Massonnet[12], Martin Ménégoz[13], Paul A. Miller[14], Eduardo Moreno-Chamarro[2], Lars Nieradzik[14], Twan van Noije[5], Paul Nolan[15], Declan O'Donnell[11], Pirkka Ollinaho[11], Gijs van den Oord[5], Pablo Ortega[2], Oriol Tintó Prims[2], Arthur Ramos[2], Thomas Reerink[5], Clement Rousset[16], Yohan Ruprich-Robert[2], Philippe Le Sager[5], Torben Schmith[16], Roland Schrödner[14], Federico Serva[17], Valentina Sicardi[2], Marianne Sloth Madsen[18], Benjamin Smith[14], Tian Tian[18], Etienne Tourigny[2], Petteri Uotila[10], Martin Vancoppenolle[19], Shiyu Wang[1], David Wårlind[14], Ulrika Willén[1], Klaus Wyser[1], Shuting Yang[18], Xavier Yepes-Arbós[2], Qiong Zhang[20]

[1]Swedish Meteorological and Hydrological Institute SMHI, Norrköping, 60176, Sweden
[2]Barcelona Supercomputing Center, Barcelona, 08034, Spain
[3]Institute of Atmospheric Sciences and Climate, Consiglio Nazionale delle Ricerche, ISAC-CNR, 40129, Bologna, Italy
[4]Karlsruhe Institute of Technology KIT, Garmisch-Partenkirchen, 82467, Germany
[5]Royal Netherlands Meteorological Institute KNMI, De Bilt, 3731, The Netherlands
[6]European Centre for Medium Range Weather Forecast ECMWF, Reading, RG2-9AX, United Kingdom
[7]Italian National Agency for New Technologies, Energy and Sustainable Economic Development ENEA, Roma, 00196, Italy
[8]Institute of Atmospheric Sciences and Climate, Consiglio Nazionale delle Ricerche, ISAC-CNR, 10133, Torino, Italy
[9]Politecnico di Torino, 10129 Torino, Italy
[10]University of Helsinki, Helsinki, 00014, Finland
[11]Finnish Meteorological Institute, FMI, Helsinki, 00560, Finland
[12]Université Catholique de Louvain UCLouvain, Ottignies-Louvain-la-Neuve, 1348, Belgium
[13]IGE, University of Grenoble, Grenoble, 38400, France
[14]Lund University, Lund, 22100, Sweden
[15]Irish Centre for High End Computing, ICHECK, Ireland
[16]UPMC University Pierre and Marie Curie UPMC, Paris, 75005, France
[17]Istituto di Scienze Marine CNR-ISMAR, Venezia, 30122, Italy
[18]Danish Meteorological Institute, Copenhagen, 2100, Denmark
[19]Institut Pierre Simon Laplace IPSL, Paris, 75005, France
[20]Stockholm University, Stockholm, 106 91, Sweden

*Correspondence to*: Ralf Döscher (ralf.doescher@smhi.se)

**Abstract.** The Earth System Model EC-Earth3 for contributions to CMIP6 is documented here, with its flexible coupling framework, major model configurations, a methodology for ensuring the simulations are comparable across different HPC



systems, and with the physical performance of base configurations over the historical period. The variety of possible configurations and sub-models reflects the broad interests in the EC-Earth community. EC-Earth3 key performance metrics demonstrate physical behaviour and biases well within the frame known from recent CMIP models. With improved physical

and dynamic features, new ESM components, community tools, and largely improved physical performance compared to the CMIP5 version, EC-Earth3 represents a clear step forward for the only European community ESM. We demonstrate here that EC-Earth3 is suited for a range of tasks in CMIP6 and beyond.

## 1 Introduction

The latest challenges in climate research have evolved to include biophysical and biogeochemical processes (WCRP

Strategic Plan 2019-2028) contributing to the exchange of energy, mass, aerosols, trace and greenhouse gases and nutrients between atmosphere, land and ocean, allowing the description of various feedback processes. This challenge resulted in a need for the next generation of climate models  -  namely, the Earth System Models (ESMs).

The Paris Climate accord is calling for limiting climate change "well below 2°C and to pursue efforts to limit the increase to

1.5°C", ESMs represent our most relevant tools available for exploring the emission pathways necessary for achieving this goal, as well as for understanding the consequences of not making this target. The Paris agreement requires firm measures of mitigation, including carbon dioxide removal. Given the complexity of the climate system, alternative emission pathways towards this goal can be carefully explored only with Earth System Models (ESMs) which describe the most relevant feedback mechanisms, and provide methods for assessments of uncertainty. ESMs are the primary source of information for

understanding the Earth's climate feedbacks, for attributing changes to specific drivers, for future climate projections and predictions, and for the development of mitigation policies.

While the exact definition of ESM varies, in general, it refers to a complex model that besides the classical climate model core (consisting of physical models of the atmosphere, sea ice, ocean and land) combines additional optional components

covering biophysical and biogeochemical processes and more sophisticated treatment of aerosols. A flexible coupling framework facilitates a range of ESM configurations with or without certain model components or processes. Given the important role of ESMs, these models need to be developed together with use cases for science, climate services and decision making that control the priorities of development.

This article describes EC-Earth3, an Earth System Model with the flexibility of different configurations that allow users to consider (or exclude) various climate feedbacks and processes. It has been developed collaboratively by the European research consortium EC-Earth to  provide a community of European research institutes and universities with an integrated state-of-the-art tool for Earth system studies. While its development goals were largely motivated by the Coupled Model



Intercomparison Project phase 6 (CMIP6, Eyring et al. 2016), its suite of ESM configurations allows exploration of a broad
range of climate science questions.

The predecessor system EC-Earth2 (Hazeleger et al., 2012) approached the concept of ''seamless prediction'' to forge
models for weather forecasting and climate change studies into a joint system. EC-Earth version 2.2 was based on an adapted
version of the atmosphere model IFS 31r1, the Integrated Forecasting System of the European Centre for Medium-Range
Weather Forecasts (ECMWF), as used in their seasonal prediction system 3. In addition, a configuration including the
atmospheric composition model TM5 was developed (van Noije et al., 2014) and released as EC-Earth version 2.4. EC-
Earth2 has been used for simulations under CMIP5 and in a range of climate studies, e.g. Koenigk et al. 2013, Seneviratne et
al. 2013) A search on Google Scholar gives 1920 hits for articles mentioning the EC-Earth "climate model", which is a
substantial number, even when compared to 4280 hits for the US community model CESM, which has a much larger
community behind it.

The current version EC-Earth3 for CMIP6 still leans on the original idea of a climate model system based on the seasonal
prediction system of ECMWF. Development has started in 2012 by re-designing the software infrastructure and updating the
atmosphere model to IFS 36r4, corresponding to the ECMWF seasonal prediction system 4. Since then, various updates,
improvements and forcings have been implemented and the model has been tuned for several intermediate versions and
finally for the CMIP6 version, EC-Earth3.

Adaptation of IFS for EC-Earth follows up on the strategy of mutual benefits between short/medium range weather
prediction on the one hand and longer time scale climate prediction and projection on the other. While short term processes
and feedbacks are expected to be covered well in the seasonal prediction system, longer term conservation and trends are the
focus of climate model development. During the development process, EC-Earth has been able to feed back valuable
information to ECMWF. Examples are a stochastic physics tendency conservation fix for humidity and energy (Leutbecher
et al. 2017), forcing (tropospheric and stratospheric aerosol, ozone) and an implementation of aerosol forcing as used in
CMIP6 ("MACv2-SP")

The EC-Earth ESM exists in different coupled configurations that reflect a variety of study options and science interests. The
system comes with a pure physical core configuration in the form of a Global Climate Model (GCM) with a range of
options: a GCM with prescribed or interactively coupled dynamic vegetation, a dynamical Greenland ice sheet, and a closed
carbon cycle. Also, a configuration with interactive aerosols and atmospheric chemistry is available, and GCM
configurations have been established in different resolutions for the atmosphere and ocean.





As a community model, EC-Earth3 is run on several different HPC platforms. While expecting the same simulated climate on each machine, we cannot expect binary identical results. To ensure consistency between different machines, a test protocol and statistical test procedure have been designed.


This paper describes the EC-Earth3 model concept, and provides an overview of its component models and the range of available coupled configurations. Specific configurations will be described in more detail in forthcoming papers. The model's physical performance is illustrated based on the core GCM configurations, with a focus on results from historical simulations performed under the CMIP6 protocol. The EC-Earth consortium consists of 27 partners in 10 European

countries.

## 2. Configurations

### 2.1 The model architecture and coupling framework

EC-Earth is a modular Earth System Model (ESM) that is collaboratively developed by the European consortium with the same name. The current generation of the model, EC-Earth3, has been developed after CMIP5 and it is used in its version

3.3 for CMIP6 experiments.

EC-Earth3 comprises model components for various physical domains and system components describing atmosphere, ocean, sea ice, land surface, dynamic vegetation, atmospheric composition, ocean biogeochemistry and the Greenland ice sheet. The component models are described in section 3. The atmosphere and land domains are covered by ECMWF's IFS

cycle 36r4, which is supplemented with a coupling interface to allow boundary data exchange with other components (ocean, dynamic vegetation, aerosols and atmospheric chemistry, etc). The NEMO3.6 and LIM3 models are the ocean and sea-ice components, respectively. Biogeochemical processes in the ocean are simulated by the PISCES model. Both LIM3 and PISCES are code-wise integrated in NEMO. Dynamical vegetation, land use and terrestrial biogeochemistry are provided by LPJ-GUESS (Smith et al., 2014, Lindeskog et al., 2013). Aerosols and chemical processes in the atmosphere are described

by TM5 . The ice sheet model PISM is optionally utilized to model the Greenland ice sheet.

An overview of five ESM model configurations is given in this section. Descriptions are schematic and more detailed specifications will be given in forthcoming publications. Table 1 lists the configurations and their composition, while Table 2 shows the commonly used resolutions for CMIP6.


Most of the model components are coupled through the OASIS3-MCT coupling library (Craig et al., 2017) while some software components include more than one model component, e.g. the sea-ice model being a part of the ocean model. A new coupling interface has been developed and implemented to allow a flexible exchange between the model components



(see section 3). The OASIS3-MCT coupler provides a technical means of exchanging (sending and receiving) two- and
three-dimensional coupling fields between different model components on their different grids. Of the above named model
components, NEMO, LIM3 and PISCES exchange data directly via shared data structures. Thus, EC-Earth3 is implemented
following a multi-executable MPMD (multiple programs, multiple data) approach. The model components run concurrently
and message-passing interface (MPI) is used for parallelisation within the components. A potential configuration of all
components is illustrated in Figure 1, which also shows coupling links and frequencies. Note that a configuration including
all possible components is not implemented in practice.

In order to manage different configurations, both at build and run time, EC-Earth3 includes tools to store and retrieve
configuration parameters for different model configurations, computational platforms and experiment types. This allows
consistent control of the build and run environments and improves reproducibility across platforms and use-cases.
Initial and forcing data, in the form of data files, are provided centrally for the EC-Earth community, and the data is
versioned and checksummed for reproducibility.
For EC-Earth3 a tool was developed to convert the native model output to CF-compliant ("Climate and Forecast" standard)
netCDF format (i.e., Climate Model Output Rewriter, CMOR), thus fulfilling the CMIP6 Data Requests for the MIPs
that the community is contributing to (van den Oord et al. (2017), https://github.com/EC-Earth/ece2cmor3/).

### 2.2 Basic configurations EC-Earth3 and EC-Earth3-Veg

EC-Earth3 is the standard configuration consisting of the atmosphere model IFS (section 3.1) including the land surface
module HTESSEL (section 3.2) and the ocean model NEMO3.6 including the sea ice module LIM3 (section 3.5). Coupling
variables are communicated between the different component models (see section 3) via the OASIS3-MCT coupler. The
physical interfaces are defined specifying the variables exchanged and the algorithms used.

At the atmosphere-ocean interface, we follow the principle that the ocean provides state variables and the atmosphere sends
fluxes (Table 3). Flux formulations correspond to the documentation of IFS CY36R1, section 3, at
https://www.ecmwf.int/en/publications/ifs-documentation. As the coupler ensures conservative remapping, momentum,
energy, evaporation and precipitation fluxes are conserved.

The freshwater runoff from land to ocean is derived from a runoff mapper (Table. 4). It uses OASIS3-MCT to interpolate
local runoff and ice-shelf calving (from Greenland and Antarctica) to the ocean. The runoff and calving received from the
atmosphere and from the surface model HTESSEL are interpolated onto 66 hydrological drainage basins on a mapper grid
by a nearest-neighbour distance-based Gauss-weighted interpolation method. Then, in a coupling post-processing step





("CONSERV GLBPOS step"), the residual (target minus source grid integrals) is distributed over the target grid, proportional to the original value. The resulting runoff to the ocean is evenly and instantaneously distributed along the ocean coastal points connected to each hydrological basin. This approach does not constitute a locally conservative method in a

mathematical sense but it conserves mass.

In order to avoid a significant long-term sea-surface height reduction in coupled model runs due to a net precipitation - evaporation (P-E) imbalance in the EC-Earth3 atmosphere of about -0.016 mm/day in the historical period, the coupled model implements a runoff flux corrector, which amplifies river runoff by 7.95% in order to compensate for this effect.


EC-Earth3-Veg is a configuration extending EC-Earth3 by the interactively coupled 2$^{nd}$ generation dynamic global vegetation model LPJ-GUESS, which is described together with the coupling principles in section 3.3. Here we provide the variables exchanged through the coupler.

The coupling interface between the atmosphere and vegetation (Table 5) is characterized by the atmospheric model sending

the driving variables, as well as selected biogeophysical soil parameters computed within HTESSEL. LPJ-GUESS returns vegetation parameters for both high and low vegetation categories needed for computing surface energy and water exchange in HTESSEL. This ensures that EC-Earth makes best use of both the advanced biophysics in the HTESSEL land-surface model and of the state-of-the-art vegetation dynamics, land use functionality and terrestrial biogeochemistry (carbon and nitrogen) in LPJ-GUESS. Since HTESSEL and LPJ-GUESS have very different soil water schemes (LPJ-GUESS updates

soil moisture separately in each patch and stand-type for each gridcell (see section 3.3) whereas HTESSEL simulates soil moisture per gridcell), the water cycle is discontinuous and each model operates its own water cycle. The water cycle of LPJ-GUESS is thus loosely coupled to the rest of EC-Earth by means of the driving variables sent by HTESSEL/IFS.

**Atmospheric tuning of EC-Earth3 and EC-Earth3-Veg**

The atmospheric component of EC-Earth has been tuned with the goal of achieving a reasonably small radiative imbalance at the top of the atmosphere (TOA) at standard resolution (T255L91 – to which we refer in the following) in present-day atmosphere-standalone (AMIP) runs, using the CERES_EBAF_Ed4.0 dataset as a reference (Loeb et al. 2018). In particular the goal was to minimize the mean weighted absolute error in the global means of the net radiative flux at the surface, the TOA longwave flux, longwave cloud forcing and shortwave cloud forcing, with the first two fluxes considered most

important. The net radiative flux at the surface included the latent heat contribution associated with snowfall which is not included in the latent heat flux stored by IFS. A series of convective and microphysical atmospheric tuning parameters was identified, listed in Table 6. Similar parameters have been commonly used also for the tuning of other climate models (e.g. Mauritsen et al. 2012). An additional critical radius for the autoconversion process of liquid cloud droplets, added in EC-Earth3, was considered for tuning (see Rotstayn (2000) for a discussion on the use of such parameters for model tuning).





Changes in the tuning parameters have been adopted to avoid values too different to the original IFS CY36R4 values. In order to proceed with tuning, the sensitivity of the model radiative fluxes to changes in these parameters was determined through a series of short (6 years) AMIP runs for present-day conditions. The resulting linear sensitivities accelerate considerably the tuning process and reduce the number of simulations needed, allowing to construct a linear "tuning simulator" used to predict the impact of different combinations of tuning parameter changes on the target radiative fluxes

and to determine combinations providing an optimal score. An iterative process was followed, alternating the construction of new sets of tuning parameters using the known sensitivities, AMIP tuning runs for present-day conditions (20 years, from 1990 to 2010) and the following construction of a new set of tuning parameters to correct the residual biases, allowing to converge rapidly to a desired radiative balance. During this process model biases in other fields were monitored using a Reichler and Kim (2008) metric. Following a suggestion by ECMWF, we reintroduced in the code a condensation limiter for

clouds, which had been removed in CY36R4, but then reintroduced in later cycles starting from CY37R2. Apart from improving the upper tropospheric distribution of humidity in IFS, this change has an important impact on radiative fluxes (more than +1.6 W/m$^2$ in net flux at TOA), making it a useful tool for tuning the global radiative balance. The atmospheric tuning process showed that energy conservation in IFS is severely dependent on the timestep used. For example at standard resolution, reducing the timestep testwise from 2700s to 900s, changes net surface fluxes by -2 W/m$^2$ , mainly due to an

increase in low clouds, possibly due to resolution dependent parameterizations. This issue has been improved in later operational versions at ECMWF.

A similar tuning procedure was used to find alternative tuning parameter sets also for other configurations (EC-Earth3-AerChem, EC-Earth3-LR, and EC-Earth3-Veg-LR). The atmospheric tuning for EC-Earth3 and EC-Earth3-Veg is the same, as is the case for EC-Earth3-LR and EC-Earth3-Veg-LR. This is because the vegetation fields used for EC-Earth3 were

derived from dynamic vegetation model runs. Therefore there are only very small differences between the two configurations (with and without dynamic vegetation) for each resolution, in terms of impact of vegetation on the global energy balance.

**Coupled tuning of EC-Earth3 and EC-Earth3-Veg**

In parallel to forced ocean tuning experiments, the tuned atmosphere was used in coupled present-day experiments researching optimal ocean parameters that allow for a realistic ocean circulation. See section 3.5 for details on some of the changes developed in this phase.

Tuning the final coupled model was aimed primarily at obtaining a realistic global climate at equilibrium in CMIP6 pre-

industrial experiments, focusing in particular on the sea-ice distribution and extent, the near-surface air temperature distribution, atmospheric variability, the Sea Surface Temperature (SST) distribution (in particular the Southern Ocean temperature bias) and ocean transport due to the Atlantic Meridional Overturning Circulation (AMOC), while at the same



time reaching a realistic average global temperature at equilibrium (286.7 K to 286.9 K following IPCC 2018, Hawkins et al. 2017, Brohan et al. 2006). The goal was to where-possible modify only ocean and sea-ice parameters while maintaining the

same atmospheric tuning (even if some changes have been explored). A common set of tuning parameters suitable for both EC-Earth3 and EC-Earth3-Veg experiments was searched. To this end we performed both a range of pre-industrial simulations and, for comparison, corresponding present-day simulations (using fixed 1990 forcing fields and compared to 2010 observations). Gregory plots (Gregory et al. 2004) were used to compare different coupled experiments, to anticipate their approximate equilibrium temperatures even when only partial results were available and to derive suggested corrections

to the global net radiative forcing. The main change which was adopted during this stage was an improved pre-industrial aerosol climatology produced with a different calculation of the sea-spray source, characterized by a stronger dependence on surface wind speed (reverting from the formulation of Salisbury et al. (2013) to that of Monahan et al. (1986)) and by a dependence on sea-surface temperature, following Salter et al. (2015). These changes increased sea-spray production over the Southern Ocean and helped to reduce the Southern Ocean SST bias. Details about the revised parameterization are given

by van Noije et al. (2020). Finally, a further minor change was a small reduction of thermal conductivity of snow in LIM3 (rn_cdsn=0.27).

An interesting observation for pre-industrial equilibrium simulations is that at equilibrium we expect radiative balance at TOA and at the surface on average, but we have to take into account two additional effects: 1) While NEMO takes into

account the temperature of incoming and outgoing mass fluxes (rainfall, snowfall, evaporation and runoff fluxes) to represent dilution effects, IFS does not account for the heat content of the moisture field and of precipitation, leading to a missing closure of the global heat budget, corresponding to a heat sink in the ocean; 2) NEMO includes a representation of geothermal energy sources. Estimating the total heat imbalance in the ocean comparing ocean heating rate of increase with the net flux at surface in a pre-industrial experiment, leads to a total estimate of about -0.2 W/m$^2$ (as a global average). This

energy sink compensates to a large extent an internal energy production observed in IFS (as difference between the net TOA and net surface radiative fluxes) of about 0.25 W/m$^2$, explaining the TOA net flux close to 0 of EC-Earth3 in pre-industrial experiments.

**Low resolution configurations**

EC-Earth3-Veg-LR is a configuration with interactive LPJ-GUESS feedback at low resolution (T159 for IFS and 1° for

ORCA/NEMO). This configuration is applied in the Paleoclimate Modelling Intercomparison Project (PMIP, Kageyama et al., 2018). The major aim of PMIP is to understand the response of the climate system to different climate forcings and feedbacks in the last millennium and in earlier periods. This requires substantial computational resources for multiple multi-centennial simulations. EC-Earth3-Veg-LR makes this possible by a reduced resolution. In addition to resolution differences, new physical parameterizations are also included and tuning parameters are further modified following the same strategy

described in the previous paragraph.

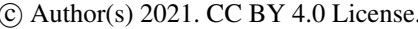

Compared to the corresponding configuration with the standard resolution (EC-Earth3-Veg), additional parameter adjustments are introduced to allow for paleoclimate simulations. The adjustments mainly include two parts. Most importantly, orbital forcing parameters are made variable in time. In other configurations, used for centennial scale

simulations, these parameters are treated as constants, representing present-day climate. That approximation does not hold for multi-centennial to millennial time scales. The new variable calculation for the orbital parameters are taken and modified from CAM3.0 (2004) using the method of Berger (1978). The annual and diurnal cycles of solar insolation are calculated with a repeatable solar year of 365 days and with a mean solar day of exactly 24 hours, respectively. This adjusted formulation facilitates paleoclimate simulations for any time within $10^6$ years of 1950 AD. More detailed description on the

implemented variable orbital parameters are provided in section 3.1.

Another adjustment is related to the description of glaciers and Greenland ice sheet. In the standard resolution configuration EC-Earth3-Veg, the physics of land ice is not accounted for. This is not appropriate for paleoclimate simulations. Therefore, a land ice physics package is implemented describing surface physics and time varying snow albedo over land ice (except for

Antarctica) without including a dynamic ice sheet model. More details are provided in the description of EC-Earth3-GrIS below in section 2.6.

Due to the revised parameterizations and reduced resolution (including the different timestep), key quantities and model biases are different from the standard configuration EC-Earth3-Veg. Therefore the EC-Earth3-Veg-LR configuration

requires a separate tuning. The difference between net TOA and net surface radiative fluxes is almost independent of the tuning and only depends on the resolution. In the standard resolution, the difference is in the order of -0.25 W/m², while the difference increases to about 0.3 W/m² for the low resolution.

Rather than tuning towards the currently observed transient climate state with a global mean imbalance of the order of 0.5

W/m² at the top of TOA (Hansen et al., 2011), we aimed at a tuning of a climate in a radiative equilibrium, to prevent the global mean surface temperature from drifting too much under the conditions of a stable climate. This approach is necessary for millenium scale simulations. We aimed at a net surface energy balance close to 0 W/m² under pre-industrial level forcing (1850) after hundreds years of spin-up. Thereby we mainly focused on the net surface energy (SFC) balance rather than the TOA energy budget as we know that the atmospheric model is not fully conservative. The resulting parameter combination,

together with historical simulations will be described in a forthcoming paper in conjunction with partners in the EU-Crescendo project.



In order to avoid a significant long-term sea-surface height reduction in coupled model runs due to a net precipitation - evaporation (P-E) imbalance in the EC-Earth3 atmosphere of about -0.0174 mm/day in the historical period, the coupled

model implements a runoff flux corrector, which amplifies river runoff by 8.65% in order to compensate for this effect.

In addition to the EC-Earth3-Veg-LR configuration, there is also a configuration without interactive vegetation, EC-Earth3-LR. In this configuration, vegetation is prescribed by the Paleo MIP (PMIP). These two configurations produce very similar results when EC-Earth3-LR is forced by the vegetation from a corresponding EC-Earth3-Veg-LR simulation. The tuning

parameters are identical in both configurations.

The high resolution configurations

**High resolution configurations**

Earlier studies with EC-Earth at high resolution, using EC-Earth 3.1, have shown improvements with resolution, e.g. in North-Atlantic Blocking (Davini et al. 2017b) and in the representation of tropical rainfall extremes (Davini et al. 2017a).

This motivated further development of EC-Earth3 configuration in high resolution, with increased atmospheric and oceanic resolution, derived from an earlier state of development. It features a T511 spectral resolution for IFS and 0.25 degree resolution for ORCA/NEMO. A preliminary tuned version EC-Earth3P-HR is used in current projects and in CMIP6 MIPs. Another high resolution configuration, EC-Earth3-HR, closer to the EC-Earth3 base configuration, is still under development. Here we focus on the so far better documented configuration EC-Earth3P-HR.

At an early stage of development, EC-Earth3P-HR has been branched off from the main line, in order to apply it for the EU project PRIMAVERA and the HighResMIP, endorsed by CMIP6. PRIMAVERA and HighResMIP are focusing on the impact of horizontal resolution on the simulation of climate and its variability. The HighResMIP protocol requires modifications of the standard configuration to allow for a clean assessment of the impact of horizontal resolution. Motivation and detailed description of those deviations from the base version, EC-Earth3, are described in Haarsma et al. (2020). Below

we give a short summary of the most important deviations of EC-Earth3P-HR:

- The stratospheric aerosol forcing is handled in a simplified way that neglects the details of the vertical distribution and only takes into account the total aerosol optical depth in the stratosphere which is then evenly distributed across the stratosphere. No indirect aerosol effect has been implemented.
- A SST and sea-ice forcing data set specially developed for HighResMIP is used for AMIP experiments (Kennedy et
al., 2017). The major differences compared to the standard SST forcing data sets for CMIP6 are the higher spatial (0.25 deg vs. 1 deg) and temporal (daily vs. monthly) resolution.
- The vegetation and its albedo are prescribed as present-day climatologies that are constant in time.

Under HighResMIP, simulations are performed with EC-Earth3P-HR in high resolution and in the standard resolution EC-Earth3P (T255 for IFS and 1.0 degree for ORCA/NEMO). A full description of EC-Earth3P-HR including a technical





implementation and post-processing can be found in Haarsma et al. (2020). EC-Earth3P-HR has not been tuned differently compared to the standard resolution at the time, due to very high computational demands. This approach is consistent with most other models in Europe, as represented in the H2020 PRIMAVERA project (Roberts et al., 2018).

Based on results of Haarsma et al. (2020), increasing horizontal resolution does not result in a general reduction of biases and overall improvement of the climate variability. Deteriorating impacts can be detected for specific regions and
phenomena such as some Euro-Atlantic weather regimes, whereas others such as El Niño-Southern Oscillation show a clear improvement in their spatial structure. Analysis of the kinetic energy spectrum indicates that the sub-synoptic scales are better resolved at higher resolution (Klaver et al., 2020) in EC-Earth.

Despite a lack of clear improvement with respect to biases and synoptic scale variability for the high resolution version of EC-Earth, the better representation of sub-synoptic scales results in better representation of phenomena and processes on
these scales such as tropical cyclones (Roberts et al., 2020) and ocean-atmosphere interaction along western boundary currents (Belluci et al. 2020). The impact of resolution for EC-Earth and other climate models participating in HighResMIP will be analyzed more in depth in upcoming publications.

### 2.3 EC-Earth3-AerChem

EC-Earth3-AerChem (van Noije et al., 2020) is the configuration with interactive aerosols and atmospheric chemistry, used
in the Aerosol and Chemistry Model Intercomparison Project (AerChemMIP; Collins et al., 2017). In this configuration, TM5 is used to simulate tropospheric aerosols and chemistry based on the CMIP6 emission pathways for aerosols and chemically reactive gases. The resolution of TM5 is 3×2 degrees (longitude × latitude) with 34 vertical levels and a top at 0.1 hPa. IFS and NEMO have the same resolutions as in the standard configuration. TM5 and IFS exchange fields with a 6 hour frequency. TM5 receives a large set of 2-D and 3-D meteorological fields from IFS, and provides 3-D distributions of
aerosols, ozone ($O_3$) and methane ($CH_4$) in return. Table 7  lists the fields exchanged between IFS and TM5 through the coupler.

### 2.4 EC-Earth3-CC

EC-Earth3-CC is the configuration that includes a description of the carbon cycle, which is used for the Coupled Climate–Carbon Cycle Model Intercomparison Project (C4MIP; Jones et al., 2016). EC-Earth3-CC allows
simulations with emissions forcing rather than with prescribed concentrations only as in the ScenarioMIP. This configuration uses a single carbon tracer in the atmosphere, advected by a version of TM5 with a reduced number of vertical levels (10 instead of 34), to simulate the transport of $CO_2$ through the atmosphere. The resolution and coupling frequency for the exchange between IFS and TM5 are the same as for the interactive aerosols and chemistry version of TM5 (EC-Earth3-AerChem), described in the preceding section. In effect the data transfer in





both directions is much reduced (see Appendix). The CO₂ exchange with the ocean and terrestrial biosphere is calculated in PISCES and LPJ-GUESS, respectively, based on surface mixing ratios from the previous day, received from TM5.

PISCES calculates the air-sea $CO_2$ flux at every time-step after solving for carbon chemistry in sea-water. This flux is
proportional to the difference in pCO2 between the atmosphere and the surface of the ocean. The exchange of $CO_2$ between the ocean and TM5 is realized once a day after accumulating the flux over each grid-cell over 24 hours. Furthermore, physical transport of passive tracers in the ocean presents a slight artificial mass imbalance. To prevent it from becoming significant for carbon during the spin-up we applied a uniform correction to dissolved inorganic carbon at the end of each year, after taking into account all sources and sinks.


A variant of EC-Earth-CC can also be run concentration driven by excluding TM5. PISCES and LPJ-GUESS then read a uniform global atmospheric $CO_2$ concentration.

Table 8 and 9 lists the fields exchanged between the CTM on the one hand side and the vegetation and ocean
biogeochemistry models on the other hand side, through the coupler.

### 2.5 EC-Earth3-GrIS

EC-Earth3-GrIS is a configuration that couples the EC-Earth3-Veg to the Parallel Ice Sheet Model v1.1 (PISM, section 3.8). It is used to model the Greenland ice sheet (GrIS) evolution and its feedback with the climate system in the Ice Sheet Model
Intercomparison project (ISMIP6, Nowicki et al, 2016).

In the configurations EC-Earth3 and EC-Earth3-Veg, ice sheets are represented by a perennial snow layer of 9 meter water equivalent. Snowfall on these areas is immediately redistributed into the ocean as ice to prevent excessive snow accumulation. Perennial snow albedo and snow density are fixed at 0.8 and 300 kg m⁻³ respectively and the snowpack is in thermal contact with the underlying soil. In EC-Earth3-GrIS, the surface parameterization in EC-Earth3 is adjusted in order
to better account for the presence of the ice sheet. The modifications include introduction of an explicit ice sheet mask obtained from PISM into HTESSEL, and application of values representative of an ice sheet to calculate the surface energy balance and subsurface heat and energy transfer for glacierized grid points. In addition, if a grid cell is with ice sheet but no snow cover (i.e., bare ice), the ice can melt and contribute to surface run-off, if the energy flux at the surface is positive. Furthermore, a time-varying snow albedo parameterization is introduced for snow on ice sheets (Helsen et al, 2017) in the
EC-Earth3-GrIS. The parameterization allows the dependence of snow albedo on snow aging, melt and refreezing. For fresh





snow a maximum value of 0.85 is used. Under dry non-melting conditions, ageing may reduce the snow albedo to 0.75 and during snow melt the albedo decreases to a lower limit of 0.6. The albedo of refrozen meltwater is set to 0.65.

The new land ice physics described above is used for EC-Earth3 low resolution configurations, in particular for PMIP experiments. In this case, there is no coupling to the ice sheet model. Instead, the ice sheet mask can be either read in as
boundary conditions or defined by snow depth exceeding a certain threshold (9 meters).

The fields exchanged between EC-Earth and PISM are listed in Table 10. Information is exchanged once a year with monthly variations. IFS provides forcing fields of surface mass balance (SMB) and subsurface temperature to PISM. The SMB is calculated from precipitation, evaporation and run-off. PISM returns the ice topography and ice mask to IFS and the calving (mass and energy) and basal melt (mass) fluxes to NEMO.

**2.6 HPC performance of different configurations**

The increasing capabilities of ESMs, such as EC-Earth3, and the ability to perform large community experiments, such as CMIP6, are strongly linked to the amount of HPC capacity available and to the efficient use of these resources. As such, CMIP6 is an excellent opportunity to study the computational performance of ESMs, in particular for models such as EC-Earth3 which are developed and used by a wide range of institutions and integrated on different computational platforms.
The computational performance of EC-Earth3 has been evaluated in order to achieve different goals:
- to detect performance bottlenecks for future improvements,
- to compare the performance of different computational platforms used by the consortium and evaluate how different hardware can affect the performance of EC-Earth,
- to compare different model configurations to analyze which components or calculations represent bottlenecks in the execution.

A first optimization and performance analysis of a preliminary version of EC-Earth3 (EC-Earth3P-HR) was presented in Haarsma et al. (2020). This particular version, which was used in the context of the H2020 PRIMAVERA project (Roberts et
al., 2018), was integrated at both standard and high resolutions, following the HighResMIP protocol (Haarsma et al. 2016). In Haarsma et al. (2020), the high resolution configuration was analyzed in order to detect performance bottlenecks and to provide solutions for these. The high resolution was used for this purpose because of easier detectability of problems related to the scalability and computational efficiency.

The rest of this section will focus on the performance of the standard resolution version of EC-Earth3 in order to fulfill the second and third goals presented.

The evaluation was done through a set of metrics independent of the platform and of the underlying parallel programming models. To make this possible, the EC-Earth standard resolution configuration discussed hereafter was analyzed through 430 CPMIP, a computational performance model intercomparison project (MIP) presented by Balaji et al. (2017).

This analysis is done in two levels. The first level (Table 11) includes basic performance metrics for four different platforms (Rhino (RN), Marenostrum4 (MN4), ECMWF-CCA (CCA) and Beskow (BK)), in order to compare the performance of two configurations (EC-Earth3 and EC-Earth-Veg) on those different platforms. The second level (Table 12) includes the 435 complete set of CPMIP metrics collected on Marenostrum4, for EC-Earth3.

In Table 11. SYPD measures the model speed by counting the number of years the model could simulate within a 24-hour period, given a certain configuration and computational platform. ASYPD is measured for a long-running experiment and includes queueing time, taking into account the sharing of HPC resources. CHSY measures the computational cost of the 440 model, for the given configuration and computational platform. Finally, Parallelization represents the number of MPI processes used.

Comparing EC-Earth3 on two platforms (MN4 and RN) with similar parallelization, BullX B500's experiment is slightly faster than LENOVO SD530 experiment but it also uses more resources and as a consequence the CHSY is slightly higher. 445 On the other hand we can obtain similar performance on BullX B500 and LENOVO SD530 experiments, even though the BullX B500 experiment is run on a platform with technology five years older, proving that configurations without an expensive computation can be simulated efficiently in more commodity clusters. Obviously, as shown in Haarsma et al. (2020), the performance of more demanding configurations will be affected by several issues such as the MPI communications overhead and a better network will ensure that better hardware will obtain better performance too. Finally, 450 the experiment of EC-Earth3 on CCA proves that the user can achieve a similar efficiency using a setup with fewer processes and obtaining a similar CHSY, though the results will need more time to be executed.

It is important to note that the workflow of these experiments comprises different steps, with dependencies between them. This is especially true when the storage is a constraint and simulation steps that need data from some prior steps before it has 455 been post-processed and. In such cases, the way these dependencies are handled may have an impact on the overall throughput.

LENOVO SD530 experiments at BSC were run using a workflow management tool, called Autosubmit. This tool handles dependencies in an automatic way and is able to pack multiple tasks or simulation steps in the same job execution, which 460 may reduce the number of job queuing and thus have an impact on the ASYPD. This does not necessarily explain the





differences between the three platforms in the study, given the different use policies, load on the machine from other users, scheduling parameters and usage existing among them.

The BullX B500 and CRAY XC40 (on BK platform) experiments are not directly comparable because the CRAY XC40 experiment includes LPJ-GUESS as a vegetation component. Both simulations use the same parallel resources and the performance of CRAY XC40 experiment on BK is lower. The results suggest that LPJ-GUESS is less efficient than the other components present in the standard configuration of EC-Earth and this difference in performance is largely due to the way the output is performed. The problem is to be studied to improve it in the future. A new approach is under development to improve the computational efficiency of LPJ-GUESS. On the other hand, the EC-Earth-Veg configuration run in the CRAY XC40 experiment on CCA suggests that when the execution time of IFS and NEMO components is long enough (since we are using less parallel resources for their execution), the LPJ-GUESS component is not a bottleneck anymore, achieving a CHSY only slightly higher. However, the single point to take into account in this case is that the user will need more time to finish the simulations, since the SYPD is lower compared to the set up used on BK platform.

These results will be used to compare the computational performance of EC-Earth with other models running the same CMIP6 configuration or with a similar complexity. However, preliminary results from the collection (provided by other institutions) prove that the efficiency of EC-Earth (comparing CHSY among models with similar complexity or number of grid points) seems to be on the average with good results on the computational performance side. The cost of indirect processes such as coupling or output costs is also similar to the results obtained by other models.

## 3. The component models

### 3.1 Atmosphere

The atmosphere component of the EC-Earth model is based on the Integrated Forecast System (IFS) CY36R4 of the European Centre for Medium Range Weather Forecasts (ECMWF). This specific cycle of the IFS has been part of ECMWF's operational seasonal forecast system S4 (https://www.ecmwf.int/sites/default/files/elibrary/2011/11209-new-ecmwf-seasonal-forecast-system-system-4.pdf). IFS solves the hydrostatic primitive equations using a two-time-level, semi-implicit semi-Lagrangian discretization. Horizontal derivatives are computed in spectral space while the computation of advection, the physical parameterizations, and in particular the nonlinear terms are conducted on the linear reduced Gaussian grid. The IFS is documented extensively at https://www.ecmwf.int/en/publications/ifs-documentation, for example https://www.ecmwf.int/sites/default/files/elibrary/2010/9232-part-iii-dynamics-and-numerical-procedures.pdf for the dynamics and https://www.ecmwf.int/sites/default/files/elibrary/2010/9233-part-iv-physical-processes.pdf for the physical processes. Here we only document the updates to the original IFS that were necessary for making long climate simulations.



The physical aspects of the atmosphere model in EC-Earth needed some adjustments and updates compared to the original IFS CY36R4. Most of these modifications are not necessary for Numerical Weather Forecast (NWP) or even seasonal

forecasts but are crucial when running long climate simulations, decadal, centennial and longer, or simulations under different climate conditions (e.g. future scenarios or paleo simulations).

The semi-Lagrangian advection scheme of IFS does not conserve mass nor energy in the NWP version. A dry air mass conservation fixer has been available in IFS since CY25R1 and is active in EC-Earth to correct global pressure for the gain

or loss of atmospheric mass. Similarly, to conserve humidity during transport we backported a simple proportional fixer from IFS cycle CY38R1 (Rasch and Williamson 1990, Diamantakis and Flemming 2014). This significantly reduced the bias of the average global precipitation-evaporation balance in the model from about +0.030 mm/day to -0.017 mm/day and, consistently (due to the associated latent heat of condensation), in the radiative balance in the atmosphere from about -1.65 W/m^2 (a source of energy) to about -0.25 W/m^2 .


The IFS CY36R4 version adopted for EC-Earth3 produces a reasonable Quasi-Biennial Oscillation (QBO) in the tropical stratosphere when running at the standard resolution (T255L91), but not for any other available horizontal or vertical resolutions. Therefore we substituted the original version-dependent latitudinal profile of the momentum flux in the non-orographic gravity wave scheme (which was originally developed ad-hoc for the ECMWF System 4 seasonal forecast

system), with a resolution-dependent parameterisation of non-orographic gravity wave drag, backporting changes later introduced in IFS CY40R1 (see Davini et al. 2017a for more details). This change allowed EC-Earth to recover a realistic QBO at all resolutions considered, without deteriorating the jet streams.

Convection in the NWP version of IFS CY36R4 reaches its maximum around local noon in contrast to observations that

peak later in the afternoon. A closure described by Bechtold et al. (2014) improves the diurnal cycle of convection in EC-Earth3. For EC-Earth3, Rayleigh friction was activated in EC-Earth IFS for all resolutions.

In atmosphere-only simulations, the sea-ice albedo is taken from a look-up table with climatological monthly values for sea-ice albedo (Ebert and Curry, 1993) that takes into account the annual cycle of highly reflective snow cover during winter and

spring and the darker surface of melting sea ice during summer. In the coupled model, the sea-ice albedo is computed in the sea-ice model LIM3 and the updated values are used by the atmospheric component. The broadband sea-ice albedo from LIM3 is then mapped on 6 shortwave bands with a mapping function.

The time stepping scheme needed technical adjustments to avoid an overflow of integer timestep counters, in order to allow

making simulations beyond 32768 timesteps. The IFS output is saved in the GRIB1 data format which also has a limit in the



number of timesteps that can be saved. This limit was overcome in EC-Earth3 by setting the timestep to 0 and updating the GRIB encoded reference time instead, each time that output is written.

CMIP6 requires transient climate forcings to account for the change in atmospheric composition and other external drivers of the climate (e.g. insolation). The necessary interfaces to read the prescribed greenhouse gas concentrations, aerosol optical properties, stratospheric aerosols, stratospheric ozone, and insolation have been implemented in the IFS code in EC-Earth. Table 13 lists the sources and versions of the CMIP6 forcing datasets.

Well-mixed greenhouse gases (WMGHGs) explicitly included in EC-Earth's radiation scheme are $CO_2$, $CH_4$, nitrous oxide ($N_2O$), CFC-12, and CFC-11. Together these are responsible for about 98% of the total radiative forcing by WMGHGs in 2014 compared to 1850 (Meinshausen et al., 2017). The radiative effects of the remaining WMGHGs (HCFC-22, CFC-113, $CCl4$, etc.) are accounted for in terms of CFC-11 equivalents (Meinshausen et al., 2017). The mixing ratios of each of the WMGHGs that are explicitly included and not provided by TM5 are prescribed by scaling their monthly zonal mean climatologies as used in IFS by a single time-dependent global factor. In this way, the global mean surface mixing ratios are
forced to their CMIP6 pathways (Meinshausen et al., 2017). To reduce discontinuities, the scale factors are calculated on a monthly basis by interpolation of the time series of annual values provided by CMIP6. Any delays due to transport from the surface to the upper parts of the atmosphere are ignored in this approach.

      Tropospheric aerosols are either simulated interactively in TM5 (in the EC-Earth3-AerChem configuration) or prescribed as
a pre-industrial climatology plus an anthropogenic contribution (all other configurations). The pre-industrial aerosol background is specified using a monthly climatology based on TM5. This climatology was obtained from an offline TM5 simulation driven by ERA-Interim meteorology for the years 1981-1985, using CMIP6 anthropogenic emissions for the year 1850. The radiative and cloud effects of the pre-industrial aerosols are calculated based on the ERA interim reanalysis and the same set of variables as when aerosols are interactively simulated by TM5. The anthropogenic contribution is specified
following the simple plume approach of MACv2-SP (Stevens et al., 2017), which provides a simplified, parametric representation of the optical properties (extinction, single-scattering albedo and asymmetry factor) of the anthropogenic contribution to the tropospheric aerosol burden (relative to 1850 levels), consistent with the CMIP6 time series of historical (Stevens et al., 2017) and future (Fiedler et al., 2019) anthropogenic emissions. In EC-Earth, MACv2-SP is coupled with the IFS radiation scheme to compute the optical properties for the 14 wavelength bands of the SW radiation. More precisely, the
optical properties are calculated at the band mean wavelengths weighted by the incoming solar radiation. In addition, MACv2-SP provides a simple way to account for the effect of anthropogenic aerosols on clouds. Specifically, it provides a scale factor for the cloud droplet number concentration (CDNC) in each column, based on the vertically integrated optical depth at 550 nm.



In the EC-Earth3-AerChem, aerosol impacts on clouds are included by calculating CDNC depending on the modal number and mass concentrations from TM5, following Abdul-Razzak and Ghan (2000). For all other model configurations  the CDNC corresponds to pre-industrial aerosol conditions, and an additional scaling factor from MACv2-SP that is included to account for the cloud forcing by anthropogenic aerosols. The resulting forcing includes contributions due to both cloud reflectivity and cloud lifetime effects, as the lifetime of clouds explicitly depends on CDNC. Currently only the activation

and autoconversion of liquid cloud droplets is linked explicitly to ambient aerosol concentrations. For ice clouds the EC-Earth3 model still retains the parameterization from original IFS CY36R4.

Stratospheric aerosols are prescribed using the CMIP6 data set of aerosol radiative properties, which covers the period 1850 to 2014 and for the more recent period is based on satellite data assembled by Thomason et al. (2018). The data set consists

of monthly resolved zonal mean fields, which are provided at the 14 shortwave (SW) and 16 longwave (LW) bands of the IFS's radiation schemes. For the SW scheme, the extinction, single-scattering albedo and asymmetry factor are specified, whereas only the absorption is taken into account for the LW scheme, since aerosol scattering in the LW is neglected in the atmospheric component of EC-Earth. Forcing data are vertically interpolated beforehand for the 62 and 91 level configurations, taking into account the seasonality of model level heights, whereas horizontal and monthly to daily

interpolation is done online. When interpolating or averaging the radiative property fields, they are first made extensive by including the appropriate weighting factors (e.g. extinction is converted to optical depth, single-scattering albedo to absorption optical depth, and likewise for the asymmetry factor). The forcing located below the online diagnosed thermal tropopause level is excluded.This implementation is used in all current EC-Earth3 configurations with the exception of the EC-Earth3P-HR configuration, which uses a simplified implementation based on a monthly vertically integrated, latitude-

dependent AOD forcing at 550 nm which is then vertically distributed across the stratosphere. In both implementations, it is possible to set the forcing fields to a constant background distribution, computed as the time average over 1850 to 2014. This background forcing is applied in preindustrial control and future simulations, as recommended in the CMIP6 protocol.

The land use forcing dataset (LUH2) from CMIP6 (Hurtt et al. 2020) cannot be used directly as input to IFS because it does

not provide the same vegetation cover or type categories as those used by the land surface scheme in IFS (HTESSEL, van den Hurk et al., 2000; Balsamo et al., 2009; Dutra et al. 2010; Boussetta et al., 2013) but instead provides agricultural management information and land-use transitions that are annually updated. The vegetation cover, leaf area index (LAI) and vegetation type that are needed for the land surface scheme and albedo parameterisation in IFS can be simulated by the dynamic vegetation model LPJ-GUESS (Smith et al., 2014). This happens automatically in the EC-Earth3-Veg configuration

where the dynamic vegetation model, which uses the LUH2 dataset as an input, is active, but for all other configurations the required vegetation cover and type need to be precomputed. This is done by first making all CMIP6 experiments with the EC-Earth3-Veg configuration and saving the vegetation variables that can then be reused when making the same experiment with other model configurations





The orbital parameters of the original IFS CY36R4 are fixed for present-day conditions, following the recommendations of the International Astronomical Union (ARPEGE-Climate Version 5.1, 2008), which is sufficient for simulations of the recent past or near future. However for paleo simulations in PMIP the orbital parameters need to be variable or set fixed for a different time period. Orbital parameters and insolation are computed using the method of Berger (1978). Using this formulation, the insolation can be determined for any year within $10^6$ years of 1950 AD. The formulation determines earth-

sun distance factor and solar zenith angle. The annual and diurnal cycle of solar insolation are represented with a repeatable solar year of exactly 365 days and with a mean solar day of exactly 24 hours, respectively. The repeatable solar year does not allow for leap years. The orbital state may be specified in one of two ways. The first method is to specify a year, which is held constant during the integration for an equilibrium simulation, or varies yearly for a transient simulation. The second method is to specify the orbital parameters: eccentricity, longitude of perihelion, and obliquity. This set of values is

sufficient to specify the complete orbital state. For example, settings for PiControl integrations under 1850 AD conditions are obliquity = 23.549, eccentricity = 0.016764, and longitude of perihelion = 100.33.

### 3.2 Land surface and vegetation

The Hydrology Tiled ECMWF Scheme of Surface Exchanges over Land (HTESSEL; van den Hurk et al., 2000; Balsamo et al., 2009; Dutra et al. 2010; Boussetta et al., 2013) is the land surface model interfacing with the atmospheric boundary layer

and solving the energy and water balance at the land surface in EC-Earth. HTESSEL discretization, for each grid point, solves for up to six different land surface tiles that may be present over land (bare ground, low and high vegetation, intercepted water by vegetation, and vegetation-shaded and exposed snow). Surface radiative, latent heat and sensible heat fluxes are calculated as a weighted average of the values over each tile.

The discretization in HTESSEL is such that coexistence in each grid point of more than one type of low and high vegetation, respectively, is not allowed. Therefore, for each grid-point and for both low and high vegetation covers, a dominant type (dominant meaning type with the higher relative area fraction for either high or low vegetation) is identified, $T_l$ and $T_h$, and a vegetation coverage for high and low vegetation types, $C_h$ and $C_l$, is specified.

Vegetation types and vegetation coverage can be
1. prescribed from a static land-use map from the Global Land Cover Characteristics (GLCC, standard HTESSEL configuration; van den Hurk et al., 2000; Balsamo et al., 2009; Dutra et al. 2010; Boussetta et al., 2013); or
2. interactively provided when coupled with LPJ-GUESS; or
3. prescribed from a previous simulation with LPJ-GUESS.


When the tile fractions are prescribed from GLCC, vegetation density is parameterized according to the Lambert Beer law of extinction of light under a vegetation canopy and is therefore allowed to change as a function of Leaf Area Index (LAI) for

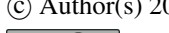



both low and high vegetation as described in Alessandri et al (2017). Otherwise, LPJ-GUESS provides its own consistently-simulated background tile fractions and vegetation densities.


The coupling of biophysical parameters in HTESSEL has been enhanced since CMIP5 (Weiss et al., 2013), where only the surface resistance to evapotranspiration and water intercepted and directly evaporated from vegetation canopies were made to depend on LPJ-GUESS vegetation dynamics. In the version for CMIP6, as used in EC-Earth3-veg, the surface albedo (including the shading effect of high vegetation), surface roughness length and soil water exploitable by roots for

evapotranspiration also vary following the variability of the effective vegetation cover. The improved representation of the effective vegetation cover variability brought a significant enhancement of the EC-Earth performance over regions where the land-atmosphere coupling is strong, in particular over boreal winter middle-to-high latitudes (Alessandri et al., 2017).

To represent time-dependent albedo for each grid-point, a new scheme has been adopted that computes the total surface albedo ($A_{tot}$) as a weighted combination of contributions from the albedo of the low and high vegetation types present in each

grid point ($a_v(type)$, a function of the low or high vegetation type) plus a time-constant background soil albedo ($a_s$, function of space):

$$A_{\mathrm{tot}} = a_v(T_l)C_{\mathrm{low}}^{\mathrm{eff}} + a_v(T_h)C_{\mathrm{high}}^{\mathrm{eff}} + a_s \left[1 - C_{\mathrm{low}}^{\mathrm{eff}} - C_{\mathrm{high}}^{\mathrm{eff}}\right]$$

where $C^{\mathrm{eff}}_{\mathrm{low}}$ and $C^{\mathrm{eff}}_{\mathrm{high}}$ are the effective fractional coverages for low/high vegetation and $T_l$ and $T_h$ are the low and high vegetation types respectively at each gridpoint. The background soil albedo was adopted from the map from Rechid et al. (2009) and a look-up table of the albedo values $a_v$ for each vegetation type was estimated using least square minimization of errors against available monthly climatology of snow-free monthly MODIS albedo (Morcrette et al., 2008).

**3.3 Dynamic vegetation and terrestrial biogeochemistry**

LPJ-GUESS (Smith et al. 2001, 2014; Lindeskog et al. 2013; Olin et al. 2015a, b), a process-based 2nd generation dynamic vegetation and biogeochemistry model, is the terrestrial biosphere component of EC-Earth, globally simulating vegetation dynamics, land use and land management following the LUH2 dataset (Hurtt et al., 2020), and both carbon (C) and nitrogen (N) cycling in terrestrial ecosystems. LPJ-GUESS has been evaluated in numerous studies (Smith et al. 2014; Wårlind et al.

2014), and reproduces vegetation patterns, dynamics and productivity, C and N fluxes and pools, and hydrological cycling from global to regional scales, in line with independent datasets and comparable models (e.g. Piao et al. 2013; Zaehle et al. 2014; Sitch et al. 2015, Peters et al. 2018).



LPJ-GUESS is a new component in EC-Earth3 (Miller et al., in prep), though it has previously been coupled to EC-Earth
v2.3 (Weiss et al. 2012; Alessandri et al. 2017) using a simplified coupling scheme in which updates to leaf area index (LAI)
alone were transferred between the submodels.

LPJ-GUESS is one of the first vegetation submodels coupled interactively to an atmospheric model, in which the size, age
structure, temporal dynamics and spatial heterogeneity of the vegetated landscape are represented and simulated
dynamically. Such functionality has been argued to be essential for correctly capturing biogeochemical and biophysical land-
atmosphere interactions on longer timescales (Purves and Pacala 2008; Fisher et al. 2018), and has been shown to improve
realism compared with more common area-based vegetation schemes (Wolf et al. 2011; Pugh et al. 2018). Different plant
functional types (PFTs) co-occur in natural and managed stands governed by climate, atmospheric $CO_2$ (Meinshausen et al.,
2017; Riahi et al., 2017), and N deposition (Hegglin et al. GMD, in prep) forcings. Evolving stand structure impacts growth,
survivorship and the outcome of competition by affecting the availability of the key resources: light, space, water and
nitrogen. Disturbances due to management actions such as forest clearing, prognostic wildfires and a stochastic generic
disturbance regime affect patches at random, inducing biomass loss and resetting vegetation succession (Hickler et al. 2004).
N cycle-induced limitations on natural vegetation and crop growth, C-N dynamics in soil biogeochemistry and N trace gas
emissions are included (e.g. Smith et al. 2014; Olin et al. 2015a, b) as well as biogenic VOC emissions (Hantson et al. 2017).

Meteorological inputs imposed on LPJ-GUESS are daily fields of surface air temperature and 25 cm soil temperatures,
precipitation, net shortwave and net longwave radiation from IFS/HTESSEL (Table 5). LPJ-GUESS calculates its own soil
moisture for potential plant uptake in all patches in each of the six simulated stands, independently of the single grid cell-
averaged hydrology scheme used in HTESSEL.

Vegetation dynamics are simulated on six stand types in the land portion of the gridcell (excluding large water bodies based
on the static LUH2 ice and water fraction information), five stands having dynamic gridcell fractions consistent with the
LUH2 dataset, namely Natural, Pasture, Urban, Crop, and Irrigated Crop, and one, Peatland, having a fixed gridcell fraction
derived from the GLCC global map used in the standard HTESSEL configuration - see Sec. 3.2. The LUH2 dataset,
including land cover fractions, management options (N fertilization in this case) and land cover transitions, are read in
yearly, after aggregation to the atmospheric and land surface model resolution in a preprocessing step. Ten woody and two
herbaceous PFTs compete in the Natural stand (Smith et al. 2014), whereas two herbaceous species, one each conforming to
the C3 and C4 photosynthetic pathways, are simulated on Pasture, Urban and Peatland fractions. The Crop stands each have
five crop functional types (CFTs) representing the properties of global crop types and encompassing the classes found in the
LUH2 database, namely both annual and perennial C3 and C4 crops, and C3 N fixers (Lindeskog et al. 2013).

At the end of each day, LPJ-GUESS calculates the effective cover for low (high) vegetation, $C_l$ ($C_h$), and LAI for low (high)
vegetation, $LAI_{low}$ ($LAI_{high}$), taking into account phenology and stand fractions in the gridcell. Dominant high and low





vegetation types corresponding to the standard HTESSEL types are calculated and sent by LPJ-GUESS to IFS/HTESSEL on Dec 31st each year. These six fields link the vegetation dynamics and land use in LPJ-GUESS to the biophysical processes

simulated at the land surface in HTESSEL, namely albedo, latent and sensible heat exchange, runoff and momentum exchange.

In the EC-Earth-CC configuration, LPJ-GUESS is coupled to TM5 in addition to IFS, and exchanges additional fields to enable prognostic global C cycle calculations. Spatiotemporally variable surface $CO_2$ concentrations are sent by TM5 to

LPJ-GUESS (and PISCES) to replace the annual and global mean $CO_2$ concentrations used in the EC-Earth-Veg configuration. LPJ-GUESS sends daily averaged fields of net ecosystem C exchange (i.e. uptake or release) to TM5 to complement the surface C exchange with the ocean calculated in PISCES (see below), thereby completing the carbon cycle in EC-Earth-CC. This daily flux includes contributions from net primary production (NPP), heterotrophic respiration (Rh), wildfires, land use (including crop and pasture harvest) and natural disturbances on non-managed land. Since some processes

in LPJ-GUESS are simulated with a yearly timestep (e.g. wildfires, disturbance, establishment of new individuals and mortality, land use change), these annual fluxes are distributed evenly throughout the year and added to the daily NPP and Rh fluxes the following year to conserve carbon mass. Negative NPP fluxes account for CO2-uptake by vegetation.

### 3.4 Atmospheric chemistry

The Tracer Model version 5 (TM5) is the atmospheric composition model of EC-Earth (Van Noije et al., 2014) used in the EC-Earth3-AerChem and EC-Earth3-CC configuration. It can be used for the interactive simulation of carbon dioxide ($CO_2$), methane ($CH_4$), ozone ($O_3$), tropospheric aerosols, and other trace gases. These components are prescribed in IFS from forcing datasets (see Sec 3.1) if not provided interactively by TM5. Other well-mixed greenhouse gases and stratospheric aerosols are prescribed in all configurations. This section briefly describes how the various components are configured.


As an alternative to the scaling approach for WMGHG presented in 3.1, the 3-D distributions of $CO_2$ and $CH_4$ can be calculated online by TM5. In the EC-Earth-CC configuration a single-tracer version of TM5 is used for simulating the transport of $CO_2$ through the atmosphere. Anthropogenic emissions of $CO_2$ are prescribed following the CMIP6 historical inventory (Hoesly et al., 2018) or future scenarios (Gidden et al., 2019). Exchange of $CO_2$ with the ocean and terrestrial

biosphere is included by coupling TM5 to PISCES and LPJ-GUESS, respectively (see Section 3.3). An important feature of the model is that the transport in TM5 is mass conserving (Krol et al., 2005). For the simulation of $CH_4$, a version of TM5 that includes atmospheric chemistry and aerosols is used (Van Noije et al., 2020). A recent description of the chemistry scheme applied in EC-Earth has been presented by Williams et al. (2017). Emissions of aerosols and chemically reactive gases are taken from the CMIP6 historical data sets for anthropogenic sources (Hoesly et al., 2018) and biomass burning

(van Marle et al., 2017) or the corresponding CMIP6 scenario data sets (Gidden et al., 2019). To force the $CH_4$ simulation to





follow the pathway provided by CMIP6, its surface mixing ratios are nudged towards the monthly zonal means from CMIP6 interpolated to daily values. Moreover, because TM5 lacks a comprehensive stratospheric chemistry scheme, the $CH_4$ mixing ratios in the stratosphere are nudged towards a monthly zonal mean observational climatology representative for the 1990s (interpolated to daily values), scaled by a global factor based on the CMIP6 time series of global annual mean surface values.

To calculate the scale factor, we assume a one-year delay between the mixing ratios at the surface and in the stratosphere (Meinshausen et al., 2017), and a reference value based on a 10-year average.

The chemical production of water vapour ($H_2O$) by oxidation of methane in the stratosphere is included in IFS in a similar way as in the standard version of IFS. The assumption made in the standard version of IFS is that


$$2 \times [CH4] + [H2O] = Co,$$

where square brackets denote local mixing ratios (in ppmv) and the constant is set to 6.8 ppmv based on observations for the present day. To account for long-term variations in $CH_4$, in EC-Earth it is assumed instead that


$$2[CH4]+[H2O]=C(t),$$

where

$$C(t)=Co+2([CH4]S(t)-[CH4]0S).$$

Here [CH4]S(t) is the monthly-varying global mean surface mixing ratio obtained by linear interpolation from the CMIP6 time series of annual values, and [CH4]0S is a reference value for the present day, which is set to 1.78 ppmv.

Ozone is simulated by TM5. As for $CH_4$, TM5 applies a nudging scheme for $O_3$ in the stratosphere. In EC-Earth3, the mixing ratios are nudged towards daily zonal means obtained from the CMIP6 data set.


For aqueous-phase chemistry in the troposphere, the acidity of cloud droplets is calculated assuming a uniform $CO_2$ mixing ratio, following the CMIP6 time series of annual global mean surface values.

TM5 simulates tropospheric aerosols, namely sulphate, black carbon, primary and secondary organic aerosol, sea salt and

mineral dust in four size ranges describing nucleation, Aitken, accumulation and coarse modes, using the M7 aerosol microphysical model (Vignati et al., 2004). In addition, it simulates the total mass of ammonium, nitrate and methane sulfonic acid (MSA). Optical properties of the aerosol mixture are calculated based on Mie theory in combination with the mixing assumptions described by Van Noije et al. (2014).



For calculation of the SW radiative effects of the aerosol mixture, TM5 provides the extinction, single-scattering albedo and asymmetry factor at the 14 wavelength bands of the SW radiation scheme (using the same wavelength values as in MACv2-SP). In addition, TM5 provides the particle number and component mass mixing ratios for each of the M7 modes, plus the total mass mixing ratios of nitrate and MSA. LW absorption is calculated based on the mass mixing ratios of the M7 components using absorption efficiencies from IFS. The contribution of the aerosol mixture described by MACv2-SP and/or

TM5 to SW extinction and LW absorption is removed above the tropopause, where the stratospheric aerosol data set from CMIP6 is applied. The tropopause level is diagnosed online following the thermal tropopause definition of the World Meteorological Organization (WMO, 1957) as detailed by Reichler et al. (2003). Where the thermal tropopause does not exist according to this definition, tropospheric and stratospheric aerosols are merged at the 100 hPa level.

**3.5 Ocean**

The ocean component of the EC-Earth model is the Nucleus for European Modelling of the Ocean (NEMO; Madec 2008, Madec et al., 2015) that includes the ocean model OPA (Océan Parallélisé), the LIM3 sea ice model (see section 3.6) and the PISCES biogeochemistry model (see section 3.7). The CMIP6 version of the EC-Earth model uses NEMO3.6 (revision r9466) in combination with the ORCA1 shared configuration.

OPA is a primitive equation model of ocean circulation. Prognostic variables are velocity, hydrostatic pressure, sea-surface height and thermohaline variables (potential temperature and salinity). The distribution of variables is given by a three-dimensional Arakawa-C-type grid (Arakawa and Lamb 1977). OPA uses a partial step implementation for the geopotential z*-coordinate (grid boxes do not continue below topography) and a diffusive bottom boundary layer scheme (similar to that of Beckmann and Döscher, 1997) with implicit bottom friction to mix dense water down a slope.


NEMO allows for various choices for the physical sub-gridscale parameterizations as well as the numerical algorithms. EC-Earth uses the Turbulent Kinetic Energy (TKE) scheme for vertical mixing. The vertical eddy viscosity and diffusivity coefficients are computed from a 1.5 turbulent closure model based on a prognostic equation for the turbulent kinetic energy, and a closure assumption for the turbulent length scales. This turbulence closure model has been developed by Bougeault

and Lacarrère (1989) in atmospheric cases, adapted by Gaspar et al. (1990) for oceanic cases and embedded in OPA by Blanke and Delecluse (1993).

Since the CMIP5 version of EC-Earth, major changes in the TKE schemes have been implemented: it now includes a Langmuir cell parameterization (Axell 2002), the Mellor and Blumberg (2004) surface wave breaking parameterization, and

has a time discretization which is energetically consistent with the ocean model equations (Burchard 2002, Marsaleix et al. 2008). A mixed layer eddy parameterization following Fox-Kemper et al. (2008) has been newly implemented in NEMO3.6.





An enhanced vertical diffusion and a double diffusive mixing parameterization are part of the OPA code in EC-Earth. Since CMIP5, a tidal mixing parameterization has been added to OPA (de Lavergne et al. 2020).

Horizontal tracer diffusion is described by the Gent-McWilliams (Gent and McWilliams 1990)   parametrization of
mesoscale eddy-induced turbulence.

The ORCA family is a series of global ocean grid configurations. The ORCA grid is a tripolar grid and based on the semi-analytical method of Madec and Imbard (1996). ORCA1 with a resolution of about 1 degree is used for standard or low resolution simulations, and ORCA025 (resolution 0.25 deg) for high resolution simulations with EC-Earth3-HR. A
meridional grid refinement of 1/3 deg in the tropics allows a partial representation of tropical instability waves. There are 75 vertical levels in the ocean with an upper level of about 1 m and 24 levels distributed over the uppermost 100m.

The main difference of the OPA-version used in EC-Earth compared to the reference OPA-version of NEMO3.6 is that the parameterization of the penetration of TKE below the mixed layer due to internal and inertial waves is switched off
(nn_etau=0). This has been done because the penetration of TKE below the mixed layer caused a too deep surface layer of warm summer water masses in the North Atlantic convection areas which lead to a breakdown of the Labrador Sea convection within a few years and a strongly underestimated Atlantic Meridional Overturning Circulation (AMOC) in EC-Earth. A minor modification compared to the standard NEMO setup from the ORCA1-shared configuration for NEMO (ShacoNemo) is an increased tuning parameter rn_lc (=0.2) in the TKE turbulent closure scheme that directly relates to the
vertical velocity profile of the Langmuir Cell circulation. Consequently, the Langmuir Cell circulation is strengthened.

### 3.6 Sea ice

The sea ice component is version 3.6 of the Louvain-la-Neuve Ice Model (LIM, Vancoppenolle et al., 2009; Rousset et al, 2015), which works directly on the NEMO environment, including the ORCA grid. LIM3.6 is based on the Arctic Ice
Dynamics Joint EXperiment (AIDJEX) framework (Coon et al., 1974), combining the ice thickness distribution (ITD) framework, the conservation of horizontal momentum, an elastic-viscous plastic rheology, and energy-conserving halo-thermodynamics (Vancoppenolle et al., 2009). All of these components of the sea ice model have been introduced or revised since CMIP5.

The ice thickness distribution framework was introduced (Thorndike et al, 1975) to deal with meter-scale variations in ice thickness, which cannot be resolved explicitly, but should preferably be accounted for, as many sea ice processes, in particular growth and melt, depend non-linearly on thickness $h$. In practice, this is achieved by treating $h$ as an independent variable, leading to the introduction in discrete form of $L=5$ thickness categories, each characterized by a specific set of state variables (namely ice concentration, ice volume per unit area, snow volume per unit area, ice enthalpy, snow enthalpy, sea





ice salt content). Ice and snow enthalpy also depend on vertical depth in the ice (z). All sea ice state variables *Xijl, l=1,..., L* are updated due to transport and thermodynamic processes. The default choice of 5 categories, with the upper category above 4 meters, has been shown to provide reasonable results at an acceptable computing cost (Massonnet et al., 2019).

Vertical sea ice motions are irrelevantly small and hence neglected, and the sea ice velocity field reduces to its horizontal
components. The 2D ice velocity vector is considered the same for all categories and stems from the horizontal momentum conservation equation. The internal stress term is formulated assuming that sea ice is a viscous-plastic material, i.e., assuming viscous ice flow at very small deformation and plastic flow (stress independent of deformation) above a plastic failure threshold. This threshold lies on an elliptical yield curve in the principal stress components space, whose size can be changed by tuning classical ice strength parameter $P^* = 20000$, following the classical formulation of Hibler (1979). The
horizontal momentum equation is resolved using the Elastic-Viscous-Plastic (EVP) C-grid formulation of Bouillon et al (2009), using 120 sub-time steps. Once the velocity field is computed, the sea ice state variables are transported horizontally, using the second-order moment-conserving scheme of Prather (1986).

Ice thermodynamics are based on the Bitz and Lipscomb (1999) enthalpy formulation, and account for dynamic changes in
ice salinity, through temperature and salinity-dependent thermal properties (Ono, 1967; Pringle et al., 2007). The salt entrapment and drainage parameterizations follow from Vancoppenolle et al. (2009): each category is characterised by a dynamic mean salinity, from which a profile shape is derived for the computation of the vertical diffusion of heat. The broadband surface albedo of each ice category empirically depends on ice thickness, snow depth, surface temperature and cloud fraction, based on a reformulation of the Shine and Henderson-Sellers (1985) parameterization, that solves a few
inconsistencies associated with state transitions (e.g. snow / no snow) following Grenfell and Perovich (2004), and tuned to match observations of Brandt et al. (2005). The impact of melt ponds is implicitly accounted for through imposed changes on the albedo activated when the surface temperature is 0°C. Energy, salt and mass conservations have been carefully checked within the ice component and its interfaces with the atmosphere and ocean (Rousset et al 2015).

All surface fluxes are computed in the atmosphere, and the IFS atmospheric model has only one ice thickness category. The solar and non-solar heat fluxes are therefore distributed on the different sea ice categories in LIM3, taking into account the differences in albedo and temperature among the sea ice categories in each gridpoint.

During the tuning phase it was found that the Arctic sea ice volume grew to unrealistically high values, especially during
phases with reduced AMOC. An analysis showed that the thermal conductivity of snow needed a slight reduction (rn_cdsn=0.27) to reduce basal growth and increase bottom melt (see also section 2).





### 3.7 Ocean biogeochemistry

PISCES-v2 (Pelagic Interactions Scheme for Carbon and Ecosystem Studies volume 2) is a biogeochemical model that simulates the nutrient cycle and the inorganic and organic carbon cycle and comprises lower trophic phytoplankton and

zooplankton (Aumont et al. 2015). It has two functional groups for phytoplankton (nanophytoplankton, including calcite producers, and Diatoms that can produce siliceous shells) and two size classes for zooplankton (mesozooplankton and microzooplankton). Growth rate of phytoplankton depends on photosynthetic available radiation (PAR) intensity and temperature. A limitation for primary production is computed based on the availability of the main nutrients (P, N, Si, Fe). In case of low nitrate concentrations nitrogen fixation by diazetrophiccyanobacteria is parameterized in waters warmer than

20°C (Aumont et al., 2015). PISCES uses a constant P/N/C ratio of 1/16/122 for primary production. Organic particulate matter produced by food-web processes in the euphotic layer is represented by two size classes. These sink throughout the water column with different velocities while being decomposed into dissolved inorganic nutrients (DIN, DOP) and dissolved inorganic carbon (DIC). A further pool for dissolved organic matter (DOM) is fed by phytoplanktonic exudation and excretion by zooplankton. DOM in PISCES represents only the semi-labile fraction with turnover times ranging from

months to years and it is further remineralized at a constant rate. PISCES includes two different chemistry models to describe iron pools interactions. In EC-Earth3 we use the complex model by Tagliabue and Arrigo (2006). The global river and atmospheric deposition input of nutrients are not balanced to match the fraction lost by sediment burial. For this reason, PISCES allows for a homogeneous correction towards global mean values for alkalinity, nitrate, phosphate and silicate. Furthermore, physical transport of passive tracers presented a slight artificial mass imbalance. To prevent it from becoming

significant for carbon during the spin-up we applied a uniform correction to dissolved inorganic carbon at the end of each year, after taking into account all sources and sinks.

With respect to climate studies PISCES is capable of simulating the relevant processes of the marine carbon cycle, i.e. it comprises the soft-tissue carbon pump, and the carbonate counterpump to realistically simulate the feedback of the marine

carbon cycle to the climate.

The air sea gas exchange for carbon dioxide and oxygen is parameterized according to Wanninkhof (1992). The interface to the seafloor is given by basic assumptions for the exchange between the active sediment layer and the water bottom layer where different assumptions are made for the burial efficiency for silicate, calcite, and particular organic matter (see Aumont

et al. 2015 for further details).

PISCES is part of the community model NEMO and runs on the same model grid. In EC-Earth3 the horizontal resolution is about 1 degree (ORCA1) with 75 vertical levels. Advection/diffusion of the 24 biogeochemical tracers are done in the hydrodynamic ocean model. A detailed description of the PISCES reference version is given in Aumont et al. (2015). In EC-



Earth3 PISCES can be run in passive mode or with feedback to the atmosphere by prognostically simulating air-sea carbon fluxes and contributing to determine atmospheric pCO2 when the global carbon cycle is fully closed in the case that the atmospheric chemistry model TM5 and the terrestrial biosphere model LPJ-GUESS are also enabled. A feedback to the ocean physics is not foreseen, i.e. the thermal effect of light absorption by chlorophyll on water temperature is not communicated to NEMO (although possible).


### 3.8 Greenland Ice Sheet

The Parallel Ice Sheet Model v1.1 (PISM, Bueler and Brown, 2009, and Winkelmann et al., 2011, The PISM Team, 2019) is used in the EC-Earth3-GrIS configuration to model the Greenland ice sheet (GrIS) evolution in the climate system. PISM is an open source model jointly developed by a group of universities, and available from www.pism-docs.org. While all surface

processes over ice sheet (such as the snow layer) are modelled in EC-Earth3, PISM handles the ice sheet dynamical and thermodynamical processes, including ice flow, subglacial hydrology, bed deformation, as well as the basal ice melt.

The spatial domain of PISM is built on a three-dimensional, equidistant polar stereographic grid. The equations are solved with an adaptive time stepping procedure. Boundary conditions include subsurface temperature and mass balance on the ice surface (provided by EC-Earth3), bedrock elevation and bed geothermal heat flux (considered as invariant, geographic

conditions).

PISM considers the ice sheets as a slow, nonlinearly viscous isotropic fluid, characterized by a creeping flow induced by gravitational forces and constrained by the conservation laws of momentum, mass and energy for ice. A combination of two shallow ice approximations, the non-sliding shallow ice (SIA) and the shallow shelf (SSA) approximations is applied, depending on the ice regime (Bueler and Brown, 2009). The former is applied to bed-frozen parts of the ice sheets, while the

latter is applied to ice shelves, and also used as a sliding law in areas with low basal resistance. This hybrid formulation enables the modelling of fast-flowing ice streams and outlet glaciers, and is commonly used for simulations of whole ice sheets for which it is too expensive to solve the full set of stress balance equations.

The ice velocities are determined from geometry (i.e, ice thickness and ice surface elevation), ice temperature and basal strength using momentum/stress balance equations. The ice thermodynamics is formulated as the energy balance based on

enthalpy that enables solutions for polythermal ice masses (Ashwanden et al 2012), and the Glen-Paterson-Budd-Lliboutry-Duval flow law (Lliboutry et al. 1985) that accounts for softening of the ice as the liquid water fraction increases. The ice flow law is a single-power law in which the exponent can be selected independently for the SIA and SSA. Furthermore, an enhancement factor is used to account for the anisotropic nature of the ice.





The subglacial processes are resolved by the sliding law that relates the basal sliding velocity to the basal shear stress. PISM
uses a pseudo-plastic sliding law and the Mohr-Coulomb model for yield stress that depends on the till friction angle and the
effective pressure of the saturated till. The latter is based on a subglacial routing scheme and the basal melt rate is calculated
from energy conservation across the ice-bedrock layer. A geothermal heat flux map is applied at the basal boundary to
account for the heat entering the ice sheet from below. Ice bed deformation is approximated by the viscoelastic deformable
Earth model formulated in (Bueler et al 2007).

Calving of marine terminating glaciers at the ocean boundary is parameterized in PISM as the model does not have a good
representation of the narrow fjord systems of the marine outlet at the considered resolution. Several calving schemes are
implemented in PISM to cope with different conditions, including the Eigencalving, von Mises calving, thickness calving,
and flow kill calving, etc. A commonly used calving scheme for GrIS is adapted from the von Mises yield criterion which is
suited for ice flows confined in narrow valleys and fjords (Morlighem et al. 2016). The parameterization assesses the calving
speed from the amount of ice fracturing.

**4. Procedures to test possible model climate dependencies on computing platform, compilers, and domain decomposition**

EC-Earth3 runs under different high-performance computing (HPC) environments in the EC-Earth partner supercomputing
centres. This has several advantages, from allowing different groups to work with the same tool in parallel, to leveraging the
burden of ensemble climate simulations. However, for obvious scientific reasons, it is critical to ensure that ESMs provide
replicable results under changes in the computing environment. While bit-for-bit replicability is in general infeasible because
of the existence of hardware/software constraints that are beyond the control of climate researchers, it must be expected that
results obtained under one computing environment are statistically indistinguishable from those obtained under another
environment.

EC-Earth as a community has developed a protocol (Massonnet et al., 2018, 2020) to assess the replicability for the EC-
Earth3 model system. This protocol is based on a statistical comparison of standard climate metrics derived from control
integrations executed in different HPC environments. This protocol has been tested with EC-Earth 3.2, allowing judgement
of replicability of EC-Earth 3.2 results. It was also shown that the interim version of the model, EC-Earth 3.1 (developed in
between the CMIP5 and CMIP6) was not fully replicable. The experience gained (Massonnet et al., 2020) suggested that
codes (especially when they are bugged) interfere with computing environments in sometimes unpredictable ways. The
default assumption in EC-Earth is that a given model simulation is *not* replicable when the HPC environment is changed
until proven otherwise, i.e., that the model executed in the two computing environments gives results that cannot be deemed



incompatible. The protocol developed within EC-Earth fulfills this goal, and it is now required to check the replicability of
EC-Earth each time it is ported to a new machine, before new production runs can be started in this machine.

**BOX: The protocol for testing replicability**

The protocol is designed to test whether EC-Earth3 gives replicable results under two computing environments, named "A" and "B" hereinafter.

1. ***Deterministic check.*** This initial step is performed to confirm that EC-Earth is fully deterministic. Two one-year integrations are conducted consecutively under the same computing environment (same executable, same machine, same domain decomposition). The results are required to be bit-for-bit identical.

2. ***Accounting for internal variability.*** Ensemble simulations are conducted so that the imprint of internal variability on climate indices like time-means can be gauged. In an attempt to find the right balance between statistical power and limited computational resources, we run a 5-member, 20-year integration for both "A" and "B" computing environments. In the subsequent steps, these simulations are referred to as "ensembles".

3. ***Generation of simulations***. The five members of ensembles A and B always start from unique atmospheric and sea ice restarts, obtained from a long equilibrium simulation conducted on one computer. An oceanic restart is also obtained from this equilibrium simulation, and five random but deterministic perturbations are added to the sea surface temperature of this restart (gaussian perturbation, standard deviation: $10^{-4}$ K). The introduction of these tiny perturbations allows ensemble spread to develop in ensembles A and B. Note that by the deterministic nature of the perturbations, pairs of members always start from the same triplet of atmospheric, oceanic and sea ice restarts: the first member of ensemble A and the first member of ensemble B start from identical initial conditions, and so on. Ensembles A and B are conducted under an annually repeating pre-industrial constant forcing. The ensembles start in 1850 and extend to 1869.

4. ***Calculation of standard indices.*** Due to the large amount of output produced by each simulation, the outputs from ensembles A and B are first post-processed in an identical way. Based on the list of standard metrics proposed by Reichler and Kim (2008), we record for each ensemble standard ocean, atmosphere and sea ice parameters: 3-D air temperature, humidity and components of the wind; 2-D total precipitation, mean sea-level pressure, air surface temperature, wind stress and surface thermal radiation; 2-D sea surface temperature and salinity, and sea ice concentration. These fields are averaged monthly (240 time steps over 20 years) .

5. ***Calculation of standard metrics.*** Then, the model fields are compared to the same reference data sets as those used in Reichler and Kim (2008), which consist of observational and reanalysis datasets. For each field, a grid-cell area weighted average of the model departure from the corresponding reference is evaluated and then normalized by the variance of that field in the reference data set. Thus, for each field, one number is retained that describes the mismatch between that field in the ensemble, and the reference. Five such numbers are available for each ensemble, since each ensemble





uses five ensemble members.

6. ***Statistical testing.*** For each index we compare two five-member ensembles and determine whether the two ensembles are statistically indistinguishable from one another. Since no prior assumption can be made on the underlying statistical distribution of the samples, we use a two-sample Kolmogorov-Smirnov (KS) test. KS tests are non-parametric, which makes them suitable for our application. A Monte-Carlo analysis reveals that for a prescribed level of significance of 5%, the power of the two-sample KS test exceeds 80% (a standard in research) when the means from the two samples are separated by at least two standard deviations (Figure 2). Stated otherwise, there is a non negligible probability (>20%) that small but actual differences (less than 2 standard deviations) are not detected by the test. In this test, the null hypothesis is that the two samples are statistically indistinguishable from each other with a confidence of 95%. This means that under this null hypothesis of no difference, significant differences are expected to occur 5 % of the time.


Replicability is a cornerstone of science and, by extension to climate research. Testing that climate models provide replicable output under changes in HPC environments is a prerequisite to ensure trusted distribution of large ensembles across multiple platforms, which were foreseen within the EC-Earth community. The workload of CMIP6 experiments were shared and run in collaboration by different institutions of the consortium across different platforms.


Here we give an example for an application of the replicability test. It consists of the executions of the same code in coupled (CMIP) and atmosphere only configurations (AMIP), with 5 ensemble members, carried out by different institutions using their respective platforms. In essence, this test (see Box) assesses the differences due to the varying HPC environment in 20-year simulations, after accounting for internal variability.


The performance indices by Reichler and Kim (2008) (R&K), which compare model performance against the Reanalysis product ERA40, were calculated using the EC-Earth analysis tool EC-mean, available in the model source tree. The tests included six different machines (Tetralith, Rhino, Marenostrum4, CCA, Sisu (Finnish IT Center for Science), Hpcdev (CRAY XT-5) and Kay), two different compilers (Intel and Cray), different domain decomposition (number of parallel
resources used) and different versions of libraries and compilers used (subject to the availability of each platform).

The results of this test showed that the climates simulated by EC-Earth were statistically indistinguishable in most of the cases. The differences were evaluated for 13 different variables (Table 14) with similar results for all variables and vertical levels. An example for 2m temperature (t2m) is shown in Figure 2 for the HPC systems Rhino and CCA. Both the AMIP and
CMIP experiments return similar mean values when averaged spatially (Figure 2a and 2b respectively). Figure 2c also shows that the results at the end of the execution (20 years) are statistically similar in spite of the internal variability, where only





1.2% of the total results could be considered significantly different (under the null hypothesis of no difference, significant differences are expected to occur 5% of the time). Similar results were obtained for the remaining comparison of HPC systems, except for two integrations which failed this test. These failures were traced back to the specific compiler versions used. Using a different setup (with a different version of the compilers) removed the differences, thus proving that the replicability test can be used to highlight incorrect configurations.

## 5. Climate conditions as simulated in historical ensembles

To illustrate the performance of the EC-Earth3 model we present results from the ensemble of historical experiments. The forcing of the historical experiments follows the CMIP6 protocol and has been described in detail in Section 3. After tuning the coupled configurations EC-Earth3 and EC-Earth3-Veg, spinup and piControl (pre-industrial control) simulations have been carried out.

Each ensemble member of the historical experiment was branched off from the corresponding piControl experiment, at a different time, with branching times separated by intervals of 20 years (Figure 3). The complete information about the branch time of each member is included in the metadata of all variables. Separate piControl experiments have been produced for EC-Earth3 and EC-Earth3-Veg, and more will follow for other model configurations. Most of the results presented here are based on the 20-member large ensemble of the EC-Earth3 configuration that were available from the ESGF at the time of writing. The results from four available EC-Earth3-Veg members were found to be very similar and therefore we focus on the analysis of results from EC-Earth3.

### 5.1 Temperature

The timeseries of the global annual mean near surface air temperature (TAS) is shown in Figure 4. Reanalysis data from ERA5 and ERA20C are shown for comparison. Compared to ERA5 the model ensemble has a warm bias of about 0.5 K. ERA5 can be considered more relevant for the global mean TAS, in comparison with ERA20C, because ERA5 has stronger observational constraints and updated physical parameterization in the underlying global model, while ERA20C is limited to assimilation of daily surface pressure and surface winds over the ocean (Poli et al. 2016).

The bias in the EC-Earth3 global mean TAS is mainly due to a strong warm bias in the Southern Ocean as we will show below. Another feature is the large ensemble spread shown as a shaded area around the ensemble mean. The difference between the coldest and warmest ensemble member is 0.8 K on average. For further comparison, the CMIP5 ensemble of EC-Earth2.3 is also shown, which is about 1.2 K colder and indicates a smaller spread among the 10 members, in comparison to the current 20-member ensemble. When analysing the long piControl run we found that the EC-Earth3 model oscillates between two states that are characterised by low/high values of the AMOC, cold/warm North Atlantic





temperatures and more/less sea ice in the Arctic, with a period of about 200 years. The temperature difference over the North Atlantic is large enough to have a discernible impact on the global mean TAS, resulting in a warm and a cold state. The model can remain in either of these states for several decades before turning to the other state with transitions occurring at irregular intervals. We speculate here that the oscillation decreases with the warming climate. This is indicated by a larger ensemble of historical and scenario simulations than shown here. The exact processes as well as the triggers for the oscillation are still under investigation on the basis of the larger ensemble and will be presented in a later study. However, it is already clear that the large differences between the warmest and coldest member of the ensemble of historical simulations are related to the two states of the model climate, i.e. whether the initial states for the historical run stem from the cold or warm phase of the piControl run, and that transitions between different states occur during the historical simulation.

The TAS trend after 1980 is found to be stronger in the EC-Earth3 ensemble mean (0.25 K/decade) than in the ERA5 reanalysis (0.18 K/decade), indicating a too strong sensitivity of the model to the observed forcing. However, when looking at the warming during the entire historical period by comparing the mean TAS from 1851-1880 against 1981-2010 we find that the model warmed by 0.7 K which is only slightly higher than the 0.63 K estimate for the observed warming (IPCC, Hoegh-Guldberg et al. 2018)

To study the EC-Earth3 recent past climate we focus on the period 1980-2010. The ensemble mean spatial TAS is compared to ERA5 for this period in Figure 5. We find cold TAS biases over the land regions and the Arctic, and warm biases over the Southern Ocean and Antarctica as well as for the stratocumulus regions of the continents (Figure 5b). The individual member biases have a large spread in the northern hemisphere as previously mentioned, while the southern hemisphere biases are similar for all members as seen in the zonal mean bias plot (Figure 5d). The CMIP5 ensemble EC-Earth2.3 is colder than ERA5 except over the Southern Ocean (Figure 5c) and the spread is much smaller than for EC-Earth3 (Figure 5d).

**5.2 Precipitation**

Global mean precipitation (pr) patterns are well represented in EC-Earth3 in comparison with ERA5 for present day conditions (Fig 5.4), except for the double Inter-Tropical Convergence Zone (ITCZ) pattern noticeable as a distinct peak in the zonal mean pr at 9°S (Fig 5.4c). This is a persistent model bias common to most global climate models. The largest pr biases relative to the observational data set Global Precipitation Climatology Project GPCP (Gehne et al 2016) are found in the tropics, in particular in the Southern Hemisphere (SH) and to a lesser extent in the Northern Hemisphere (NH). EC-Earth3 precipitation is closer to ERA5 but has less precipitation at the equator. The overestimation of pr by the model in the SH and the double ITCZ are likely consequences of the strong warm temperature bias over the Southern Ocean that leads to a more southward displaced tropical rainfall belt and more vigorous hydrological cycle (Hwang and Frierson 2016).

### 5.3 Sea level pressure

EC-Earth3 mean sea level pressure (psl) fields are close to ERA5 as shown in Fig 5.5. The psl bias of individual members and the ensemble values are within 1.5 hPa compared to ERA5 except over Antarctica where the pressure is between 0.5 to 2 hPa too high, while over the Southern Ocean the pressure is 1 hPa too low. The variability, measured as the annual standard deviation, of EC-Earth3 psl is very close to ERA5 (Fig 5.5 c,d), indicating realistic surface winds (not shown) and modes of variability as shown later in this section.

### 5.4 Zonal Wind

An overview of the atmospheric circulation is shown in Figure 8, where the yearly-averaged biases of the zonal component of wind of EC-Earth2.3 and EC-Earth3 are assessed against ERA5. Zonal averages (Figure 8a,b) show that both models are characterised by an underestimation of the Southern Hemisphere jet, which is larger in the upper troposphere. However, EC-Earth3 shows a reduction of this negative bias compared to EC-Earth2.3.

Similar improvements are seen in the Northern Hemisphere: here EC-Earth2.3 was characterised by a negative bias on the order of -5 m s$^{-1}$ in the core of the upper tropospheric jet stream, while EC-Earth3 shows limited bias with a slight overestimation on its equatorward side at the tropopause level.

Looking at upper level zonal wind patterns (Figure 8c,d) it can be seen that considerable improvements have been obtained for the Pacific jet, strongly underestimated in EC-Earth2.3 and showing a minor southward bias in EC-Earth3. Conversely the North Atlantic jet still shows a poleward displacement - extending from Western Atlantic up to Eastern Europe - of the same magnitude in both configurations. A southward displacement of the subtropical jet over Africa is also emerging in EC-Earth3.

Overall, CMIP6 EC-Earth3 shows a reduction of the bias when compared with CMIP5 EC-Earth2.3. This is confirmed by smaller figures for both root-mean-squared-error and mean bias shown in the top of each panel in Figure 8. Larger biases in both models are observed when looking at specific seasons (not shown): EC-Earth3 shows overall smaller biases in DJF and larger biases in JJA than its CMIP5 predecessor. In general, EC-Earth3 is characterised by an underestimation of the winter jet and by an overestimation of the equatorward component of the tropical jet in the summer hemisphere. An underestimation of the winter stratospheric polar vortex - stronger in the Southern Hemisphere - is also found ).




### 5.5 Blocking

Atmospheric blocking - the recurrent long-lasting quasi-stationary high-pressure system developing at the exit of the jet streams in mid-latitudes (Woolings et al, 2018) - is assessed in Figure 9 for both EC-Earth2.3 and EC-Earth3 against ERA5 reanalysis. Analysis of blocking is relevant for climate models considering both 1) the large impact on regional weather that blocking has in both summer and winter seasons (e.g. Sillmann et al., 2011, Schaller et al, 2018) and 2) the constant struggle that GCMs experience in correctly simulating the observed blocking frequencies (e.g. Masato et al, 2013, Davini and D'Andrea, 2020). Blocking is here shown as percentage of blocked days per season following the definition of Davini and D'Andrea (2020), where a bidimensional blocking index based on the reversal of the geopotential height gradient at 500hPa is used.

Small differences arise from the comparison between EC-Earth2.3 and EC-Earth3, supported by the negligible changes in root mean squared error and mean bias reported in each panel. Both models are in line with CMIP5 and CMIP6 multi model mean (see Davini and D'Andrea 2020). The most evident improvement is the reduced bias in winter North Pacific blocking in EC-Earth3 compared to its predecessor. However, both EC-Earth2.3 and EC-Earth3 show the common underestimation of winter European blocking with similar magnitude. It is interesting to notice that while in EC-Earth2.3 the winter European bias is characterized by a north-south dipole probably associated with an equatorward displacement of the Atlantic eddy-driven jet, in EC-Earth3 the dipole is located on the east-west axis, suggesting the presence of a too penetrative jet over the European continent (Figure 9a,b )

Larger biases are seen in summer: negligible differences between EC-Earth2.3 and EC-Earth3 are also found, where slightly larger North Pacific and European blocking biases are observed in EC-Earth3.

### 5.6 Sea ice

We compare EC-Earth3 historical ensemble mean sea ice variables and spread to observations and reanalysis data sets. The ensemble mean of EC-Earth3 slightly overestimates the Arctic sea-ice area (Figure 10), whereby the ensemble spread encases the OSI SAF satellite observations (Lavergne et al., 2019). The trend in the Arctic sea ice area over 1980-2014 is captured well by the model during March and September (Figure 10).

Interestingly, the modelled summer Arctic sea ice area minimum occurs in August, instead of September as in observations. This result is in agreement with previous studies using NEMO-LIM forced by atmospheric forcing (Rousset et al., 2015; Docquier et al., 2017) and several coupled CMIP6 models including NEMO as an ocean component (Keen et al., 2020). The exact reason for the minimum sea ice area occurring in August is not clear (Keen et al., 2020).





While the total Arctic sea ice area is captured well by the model, there are large regional differences in the sea ice concentration between EC-Earth3 and satellite observations (Figure 11). In March, the model overestimates the concentration near the ice margins in the Atlantic sector, including the Labrador, Greenland-Island-Norwegian (GIN) and Barents Seas, while the concentration is underestimated by the model in the Bering Sea. In September, the sea ice concentration is generally overestimated by the model at the ice margins, with exceptions in the Kara, Laptev and Chukchi

Seas, where the concentration is underestimated.

The total Arctic sea ice volume is higher in the ensemble mean of EC-Earth3 compared to the PIOMAS reanalysis (Figure 12). The PIOMAS volume is close to the lower edge of the EC-Earth3 ensemble spread, which is considerably large, of about 30 x 10³ km³ in the 1980s and 1990s. This is consistent with the Arctic sea ice being generally thicker in the model

compared to PIOMAS (Figure 13). In September, the Arctic sea ice is clearly too thick in the model with a bias up to 2 meters compared to PIOMAS. In March, the Arctic sea ice thickness is overestimated by EC-Earth3 in the Central Arctic, while in the Bering and Karas Seas the thickness is lower compared to PIOMAS.

We chose PIOMAS as a reference product for sea ice thickness and volume because of the relatively long available time

frame (i.e. from 1979 to now), compared to observational products, which cover much shorter periods. Uncertainties in PIOMAS are related to the underlying ocean and sea ice models, to the atmospheric forcing, as well as to the observational data available to the assimilation scheme. PIOMAS ice thickness estimates agree well with the ICESat ice thickness retrievals for the Central Arctic, the area for which submarine data are available, with a mean difference smaller than 0.1 m, while differences outside this area are larger (Schweiger et al., 2011). Also, PIOMAS spatial thickness patterns agree well

with ICESat thickness. PIOMAS appears to overestimate thin ice thickness and underestimate thick ice. The latter feature partly explains the higher ice thickness in EC-Earth3 in the Central Arctic compared to PIOMAS (Figure 13).

While the EC-Earth3 historical ensemble represents the Arctic sea ice area relatively well, it clearly underestimates the Antarctic sea ice area (Figure 10) by ~5 million km$^2$ in September and ~2 million km$^2$ in March. This underestimation is

linked to the warm bias in the Southern Ocean (Figure 6), and is visible for all areas around Antarctica during the southern hemisphere summer (Figure 14). The underestimation is more pronounced close to the ice edge than in the central pack ice during the southern hemisphere winter (Figure 14). Also, the modelled trend in Antarctic sea ice area in both September and March is slightly negative, while observations show a slightly positive trend over 1979-2014 (Figure 10). However, including the most recent years (after 2014), the observed Antarctic sea ice area does not exhibit any significant trend

(Meredith et al., 2019).

Due to the absence of reliable long-term reanalysis / observational products for Antarctica, we do not show maps of sea ice thickness in the southern hemisphere.



### 5.7 AMOC

The Atlantic Meridional Overturning Circulation (AMOC) is connected with a northward flow of warm and salty water in the upper layers of the Atlantic Ocean, and exports of cold and dense water southward in the deeper layers (Buckley & Marshall, 2016). The ensemble mean of the AMOC stream function obtained from the EC-Earth3 ensemble simulations (Figure 15), after being averaged over 1980-2010, features the expected overturning clockwise circulation cell with a maximum transport of 18 Sv centered at around 35º N and a depth of 1000 m. Compared to the 12-member ensemble mean of EC-Earth 2.3 (Brodeau and Koenigk, 2016), used for CMIP5 (no figure), the CMIP6 version of EC-Earth presented here has a stronger AMOC closer to observations (Smeed et al., 2017).

The ensemble mean time series of the AMOC index, defined as the maximum volume transport stream function between 24.5-27.5º N, covers values from well within the range of the RAPID-MOCHA array observations (Smeed et al., 2017). The ensemble mean shows a weak decrease of about 0.5 Sv from year 1850 to 1876 with a relatively steady period until 1931 around 17.5 Sv, followed by an increase of around 2 Sv until 1980 and a decrease afterwards. Individual members of the ensemble vary between 2 to 5 Sv, with the upper range matching the RAPID observational variability well. It has to be noted that the RAPID data is available only for the last 20 years. Several other studies with ocean models forced by atmospheric reanalysis data (e.g., Yeager and Danabasoglu, 2014; Huang et al., 2012) show a later increase of AMOC, between 1980 and 1990's and a decrease after mid 1990's. Most CMIP5 models also show a later decrease of AMOC mainly after the year 2000 (e.g., Collins et al, 2013; Cheng et al., 2013).

There is a wide range of variability between ensemble members possibly because each member starts from a different initial condition that evolves differently depending on the state of the model's internal variability. A first analysis (no figure) suggests that the lowest AMOC strength values correspond with extended periods with absent deep water formation and expanded sea ice in the Labrador Sea. Our preliminary analysis suggests that those ensemble members with similar initial conditions (in terms of SST and SAT) have similar curves in their AMOC index time series. This further suggests that those members with a more realistic initial state, might be better in capturing the AMOC variability. Nevertheless, time series of 12 individual members (out of 20 members) show a roughly similar trend after 1950.

The AMOC (Figure 16) shows a variability of 70 to 100 years as in ECHAM5/Max-Planck-Institute Ocean Model (Jungclaus et al. 2005) but a narrower range than the variability of 50–200 years in CCSM4 (Danabasoglu et al. 2012). One reason that could explain the somewhat long period of variability for our EC-Earth simulations is that the modelled sea ice is extending too far into the Labrador Sea which keeps AMOC in a weaker state for a longer period of time before recovering. In CMIP5, most models without the convection were also covered by sea ice in the Labrador Sea in winter (Heuzé, 2017).



**5.8 Modes of variability**

Since the North Atlantic Oscillation (NAO) is one of the most relevant modes of variability for the European climate, we
assess its representation separately. The NAO is characterized by atmospheric oscillation between the Arctic and the
subtropical Atlantic and can also be defined through changes in surface pressure (Hurrell et al, 2003). During winter, the
positive phase of NAO has been defined when the difference between the Icelandic low-pressure center and the Azores high-
pressure center is intensified: conversely, a negative NAO phase is when this difference is weaker than usual. The NAO
pattern is estimated calculating the leading Empirical Orthogonal Function (EOF) of the detrended December to February
(DJF) monthly Sea Level Pressure (SLP) anomalies over the Atlantic sector (20-80N, 90W-40E). The ERA20C and both
EC-Earth ensemble means show very similar spatial patterns (Figure 19), with EC-Earth2.3 showing more intense negative
values over Iceland and less intense positive values over the Biscay Bay and Northern Spain when compared to EC-Earth3
and observations.

Overall the representation of the NAO in EC-Earth3 shows minor improvements with respect to EC-Earth2.3: The RMSE of
EC-Earth3 is 0.64 compared with 0.52 for EC-Earth2.3.
Previous assessments of coupled models have found that the CMIP5 and CMIP3 models represent NAO very similarly to
reanalysis, with no general improvements of CMIP5 compared to CMIP3 models (Davini and Cagnazzo 2014).

In EC-Earth3, the ENSO spatial patterns and amplitude are well represented (not shown). In order to assess the variability of
ENSO in the EC-Earth3 ensemble we calculated the Niño 3.4 SST power spectra (Figure 17) for the model and for the
HadISST observational data set (Rayner et al. 2003). HadISST shows the highest peak at 5.5 years, and a range of periods
between 3.3 and 3.7 forming the second highest peak. The mean of the EC-Earth3 ensemble spectra shows the highest peak
at around 3.7 years. Some individual members show their main peak at about 5.5 years, similar to the HadISST data set.
Most of the ensemble members (two with the highest peaks) show their maximum in the 3-4 years period.

EC-Earth3 shows larger ENSO spectral power than EC-Earth2.3 with an ensemble mean closer to the HadISST observations,
especially in frequencies with high energy. Thus we see a clear improvement in the representation of ENSO in EC-Earth3.
Similar improvements have been seen earlier for CMIP5 models (Jha et al. 2014). While the frequency distribution in EC-
Earth3 is distinctly improved compared to EC-Earth2.3, a challenge is still seen in representing distinct peaks in the power
spectrum.

Previous studies have shown that winters of El Niño have an impact of the circulation over the North Atlantic, favoring a
negative phase of the NAO in the late winter (e.g. Garcia-Serrano et al. 2011) teleconnected via the extratropical Pacific-





North American (PNA) pattern (Enfield and Mayer 1997; Giannini et   al. 2000). This nonstationary relationship between
ENSO and the atmospheric variability over the North Atlantic and European sector is reproduced in the internal variability
behavior of coupled ocean–atmosphere systems, already in CMIP5 simulations (Lopez-Parages et al., 2016)

To better assess this relationship between ENSO and NAO, we calculated the regression of Nino3.4 index on Sea Level
Pressure (SLP) for the winter season (December to February, DJF) for ERA-20C reanalysis and for both EC-Earth
ensembles (EC-Earth3 and EC_Earth2.3) (Figure 18). As reported by previous studies (Wallace and Gutzler 1981), a
positive PNA-like pattern arises with an intensified lower Aleutian low, and lower pressure over the eastern United States.
This low pressure extends along the North American east coast into the central North Atlantic Ocean while there is higher
pressure over and between Iceland and Scandinavia, a pattern that is similar to a negative phase of NAO (Figure 18). The
EC-Earth3 ensemble demonstrates a high resemblance to ERA-20C (Figure 18), much improved compared to the EC-
Earth2.3 ensemble (Figure 18). The latter shows weaker SLP anomalies which can be associated with a weaker ENSO
intensity in EC-Earth2.3 compared to the newer EC-Earth3 and to observations. Thus, EC-Earth3 gives a clear improvement
of the ENSO-NAO link.

The PNA pattern in EC-Earth3 gives strong spatial similarities with ERA20C, which is an improvement compared to EC-
Earth2.3 (not shown). RMSEs are respectively 0.9 and 0.54.

EC-Earth3 also shows a more realistic behaviour than EC-Earth2.3 with respect to typical features of the PNA patterns, i.e.
an above-average surface pressure over the subtropical Pacific (west of Hawaii) and over western Canada, together with
below-average surface pressure over the North Pacific Ocean and along the south-eastern United States.

The quasi-biennial oscillation (QBO) dominates the interannual variability in the tropical stratosphere (Baldwin et al., 2001).
As in most climate models, the vertical discretization and the parameterization of gravity waves are crucial (e.g., Bushell et
al., 2020) for a realistic representation of the QBO in EC-Earth.


In EC-Earth 2.3, with 62 vertical levels and without an ad-hoc tuning, the zonal wind in the equatorial band is easterly on
average, reaching -35 m/s close to the 5 hPa level (Figure 20a). Note that results in this plot are based on model level
outputs, interpolated onto equivalent pressure levels. CMIP5 pressure level outputs are available only up to 20 hPa for other
ensemble members.


Like other models participating in CMIP6 (Richter et al., 2020), the realism of the modelled QBO is notably improved in
EC-Earth3, with 91 vertical levels and a revised gravity wave scheme (see Section 3.1). A marked alternation of easterly and
westerly zonal wind shears can be seen, approximately between 100 and 5 hPa, in Figure 20b.While a realistic QBO-like





oscillation has been obtained, there is no improvement in the zonal wind above 5 hPa, as it remains easterly year-round
without a clear sign of any stratospheric semi-annual oscillation. This phenomenon, linked with the QBO, is also driven by
vertically propagating waves (Smith et al., 2019).

From the comparison of the amplitude profiles with ERA5 (Figure 20c), it is clear that, even if still underestimated, the zonal
wind variability in the EC-Earth3 ensemble members is improved with respect to those of EC-Earth2.3. As with other
climate models (Bushell et al., 2020), the maximum of the oscillation peaks somewhat higher (at 10 hPa) in EC-Earth3 than
in the reanalysis (closer to 20 hPa).

In Figure 20d we also report the Fourier spectra at the 30 hPa (25 km) level, again including ERA5 and ensemble members
from EC-Earth2.3 and 3. For the CMIP5 model only the annual harmonic stands out, whereas the EC-Earth 3 spectrum at 30
hPa resembles that of ERA5 much more closely, despite the modelled QBO being slightly faster.

### 5.9 Carbon cycle components

As an outlook to the EC-Earth3-CC configuration, we present here first results from the carbon cycle components of one
member of the historical experiment. In this configuration, the physical climate is equivalent to that of the EC-Earth3-Veg
configuration. However, besides the land vegetation and biogeochemistry model, the ocean biogeochemistry model has been
activated. The latter describes the evolution of biogeochemical tracers in the ocean but it does not have any feedback on
climate. Both land vegetation and ocean biogeochemistry were equilibrated with a 1600-year long spin-up keeping
atmospheric $CO_2$ at pre-industrial level (284.32 ppm). The completion of the spin up was assessed following the
recommendation of C4MIP (Jones et al., 2016) where both the ocean and land C stocks had to drift by less than 10 Pg
C/century. Once this condition was met, a piControl and historical simulation were started from the same initial conditions.


The ocean's cumulative C uptake between 1870-2014 is 150.78 Pg C, in good agreement with the estimate of 155±20 Pg C
by the Global Carbon Project (GCP) 2015 for the same period (Le Queré et al., 2015). The average air-sea $CO_2$ exchange
between the atmosphere and the ocean for the period 1981-2014 is here compared to the observation-based reconstruction of
Landschutzer et al. (2016) for the same period (Figure 21). The overall spatial distribution of air-sea $CO_2$ flux is in broad
agreement with observations with the largest differences occurring at high latitudes. In particular, in the Southern Ocean EC-
Earth3-CC exhibits a large negative bias (i.e. more outgassing) in some regions due to the presence of extended periods of
unrealistic open-sea convection. In the North Pacific and North Atlantic, EC-Earth3-CC has a stronger $CO_2$ uptake than
observations likely due to deeper mixed layer and frequent active convection in the Labrador Sea, respectively.

In Figure 22 we show $CO_2$ fluxes from the land and ocean components of EC-Earth3 for the period 1960-2014. The ocean
$CO_2$ sink (SOCEAN; Figure 22c) of the EC-Earth3-CC historical simulation closely follows the multi-model mean of the

GCP 2019 (Friedlingstein et al. 2019) with differences in variability due to the mismatch between climate in EC-Earth-CC and the atmospheric forcing products used in the GCP. Land fluxes (computed from two historical EC-Earth3-Veg runs, one with dynamic land use change and management, and one with land-use fixed at 1850 levels) have increased following increased atmospheric CO2 and temperature but these show a stronger interannual variability both in emissions from land use change (ELUC; Figure 22a) and terrestrial CO2 sink (SLAND; Figure 22b) compared to the estimates from the DGVMs in GCP 2019. This interannual variability originates both from climate differences between EC-Earth3-Veg and the observation-based forcing, which includes the right timing of weather events like El Niño in the offline simulations used for the GCP (Harris et al., 2014) and the mismatch between the dynamic vegetation in LPJ-GUESS compared to the land-use dataset LUH2 (Hurtt et al. 2020) which dictates where and when land-use change occurs.

## 6. Summary and Conclusions

The EC-Earth research consortium represents a community of European institutes developing and utilizing the EC-Earth Earth System Model. In this paper we document the overall concept of EC-Earth3, the model version used for contributions to CMIP6, and its flexible coupling framework, major model configurations, a methodology for ensuring the simulations are comparable across different HPC systems, and the physical performance of base configurations over the historical period. Simulations described in this paper have been carried out under the CMIP6 framework conditions (Eyring et al. 2016). Wherever possible, we also compare the CMIP6 results to the previous model version EC-Earth2 under the conditions of CMIP5.

The different configurations of EC-Earth3 described in section 2 are enabled by a flexible coupling framework. A traditional GCM configuration, comprising a coupled atmosphere and ocean model, in different spatial resolutions is accompanied by configurations with an interactive vegetation module, active atmospheric composition and aerosols, full carbon cycle configuration, and a configuration with a Greenland ice sheet model. The variety of possible configurations and sub-models reflects the broad interests in the EC-Earth community. The releases of the different configurations were staggered in time to efficiently gain from preceding coupling and tuning efforts.

The different component models linked for the EC-Earth3 framework, described in section 3, are partly community models or arise from developments of partner institutes, or from centralized efforts such as for the forecast model IFS, developed by ECMWF.



Model tuning has been carried out for different configurations. It was feasible to share identical tuning for both the GCM (base configuration EC-Earth3) and the configuration coupled to dynamic vegetation for a given resolution. Different tuning outcomes had to be applied to varying resolutions and for the configuration with interactive aerosols and atmospheric composition.

For standard resolutions, the goal of atmosphere tuning was to minimize the error in the global means of the net radiative flux at the surface, the top of the atmosphere (TOA) fluxes, and longwave and shortwave cloud forcing, without compromising the radiative imbalance at the top of the atmosphere. In the configurations with the coupled ocean, it was possible to adjust only ocean and sea-ice parameters while maintaining atmospheric tuning to prevent the model from drifting under constant forcing.

For low resolution configurations, largely used for millennium scale simulations, we aimed at a tuning for a climate in a radiative equilibrium, to prevent the global mean surface temperature from drifting under the conditions of a stable climate. Running the ESM on different HPC systems across various partner institutions constitutes a challenge for comparability of simulations. Therefore, the EC-Earth consortium has chosen to implement a protocol that judges compatibility based on statistical differences between ensembles carried out in different computing environments. The protocol is applied in all cases when simulations are shared between partners with their respective HPC systems.

The basic physical performance of EC-Earth3 is presented by a number of key indicators and quantities, with a focus on CMIP6 historical simulations carried out for 1850-2014. EC-Earth3 represents a large step forward compared to previous versions. As the basic configuration we chose EC-Earth3, the classic GCM configuration, because of the larger ensemble of simulations compared to the EC-Earth3-Veg configuration, which is the GCM coupled to the dynamic vegetation model. As performance metrics we chose global means of key variables, their geographical patterns, behavior of oscillation patterns and circulation features.

We find that the global mean temperature in the historical ensemble has a warm bias of about 0.5 K in comparison with ERA5, which is mainly due to a strong warm bias in the Southern Ocean area. We find an oscillatory behaviour between two states that are characterised by low/high values of the AMOC, cold/warm North Atlantic temperatures and more/less sea ice in the Arctic. There is indication that the oscillation might be reduced as the climate warms. The global warming over the historical period, given as near-surface air temperature difference between the periods 1981-2010 and 1851-1880, is 0.7 K, which is only slightly higher than the 0.63 K estimate for the observed warming (IPCC, Hoegh-Guldberg et al. 2018)

The global mean precipitation patterns are well represented in EC-Earth3, while the amplitude is overestimated, pointing to an overestimation of the hydrological cycle. Mean sea level pressures are close to ERA5 for most geographical areas, as is the interannual variability.

The ensemble mean Arctic sea-ice area is slightly overestimated in several regions, while the trend since 1980 is captured 1335    well by the model during all months. Furthermore, the total Arctic sea ice volume is overestimated compared to reanalysis data. EC-Earth3 clearly underestimates the Antarctic sea ice area as a consequence of a warm bias in the Southern Ocean.

Zonal winds in EC-Earth3 are characterised by an underestimation of the winter westerly jet and by an overestimation of the equatorward component of the tropical jet in the summer hemisphere. An underestimation of the winter stratospheric polar 1340    vortex - stronger in the Southern Hemisphere - is identified.

Atmospheric blocking shows a typical bias over the winter North Pacific and the common underestimation of winter European blocking, with indications of a too penetrative jet over the European continent.
The ensemble AMOC index covers values from well within the range of existing observations. Variability in the individual 1345    members of the ensemble varies between 2 to 5 Sv, which is in line with observed decadal scale variability.

The NAO and PNA patterns in the EC-Earth3 ensemble shows a spatially high resemblance to the ERA20C reanalysis. A small improvement compared to previous versions is in line with generally minor improvements between model generations since CMIP3. A clear improvement of the ENSO-NAO link is seen.
The ENSO power spectrum for EC-Earth3 shows peak amplitudes close to the observations, which is a pronounced improvement compared to older versions. Also the frequency distribution in EC-Earth3 is markedly improved, though n representing distinct peaks in the power spectrum is still a challenge.

The realism of the modelled QBO is notably improved in EC-Earth3, thanks to increased vertical resolution and a revised non-orographic gravity wave scheme.

Parallel papers extend the current analysis to: improve our understanding of the EC-Earth3 climate sensitivity (Wyser et al. 2020a), the impact of new emission scenarios (Wyser et al. 2020b), results from the high resolution configuration (Haarsma 1360    et al 2020) and a platform comparison study (Massonnet et al., 2020). Forthcoming papers will, for example, explore dynamic oscillations during the PI control, assess the skill of climate forecasts (Bilbao et al., 2020) and highlight future climate projections.



In summary, the EC-Earth3 key performance metrics demonstrate physical behaviour and biases well within the frame
known from CMIP5 models. With improved physical and dynamic features (sections 2 and 3), new ESM components, a
much more flexible system framework, community tools, and largely improved indicators compared to the CMIP5 version.
In short, EC-Earth3 represents a large step forward for the only European community ESM. We show here that EC-Earth3 is
suited for a range of tasks in CMIP6 and beyond.

**Code availability.**

The EC-Earth3 code is available from the EC-Earth development portal for members of the consortium. All code related to
CMIP6 forcing is implemented in the component models. Model codes developed at ECMWF, including the atmosphere
model IFS, are intellectual property of ECMWF and its member states. Permission to access the EC-Earth3 source code can
be requested from the EC-Earth community via the EC-Earth website (http://www.ec-earth.org/) and may be granted if a
corresponding software license agreement is signed with ECMWF. The repository tag for the version of EC-Earth that is
used in this work is 3.3.1. Currently, only European users can be granted access, due to licence limitations of the atmosphere
model. The component models NEMO, LPJ-GUESS, TM5 and PISM are not limited by their licenses.

**Data availability**

Data produced with EC-Earth3 in the framework of CMIP6 experiments are freely available from ESGF (Earth System Grid
Federation) nodes. Currently EC-Earth simulations from 36 experiments are available for download.

**Author contribution**

All authors contributed to the development of the code or to the analysis of results. All authors contributed to the discussion
of the results and the final manuscript.

**Acknowledgements**

This paper and the development of EC-Earth3 would not have been possible without the member institutions of the EC-Earth
consortium and their sustained support of the development and application to CMIP6. Those members are the Agencia
Estatal de Meteorología (AEMET, Spain), Institute of Atmospheric Sciences and Climate of the Consiglio Nazionale delle
Ricerche ISAC-CNR (Italy), Danmarks Meteorologiske Institut DMI (Denmark), Finnish Meteorological Institute FMI
(Finland), the Portuguese Institute for Sea and Atmosphere IPMA (Portugal), the Royal Netherlands Meteorological Institute
KNMI (The Netherlands), Department of Housing, Planning and Local Government Met Éireann (Ireland), the Swedish



Meteorological and Hydrological Institute SMHI (Sweden), the Alfred Wegener Institute AWI (Germany), Barcelona Supercomputing Center BSC (Spain), the Centro de Geofisica, University of Lisbon (Portugal), the National Agency for New Technologies, Energy and Sustainable Economic Development ENEA (Italy), Geomar (Germany), the Geophysical Institute, University of Bergen (Norway), the Irish Centre of High-End Computing ICHEC (Ireland), the Institute for Marine and Atmospheric Research Utrecht IMAU (The Netherlands), Karlsruhe Institute of Technology KIT (Germany), Lund University (Sweden), Meteorologiska Institutionen at Stockholm University MISU (Sweden), Niels Bohr Institute at University of Copenhagen (Denmark), Netherlands eScience Center NLeSC (The Netherlands), Oulun Yliopisto (Finland), SARA (The Netherlands), Université catholique de Louvain (Belgium), Universiteit Utrecht (The Netherlands), Universiteit Wageningen (The Netherlands), University College Dublin (Ireland), University of Helsinki (Finland), Uppsala Universitet (Sweden), University of Santiago de Compostela USC (Spain), Vrije Universiteit Amsterdam (The Netherlands).

The authors would like to acknowledge the use of component models as provided by either central organizations (ECMWF) or communities (for TM5, LPJ-GUESS, NEMO including LIM3 and PISCES, PISM).

The authors like to thank Anna Eronn from SMHI for supporting the preparation of the manuscript.

The computations for this publication were partly enabled by resources provided by the Swedish National Infrastructure for Computing (SNIC) at NSC and PDC, partially funded by the Swedish Research Council through grant agreement no. SNIC 2018/2-11. Computations needed for model tuning were enabled by computing and archive resources provided by ECMWF under special project SPNLTUNE. Further computation resources used for production with the EC-Earth-CC configuration were partly enabled by the Partnership for Advanced Computing in Europe (PRACE) under the allocation TOPSyCled (No.: 2019204993).

The development of EC-Earth3 was supported by the European Union's Horizon 2020 research and innovation programme under projects IS-ENES3, the third phase of the distributed e-infrastructure of the European Network for Earth System Modelling (ENES) (Grant agreement n. 824084, PRIMAVERA (GA n. 641727) and CRESCENDO (GA n. 641816).

E. T. and R.B. have received funding from the European Union's Horizon 2020 research and innovation programme under the Marie Skłodowska-Curie grant agreements No. 748750 (SPFireSD project) and No. 708063 (NeTNPPAO project).IC was supported by Generalitat de Catalunya (Secretaria d'Universitats i Recerca del Departament d'Empresa i Coneixement) through the Beatriu de Pinós programme. YRR was founded by the European Union's Horizon 2020 Research and Innovation Programme in the framework of the Marie Skłodowska-Curie grant INADEC (Grant Agreement 800154). P.A.M., L.N., D.W., R.S. and B.S. acknowledge financial support from the strategic research area "Modeling the Regional and Global Earth System" (MERGE), and the Lund University Centre for Studies of Carbon Cycle and Climate Interactions



(LUCCI)..P.A.M, D.W. and B.S. acknowledges financial support from the Swedish national strategic e-science research program eSSENCE. P.A.M. further acknowledges financial support from the Swedish Research Council (Vetenskapsrådet), project Dnr. 621-2013-5487. SY acknowledges financial support from a Synergy Grant from the European Research Council under the European Community's Seventh Framework Programme (FP7/2007-2013)/ERC (grant agreement 610055) as part of the ice2ice project, and the NordForsk-funded Nordic Centre of Excellence project (award 76654) ARCPATH. MSM

acknowledges financial support from the Danish National Center for Climate Research (NCKF). A.A. and P.A. acknowledges funding from the Helmholtz Association in its ATMO programme.

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





**Tables**

| Configuration | Atmosphere + land surface | Ocean + sea ice | Dynamic vegetation | Atmospheric composition | Greenland ice sheet |
|---|---|---|---|---|---|
| | IFS 36r4 + HTESSEL | NEMO3.6 + LIM3 | LPJ-GUESS | TM5 | PISM |
| EC-Earth3 | x | x | | | |
| EC-Earth3-Veg | x | x | x | | |
| EC-Earth3-AerChem | x | x | | x | |
| EC-Earth3-CC | x | x | x | x (CO2 only) | |
| EC-Earth3-GrIS | x | x | x | | x |


*Table 1: Configurations of the EC-Earth model for CMIP6, the name of the configuration is used as source_id in the CMIP6 context.*

| | Resolution atmosphere | Resolution ocean | Timestep |
|---|---|---|---|
| Standard resolution | T255L91 (~80 km) | ORCA1L75 (1 deg) | 2700 s |
| Low resolution EC-Earth3-LR and EC-Earth3-Veg-LR | T159L62 (~125 km) | ORCA1L75 (1 deg) | 3600 s |
| High resolution (EC-Earth3P-HR and EC-Earth3-HR)) | T511L91 (~40 km) | ORCA025L75 (0.25deg) | 900 s |


*Table 2: Commonly used resolutions for CMIP6. The suffixes LR and HR are added to the name of the model configuration where applicable (e.g. EC-Earth3-Veg-LR)*





| Atmosphere -> Ocean | Ocean -> Atmosphere |
|---|---|
| Momentum flux | Sea surface temperature |
| Heat flux solar + non solar | Sea ice concentration |
| Evaporation | Sea ice temperature |
| Precipitation liquid + solid | Sea ice albedo |
| Sensitivity of non solar heat flux over ice | Sea ice thickness (not used) |
| | Snow thickness on sea ice |

*Table 3: Variables and fluxes exchanged at the ocean-atmosphere interfaces.*

| Atmosphere -> Runoff mapper | Runoff mapper -> Ocean |
|---|---|
| Runoff | Runoff |
| Excess snow | Calving |

*Table 4: Variables and fluxes provided to the ocean via the runoff mapper*


| Atmosphere -> Vegetation | Vegetation -> Atmosphere |
|---|---|
| 2m temperature | Dominant vegetation type low + high |
| Precipitation | Leaf area index low + high |
| Soil temperature (4 layers) | Vegetation cover low + high |
| Shortwave radiation | |
| Longwave radiation | |

*Table 5: Variables exchanged between the atmosphere and the vegetation model*




| IFS parameter name | Description | EC-Earth3 | IFS CY36R4 |
|---|---|---|---|
| RPRCON | Rate of conversion of cloud water to rain | 1.34E-3 | 1.4E-3 |
| ENTRORG | Entrainment in deep convection | 1.7E-4 | 1.8E-4 |
| RVICE | Fall speed of ice particles | 0.137 | 0.15 |
| RLCRITSNOW | Critical autoconversion threshold for snow in large-scale precipitation. | 4.2e-5 | 5.0E-5 |
| RSNOWLIN2 | Constant governing the temperature dependence of the autoconversion of ice crystals to snow in large-scale precipitation. | 0.035 | 0.025 |
| ENTRDD | Average entrainment rate for downdrafts | 3.0E-4 | 2.0E-4 |
| RMFDEPS | Fractional mass flux for downdrafts | 0.3 | 0.35 |
| RCLDIFF | Mixing coefficient for turbulence, controls cloud cover | 3.6E-6 | 3.0E-6 |
| RLCRIT_UPHYS | Critical droplet radius for the autoconversion in large-scale precipitation | 0.875E-5 | 0.93E-5 |

*Table 6: Atmospheric tuning parameters changed in EC-Earth compared to IFS CY36R4. The table reports the new values adopted for T255L91 EC-Earth3 and EC-Earth3-Veg tuning.*






| Atmosphere -> CTM | CTM -> Atmosphere |
| --- | --- |
| Logarithm of surface pressure | Ozone mixing ratio |
| Vorticity (3D) | Methane mixing ratio |
| Divergence (3D) | Aerosol number and component mass mixing ratios (25 fields in total) |
| Surface orography | Aerosol extinction (14 wavelengths) |
| Surface pressure | Aerosol single scatter albedo (14 wavelengths) |
| Air temperature (3D) | Aerosol asymmetry factor (14 wavelengths) |
| Specific humidity (3D) | |
| Cloud liquid/ice water content (3D) | |
| Cloud area fraction (3D) | |
| Overhead/underfoot cloud area fraction (3D) | |
| Updraft/downdraft convective air mass flux (3D) | |
| Updraft/downdraft convective air mass detrainment rate (3D) | |
| Land sea mask | |
| Surface albedo | |
| Surface roughness length | |
| Sea ice fraction | |
| Sea surface temperature | |
| 10m wind speed | |
| Skin reservoir water content | |
| 2m temperature | |
| 2m dew point temperature | |



| | |
|---|---|
| Surface latent heat flux | |
| Surface sensible heat flux | |
| Eastward/northward surface stress | |
| Large scale precipitation | |
| Convective precipitation | |
| Surface shortwave radiation | |
| Snow depth | |
| Soil wetness in top soil layer | |
| Vegetation type fraction (15 categories) | |
| High/low vegetation cover | |

*Table 7: Variables exchanged with a 6 h frequency between the atmosphere and the chemical transport model (CTM) TM5 in EC-Earth3-AerChem.*






| Vegetation -> CTM (CC only) | CTM -> Vegetation (CC only) | Atmosphere -> CTM (every 6h) | CTM -> Atmosphere |
|---|---|---|---|
| Net primary production | CO2 mixing ratio | Logarithm of surface pressure | CO2 mixing ratio |
| Heterotrophic respiration | | Vorticity (3D) | |
| Establishment* | | Divergence (3D) | |
| Reproduction* | | Surface orography | |
| Burnt vegetation and litter* | | Surface pressure | |
| Sowing* | | Air temperature (3D) | |
| Fast and slow harvested products* | | Specific humidity (3D) | |
| Landcover change* | | Updraft/downdraft convective air mass flux (3D) | |
| Carbon leaching | | Updraft/downdraft convective air mass detrainment rate (3D) | |
| | | Land sea mask | |
| | | Surface roughness length | |
| | | 10m wind speed | |
| | | Surface latent heat flux | |
| | | Surface sensible heat flux | |
| | | Eastward/northward surface stress | |

*Table 8: Variables exchanged with a 24 h frequency between the vegetation model LPJ-GUESS and the Chemical transport model TM5 in EC-Earth3-CC. *Fluxes that occur once a year and are distributed evenly over the following year.*






| Ocean BGC -> CTM (CC only) | CTM -> Ocean BGC (CC only) |
|---|---|
| $CO_2$ flux | $CO_2$ mixing ratio |

*Table 9: Variables exchanged with a 24 h frequency between the chemical transport model TM5 and the ocean biogeochemistry model PISCES.*


| Atmosphere -> Ice sheet | Ice sheet -> Atmosphere |
|---|---|
| Subsurface temperature | Ice topography |
| Surface Mass Balance (P-E-R) | Ice extent |

| Ocean -> Ice sheet | Ice sheet -> Ocean |
|---|---|
| | Calving fluxes (mass and energy) |
| | Basal melt flux (mass) |

*Table 10: Variables exchanged between the atmosphere model IFS and the ice sheet model PISM, as well as between the ocean model NEMO and ice sheet model PISM.*






| Architecture | Configuration | Platform | SYPD | ASYPD | CHSY | Parallelization |
|---|---|---|---|---|---|---|
| LENOVO SD530 | EC-Earth3 | MN4 | 15.2 | 9.87 | 1119 | 768 |
| BullX B500 | EC-Earth3 | RN | 16.2 | 16.2 | 1276 | 864 |
| CRAY XC40 | EC-Earth3 | CCA | 6.03 | 4.84 | 1289 | 324 |
| CRAY XC40 | EC-Earth-Veg | BK | 12.4 | 6.65 | 1676 | 864 |
| CRAY XC40 | EC-Earth-Veg | CCA | 6.67 | 5.32 | 1332 | 342 |

*Table 11. Basic CPMIP metrics of EC-Earth3 and EC-Earth3-Veg Standard resolution for four different architectures and*
*platforms: Marenostrum4 (MN4, LENOVO SD530), Rhino (RN, BullX B500), ECMWF-CCA (CCA, CRAY XC40) and Beskow (BK, CRAY XC40). The basic metrics are SYPD (Simulated Years Per Day), ASYPD (Actual Simulated Years Per Day), CHSY (Core Hours per Simulated Year) and Parallelization (number of MPI processes used).*


| Configuration | Resolution [# gridpoints] | Cmpx | SYPD | ASYPD | CHSY | Paral. | JPSY (Joules) | Coup. C. (%) | Mem. B. | DO (%) | DI (GB) |
|---|---|---|---|---|---|---|---|---|---|---|---|
| EC-Earth3 | $1.60 \times 10^7$ | 0.31 | 15.2 | 9.87 | 1119 | 768 | $4.41 \times 10^7$ | 8 | 11 | 12 | 3 |

*Table 12. CPMIP metrics analysis for EC-Earth3 on Marenostrum4. Complete CPMIP metrics are shown in this table. Resolution Complexity (Cmpx), SYPD (Simulated Years Per Day), ASYPD (Actual Simulated Years Per Day), CHSY (Core Hours per Simulated Year), Parallelization, JPSY (Joules per Year Simulated), Coup. C. (Coupling Cost), Mem. B. (Memory Bloat), DO (Data Output Cost) and DI (Data Intensity). From left to right we have resolution (Resol) as the total number of*
*gridpoints for all the components used (ocean, atmosphere…). Cmpx includes all prognostic variables of a model. JPSY quantifies the energy cost of the execution. Coup.C. represents the cost associated with the coupling among components (including interpolation and communication calculations, 8% in this case with respect to the total execution time). Mem.B. is the division between the theoretical memory of a memory and the real one. DO is the cost of the output process (12% in this case with respect to the total execution time). DI is the output volume in GB per day of simulation.*




| Forcing dataset | Version | Further info | Comments |
|---|---|---|---|
| Greenhouse gas concentration | 1.2.0 | Meinshausen et al. (2017) | |
| Stratospheric aerosols | 3.0.0 | Thomason et al. (2018) | |
| Ozone volume mixing ratio | 1.0 | http://blogs.reading.ac.uk/ccmi/forcing-databases-in-support-of-cmip6/; Hegglin et al. (in prep) | |
| Solar | 3.2 | Matthes et al. 2017 | |
| Aerosol Optical Properties and Relative Change in Cloud Droplet Number Concentration | MACv2-SP | Stevens et al. (2017) | |
| Land use | v2.1h | https://cmip.ucar.edu/lumip | Used only in combination with dynamic vegetation model |
| Nitrogen deposition | v2.0 | http://blogs.reading.ac.uk/ccmi/forcing-databases-in-support-of-cmip6/ ; Hegglin et al., (in prep) | Used only in combination with dynamic vegetation model |


Table 13: CMIP6 forcing datasets used by EC-Earth3 and EC-Earth3-Veg for DECK( Diagnostic, Evaluation and Characterization of Klima) and historical experiments. All datasets are available from https://esgf-node.llnl.gov/search/input4mips/. A more detailed description of the CMIP6 forcing datasets is available at http://goo.gl/r8up31.







| Variable | Description | Spatial Mean differences (%) |
|----------|-------------|------------------------------|
| t2m | 2m air temperature | 1.2 |
| msl | Mean sea level pressure | 0.7 |
| qnet | Net thermal radiation | 0.8 |
| tp | Total precipitation | 1.4 |
| ewss | Zonal wind stresses | 1.1 |
| nsss | meridional wind stress | 1.3 |
| SST | sea surface temperature | 0.7 |
| SSS | sea surface salinity | 0.8 |
| SICE | sea ice concentration | 1.1 |
| T | 3-D air temperature | 1.3 |
| U | 3-D zonal wind | 1.5 |
| V | 3-D meridional wind | 1.2 |
| Q | specific humidity | 0.7 |

Table 14.: Difference in 20-year mean and ensemble mean near-surface for different variables between Rhino and CCA

platform experiments.






**Figures**

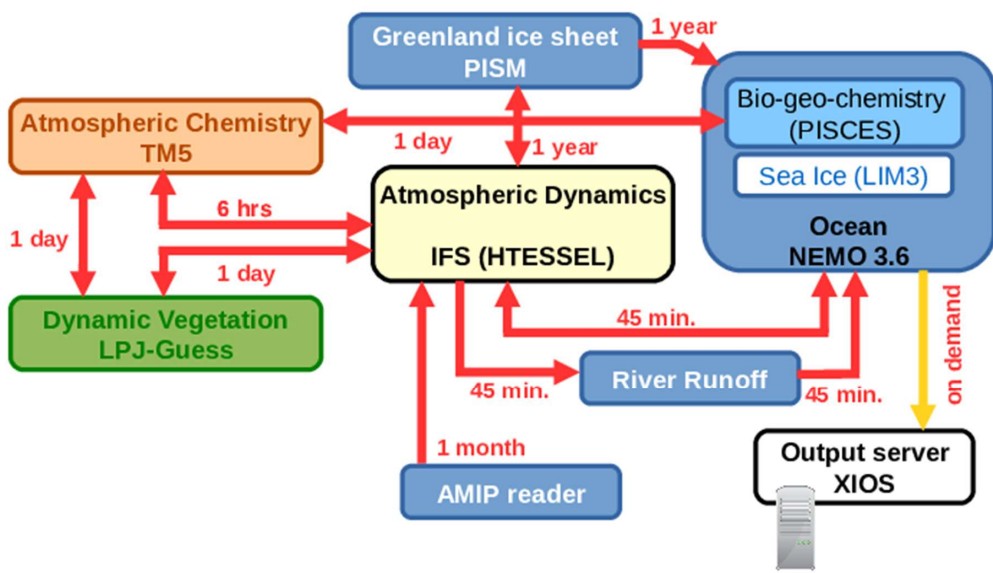

Figure 1: Coupling links and typical frequencies at standard resolution between all components that potentially can be coupled. Existing configurations include sub-sets of component models and associated couplings.

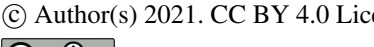



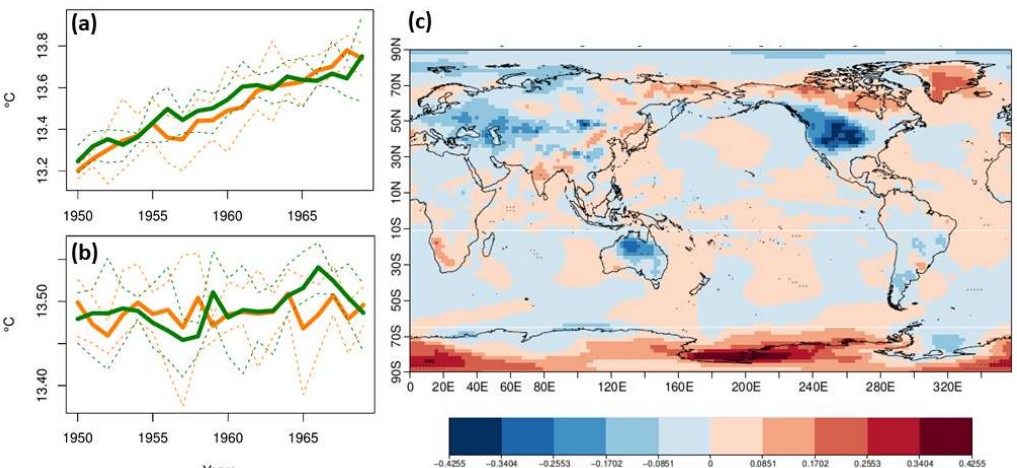

Figure 2: Comparison between Rhino and CCA platforms for AMIP experiments and historical CMIP experiments for 2m-

temperature (t2m). Panels (a) and (b) show mean (solid line) and maximum/minimum values (dashed lines) between the

ensemble members  for AMIP (a) and CMIP (b) experiments respectively. Green and orange colors refer to  Rhino and CCA

platforms respectively. Panel (c) shows the t2m difference between the CMIP ensemble runs. Black dotted regions indicate

statistical significance according to the Kolmogorov-Smirnov test (1.2% of grid points show a significant difference in this

case).


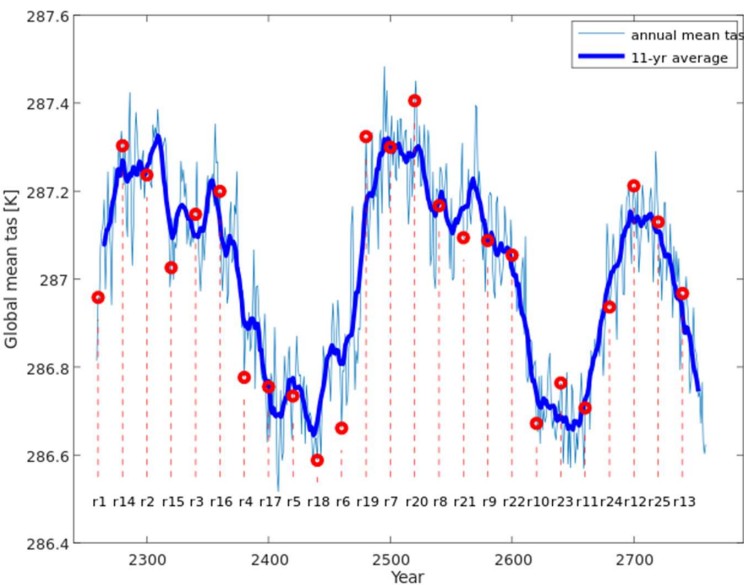

Figure 3: Timeseries of global mean of annual mean near surface temperature (tas) in the 500-yr long EC-Earth3 piControl experiment. The thick blue line is a 11-yr running average of the annual means. The time axis is arbitrary because of constant forcing. Red circles mark the initial states from where the members of the historical experiment are started. The realization_id's of the members of the historical ensemble are shown at the bottom of the figure.






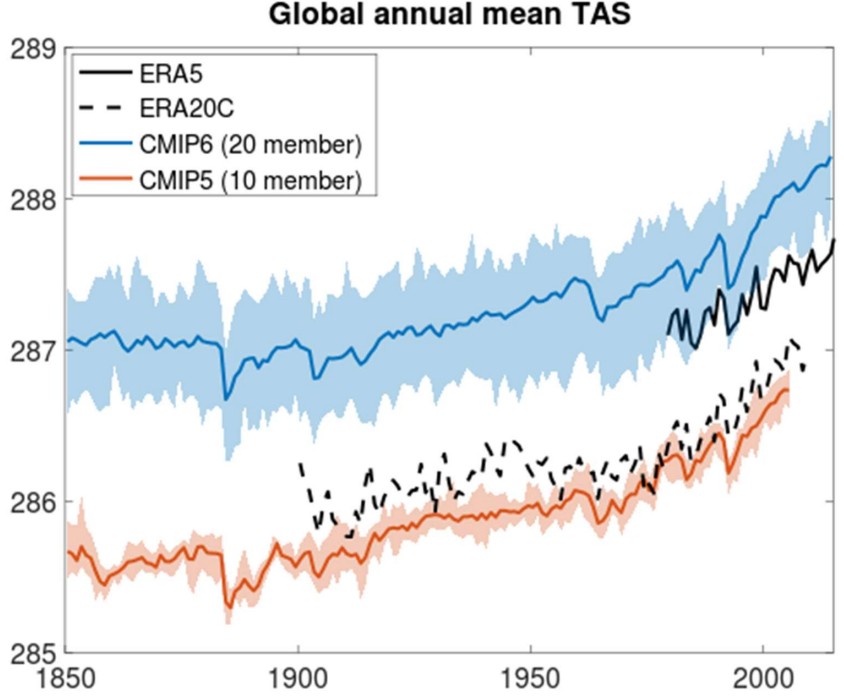

Figure 4: Global annual mean TAS in K from the EC-Earth2.3 (red, for CMIP5) and EC-Earth3 (blue, for CMIP6)
ensembles. Ensemble means are shown as thick lines and the ensemble spread is shown as a shaded area. Global annual
mean TAS from the ERA5 (black, solid) and ERA20C (black, dashed) re-analyses are shown for comparison.



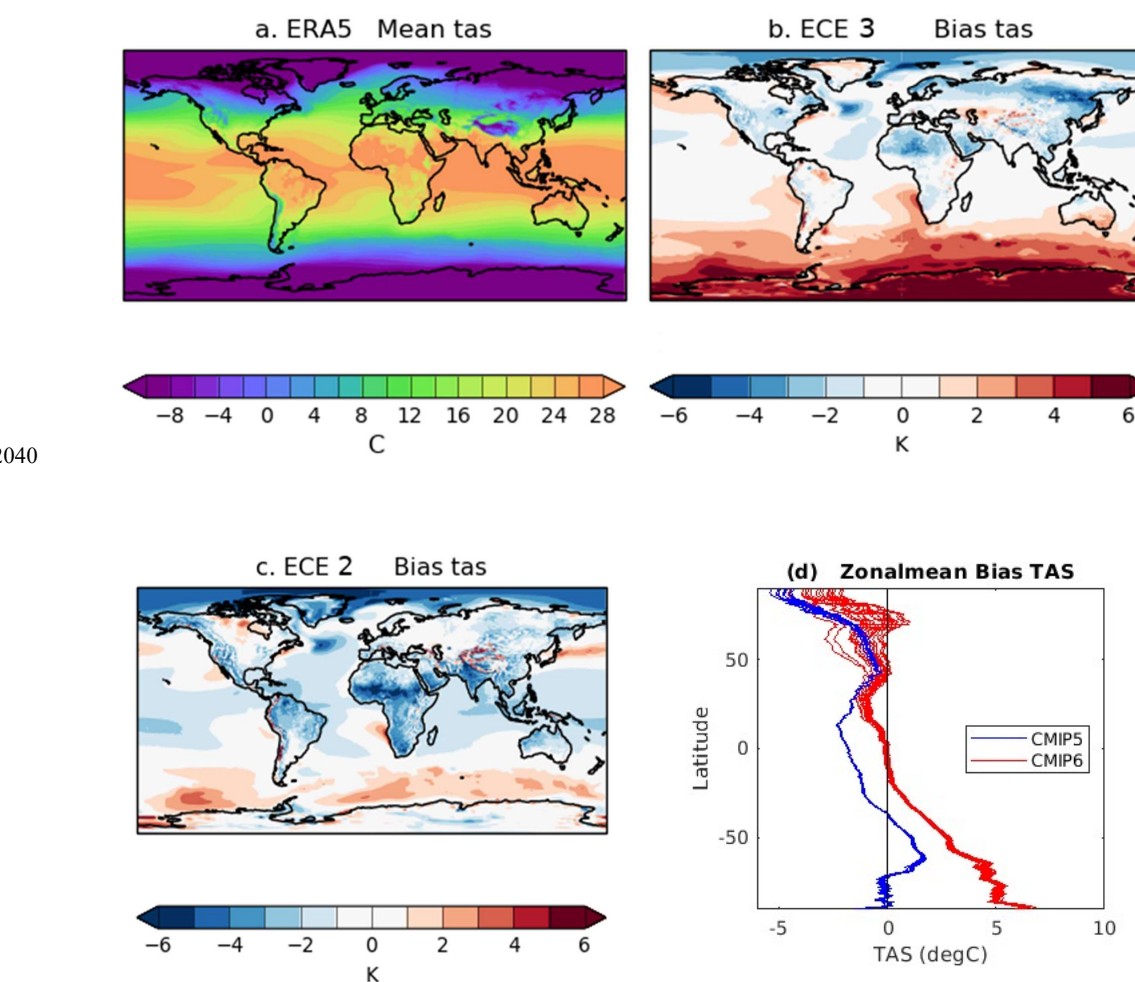


Figure 5: (a) ERA5 1980-2010 mean TAS in °C; (b) EC-Earth3 and (c) EC-Earth2.3 ensemble mean biases cf. ERA5 in °C;
(d) Zonal annual mean TAS from EC-Earth2.3 (blue, CMIP5) and EC-Earth3 (red, CMIP6) ensembles.



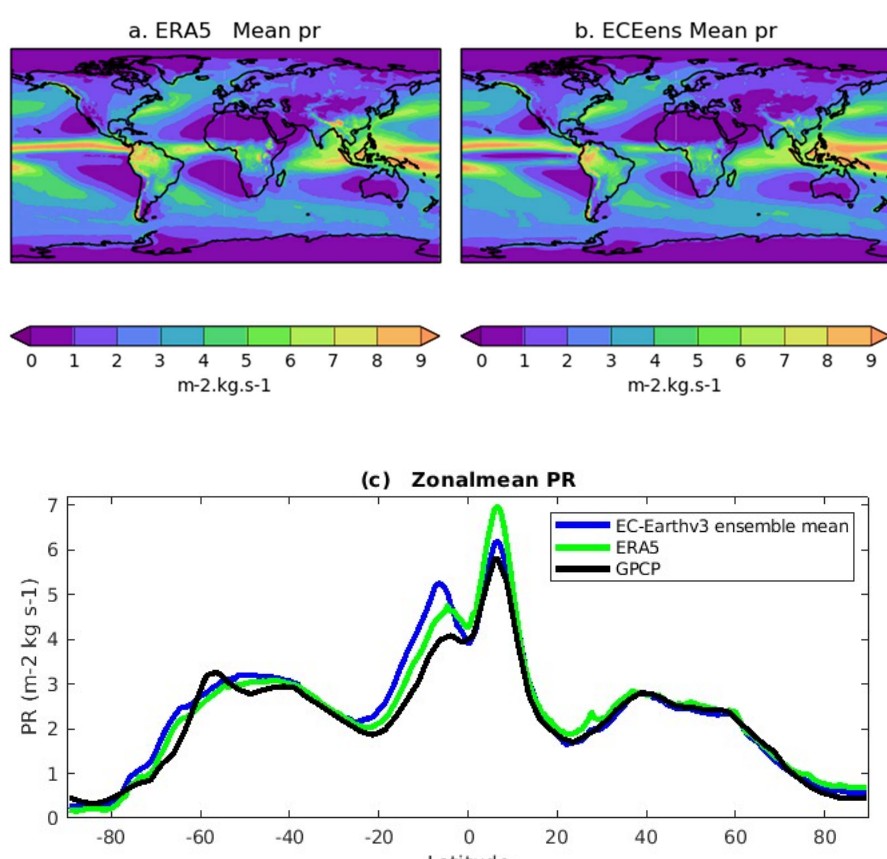


Figure 6: Mean precipitation for the period 1980-2010 for (a) ERA5, (b) EC-Earth3veg and (c) zonal mean precipitation for ERA5 (green), GPCPv2,2 (black) and EC-Earth3veg ensemble mean (blue).




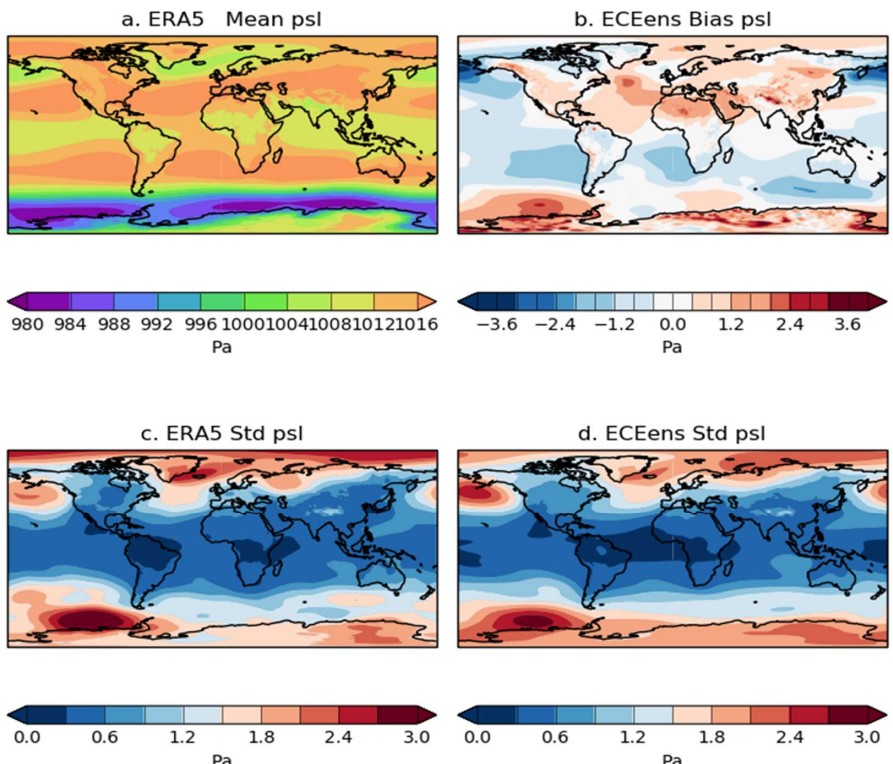

Figure 7: *Upper panel: mean PSL 1980-2010 in hPa for ERA5 (a) and for the EC-Earth3 ensemble bias cf to ERA5 (b).*
*Lower panel: Interannual standard deviation in hPa for ERA5 (c) and for the EC-Earth3 ensemble (d).*


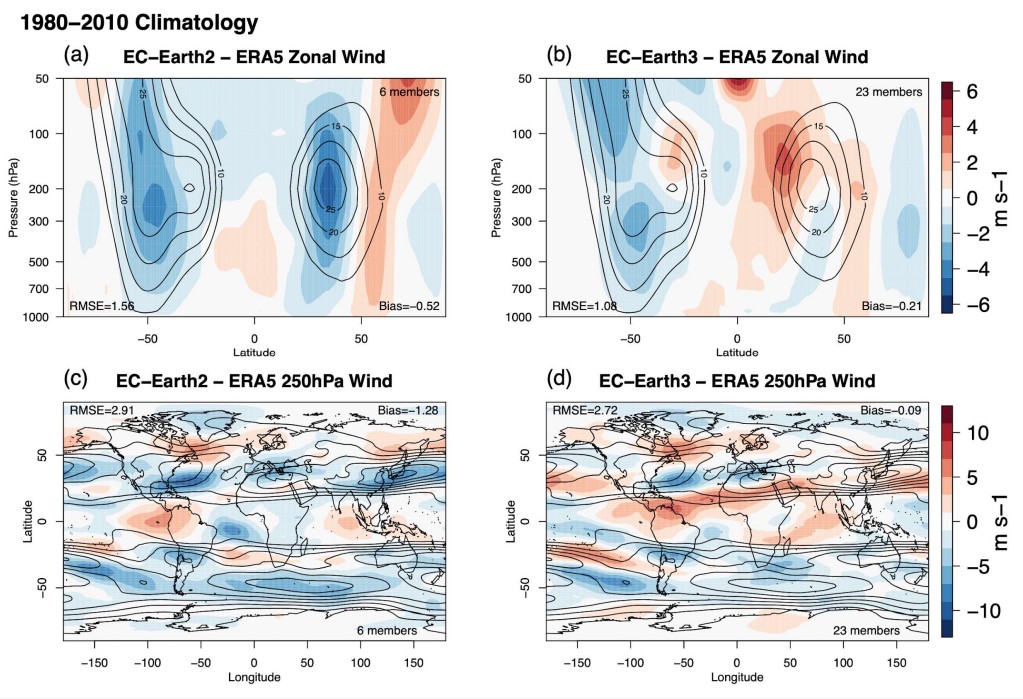

*Figure 8: Top row: Biases (colored shadings) of the zonally averaged zonal component of wind for EC-Earth2.3 (a) and EC-Earth3 against ERA5 Reanalysis over the 1980-2010 period. Contours shows the ERA-5 climatology. Bottom row: Biases (colored shadings) of the 250hPa zonal component of wind for EC-Earth2.3 (a) and EC-Earth3 against ERA5 Reanalysis over the 1980-2010 period. Contours shows the ERA-5 climatology. Root mean square error, mean bias and the number of ensemble members used are reported in each panel.*






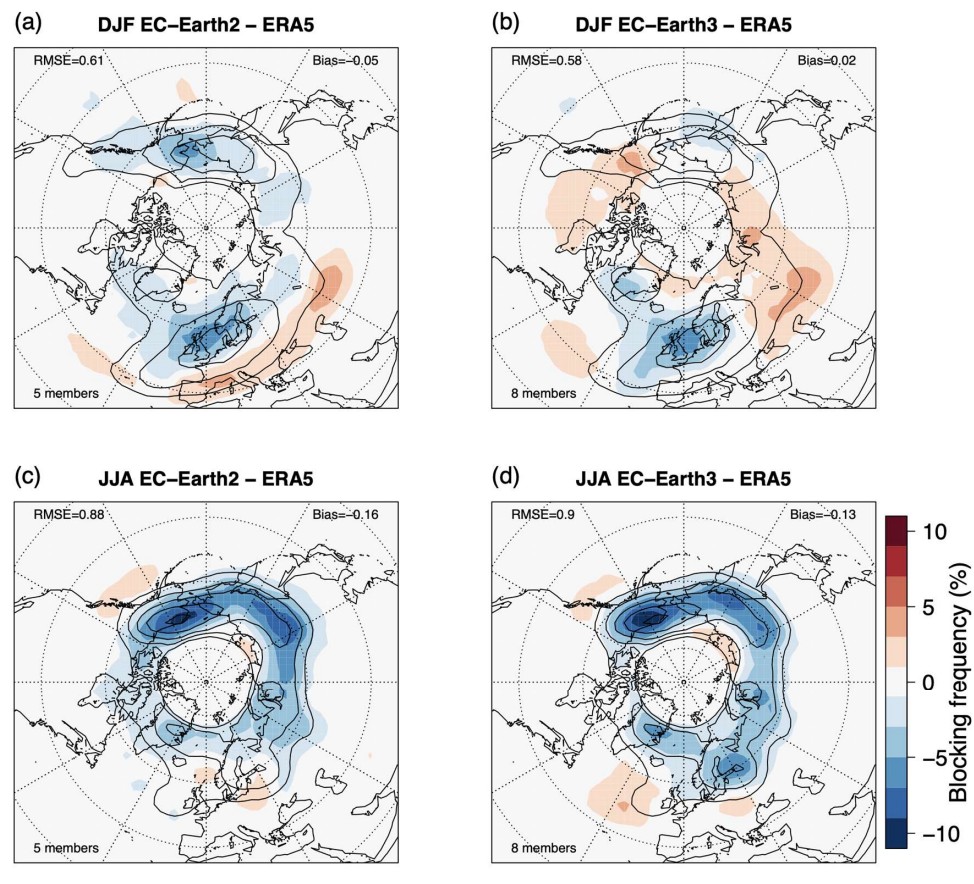


Figure 9: Biases for the blocking frequency according to Davini and D'Andrea (2020) index for EC-Earth2 (a,c) and EC-Earth3 (b,d) against ERA5 Reanalysis over the 1980-2010 period. Contours shows the ERA-5 climatology. Top row is winter, bottom row is summer. Root mean square error, mean bias and the number of ensemble members used are reported in each panel.




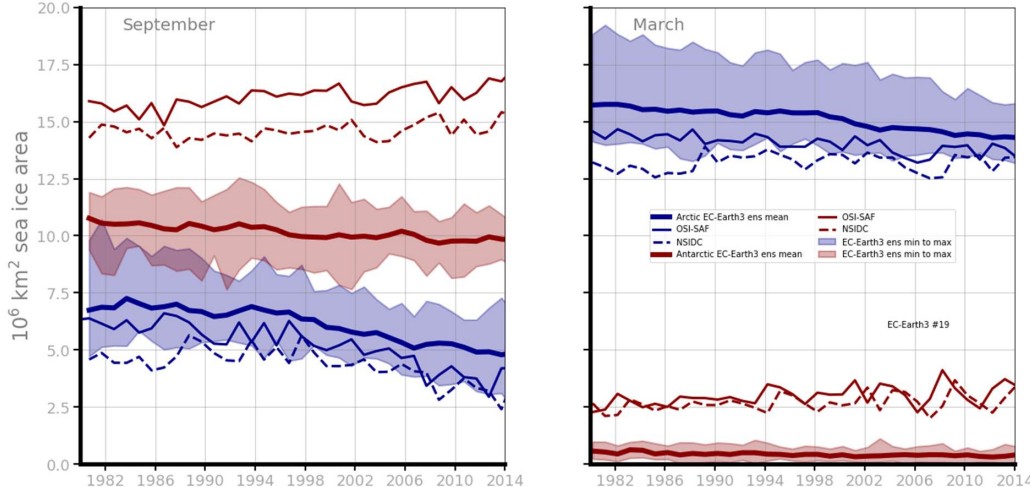

*Figure 10: Time series of Arctic (blue lines) and Antarctic (red lines) sea ice area for both EC-Earth3 (ensemble mean as thick solid lines) and satellite observations (OSI SAF as thin solid lines and NSIDC as dashed lines). The EC-Earth3 ensemble minimum up to maximum value is represented by the shading around the ensemble mean.*


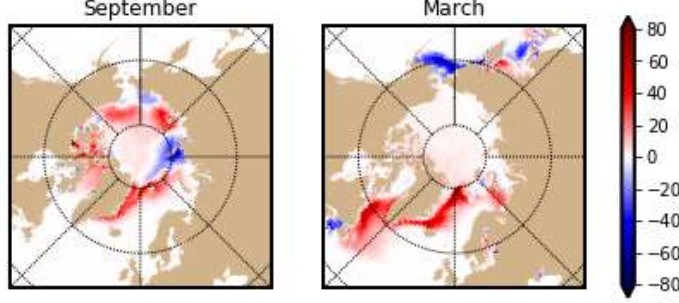

*Figure 11: Difference in Arctic sea ice concentration in % between the ensemble mean of EC-Earth3 and OSI SAF observations in September (left) and March (right), averaged over 1980-2010.*





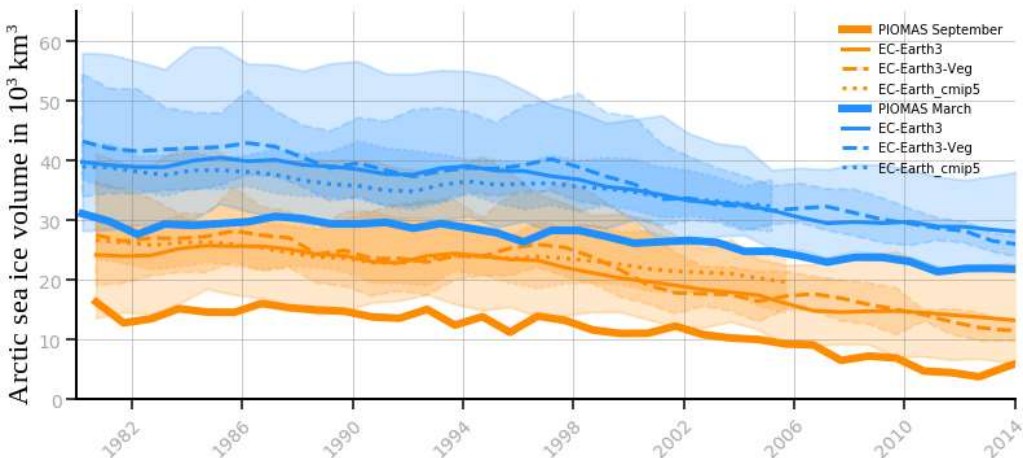

*Figure 12: Time series of September (orange) and March (blue) Arctic sea ice volume for EC-Earth3 (thin solid lines representing the ensemble mean), EC-Earth3-Veg (dashed lines representing the ensemble mean), the CMIP5 version of EC-Earth (dotted lines) and PIOMAS reanalysis (thick solid lines). The EC-Earth3 and EC-Earth3-Veg ensemble minimum and maximum are represented by the same linestyle as their means, but with transparent shading added around the ensemble means.*







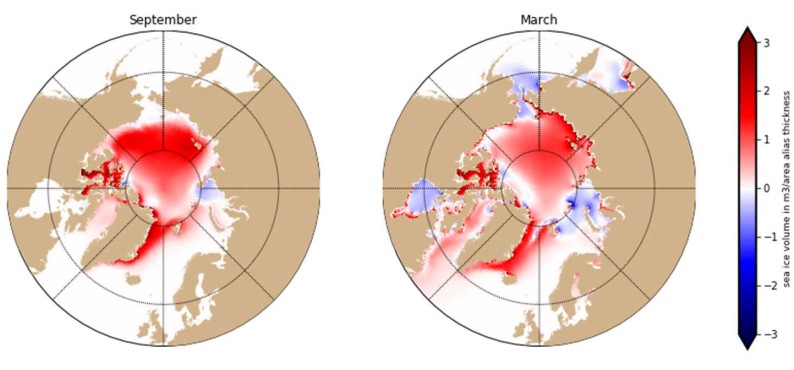

Ensemble average of 1980-2010 averaged model
minus remapped PIOMAS sea ice thickness

*Figure 13: Difference in Arctic sea ice thickness between the ensemble mean of EC-Earth3 and PIOMAS in September (left)
and March (right), averaged over 1980-2010.*


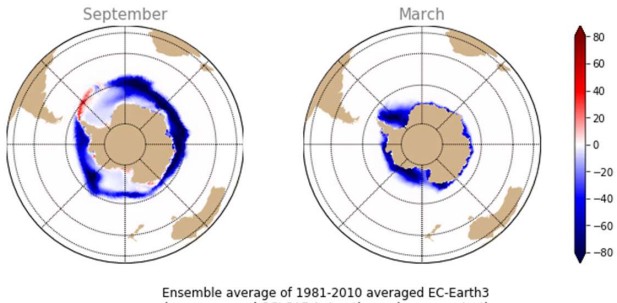

Ensemble average of 1981-2010 averaged EC-Earth3
minus remapped OSI SAF Antarctic sea ice concentration

Figure 14: Difference in Antarctic sea ice concentration between the ensemble mean of EC-Earth3 and OSI SAF
observations in September (left) and March (right), averaged over 1980-2010.





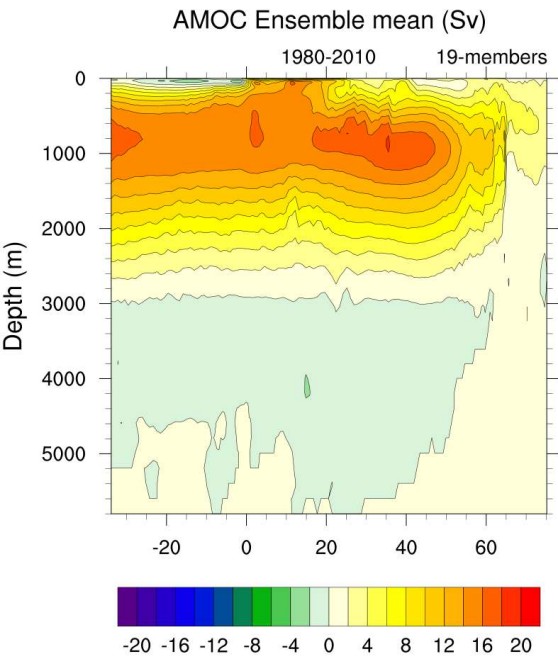

Figure 15: AMOC streamfunction in the depth and latitude plane for the EC-Earth3 ensemble mean, averaged over 1980-
2010, in Sv. The x-axis denotes degrees latitude.

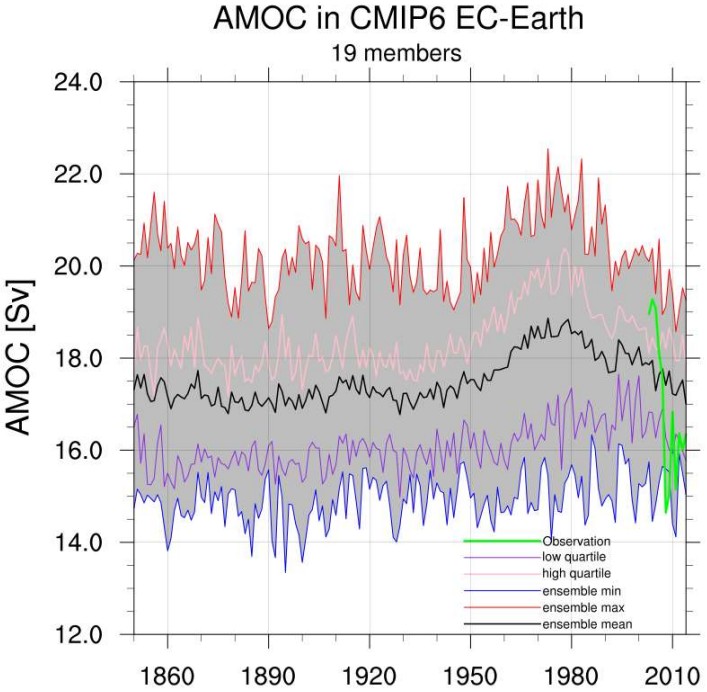

Figure 16: Time series of maximum AMOC (maximum between 24.5-27.5 N) for ensemble mean (black), low quartile
(purple), high quartile (pink), ensemble minimum (blue) and ensemble maximum (red) are shown for 19 members of EC-
Earth3. Observation data is shown with green color from RAPID-MOCHA array observations (Smeed et al., 2017).

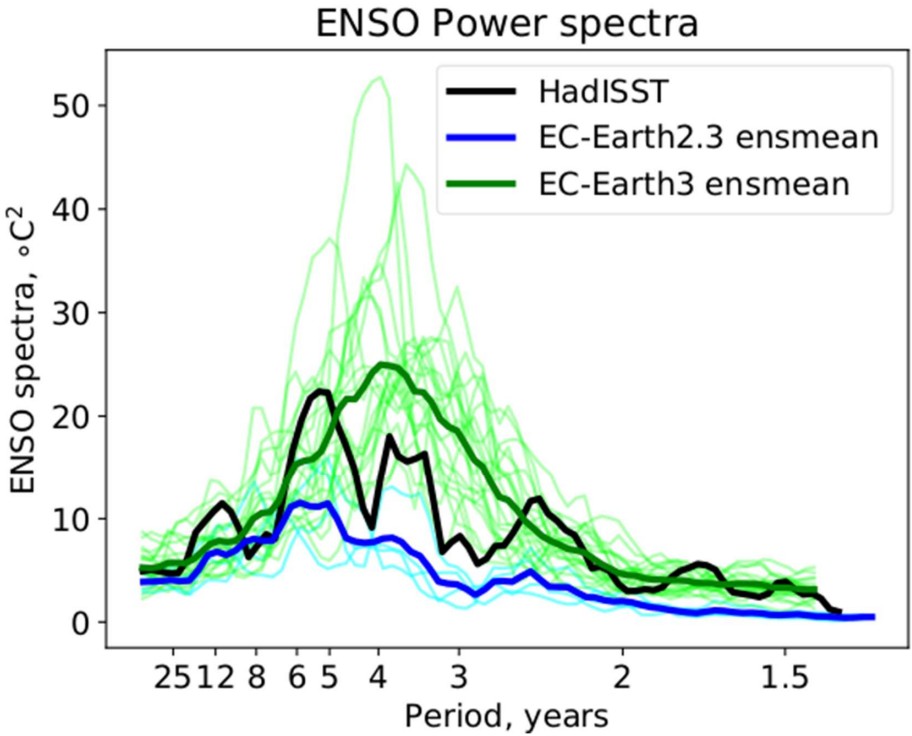

***Figure 17***: *Power spectra of Niño 3.4 Sea Surface Temperature (SST) time series for HadISST observations (black thick line), 20 EC-Earth3 ensemble members (light green lines) and 4 EC-Earth 2.3 ensemble members (cyan lines). The EC-Earth3 and EC-Earth2.3 ensemble means are shown in green and blue thick lines respectively. The Niño 3.4 is calculated between latitudes 5N to 5S and longitudes 170W-120W. The power spectra was calculated for the 1900-2009 period.*


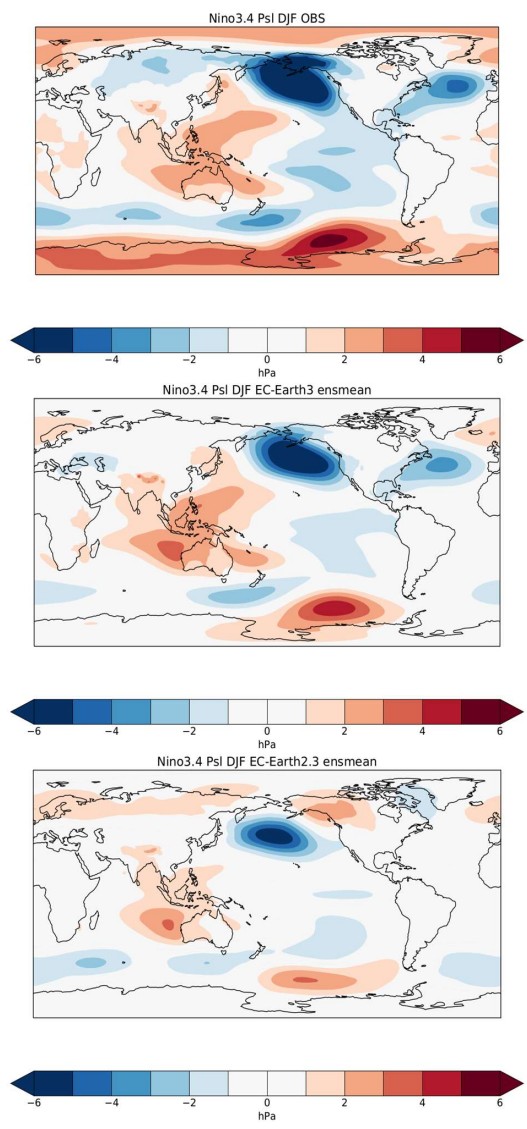


**Figure 18:** Regression of Nino3.4 SST index into Sea Level Pressure during winter (DJF) for ERA-20C reanalysis (upper panel), 20 member EC-Earth3 ensemble mean (middle panel), 4 member EC-Earth2.3 ensemble mean (lower panel). Analyses are based on three-month means Dec-Feb.

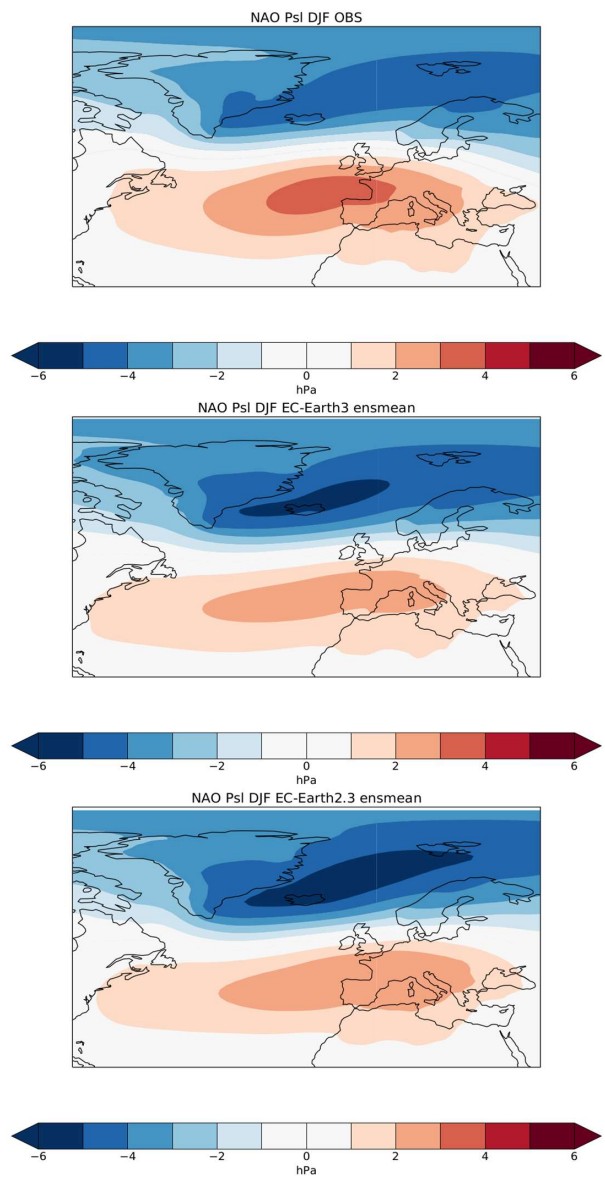


*Figure 19: Regression of Sea Level Pressure (SLP) anomalies to NAO index during winter (December to February) for ERA-20C reanalysis (upper panel), 20 member EC-Earth ensemble mean (middle panel) and 4 member EC-Earth2.3 ensemble mean (lower panel).*




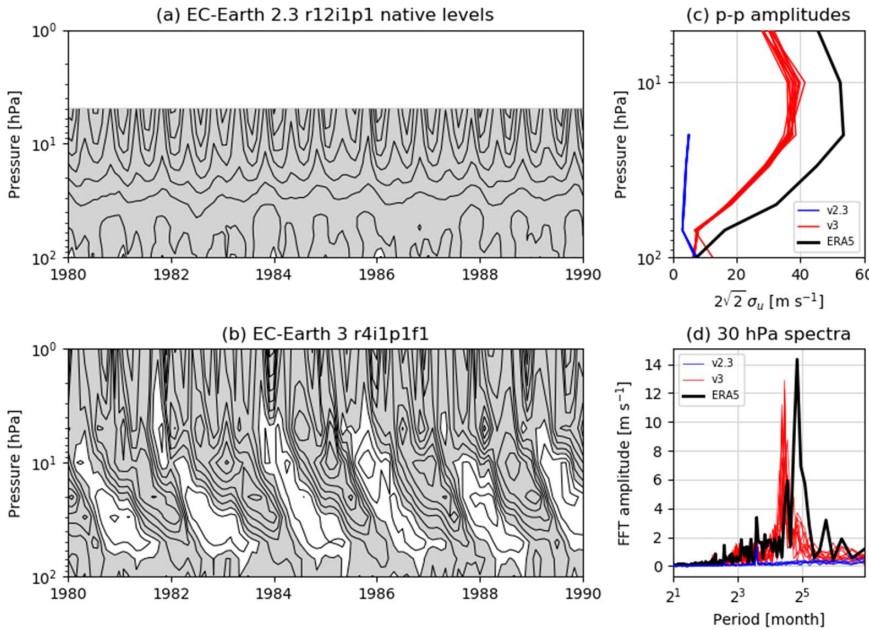

Figure 20: Ten years of the equatorially (within 5 degrees from the equator) averaged monthly mean zonal wind in EC-Earth2.3 (r12i1p1, a) and EC-Earth3 (r4i1p1f1, b), as a function of time and altitude. Easterly winds are shaded in gray, and the contour interval is 5 m/s. The peak-to-peak amplitude profiles of the equatorial zonal wind, estimated from the temporal standard deviation for the period 1980-2010, are reported in (c) for ERA5 (black), 12 CMIP6 (red) and 6 CMIP5 (blue) ensemble members. In plot (d) the Fourier spectra at the 30 hPa level for ERA5 (black), and the same experiments of plot (c), are shown. Only for plot (a), results for EC-Earth2.3 are interpolated onto pressure levels from native model levels.



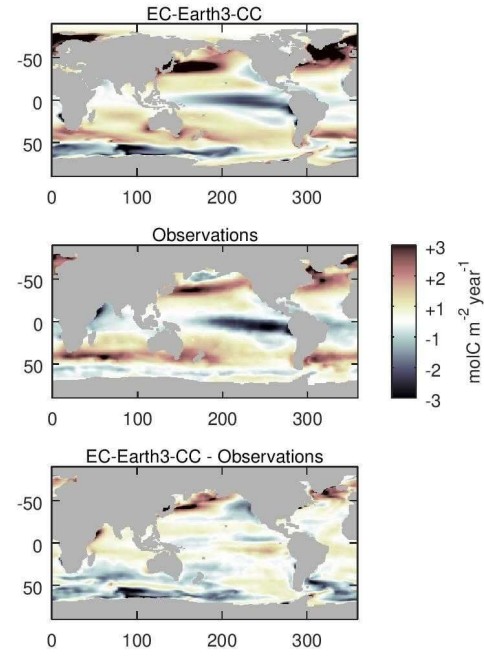

Figure 21: Air-sea CO2 flux averaged over the period 1982-2014 for EC-Earth3-CC (upper panel), the observation-based
reconstruction by Landschutzer et al. (2016; middle panel) and their difference (lower panel).



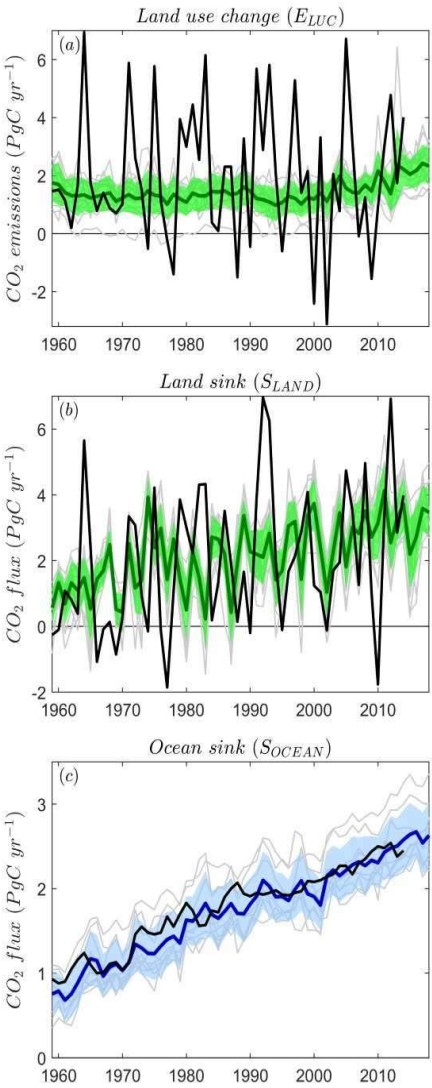

Figure 22: CO2 exchanges between the atmosphere and the terrestrial biosphere for (a) CO2 emissions from land-use change (ELUC) and (b) Land CO2 sink (SLAND) with individual DGVMs (grey), multi-model mean (dark green) and its range (±1σ, light green shading) from GCP overlaid with EC-Earth3-Veg fluxes (black). (c) shows the CO2 exchange between atmosphere–ocean (SOCEAN) with individual ocean models (grey), multi-model mean (dark blue) and its range (±1σ, light blue shading) from GCP overlaid with EC-Earth3-CC fluxes (black).