# Peer review of "The EC-Earth3 Earth System Model for the Climate Model Intercomparison Project 6"

_Geoscientific Model Development, 2020_

## Community Comment (CC1)

Referee review of Doscher et al: "The EC-Earth3 Earth System Model for the Climate Model Intercomparison Project 6"

**General comments**

This manuscript describes the EC-Earth3 modelling system and components as used for the CMIP6 experiments. It describes different configurations, model performance, tuning, reproducibility on different supercomputers, and some aspects of scientific performance.

The manuscript is very well written, logically arranged, and gives a good overview of what model components are used in different configurations and how the model performs compared to the CMIP5 equivalent. As well as describing the EC-Earth system, some aspects such as the reproducibility on different machines may be more generally applicable. As noted below, there are several places where a little more explanation could make some of this work more relevant to other models.

**Specific comments**

Line 83: Are google hits a good measure, since they will vary over time.

Line 124: I realise that you do this later, but maybe the references for the different model components should be here as well or instead of later?

Line 179: Do different configurations of the model have different E-P imbalances, and hence is this flux corrector changed? Also how do future projections work, might having this corrector affect how the future change in runoff is simulated?

L219: can you state which timestep is used in the final model (perhaps refer to the table)

L239: I think this sentence would be clearer if rewritten, e.g.: The goal was to maintain the same atmospheric tuning as much as possible, and only modify the ocean and sea-ice parameters...

L386: Do I understand that this is not a dynamic ice sheet (i.e. it cannot grow or shrink). It may be worth noting this just for clarity, or if I misunderstand then clarifying what the ice sheet can do.

L516: was the closure of Bechtold included in the model (it is implied but not said).

L516: for what reason was the Rayleigh friction included - the other changes have reasons why they were included.

L566: it might be useful to briefly mention what processes are missed by the MACv2-SP scheme, such as natural aerosol variability.

L1025: Can you say any more about the Southern Ocean warm bias? For a 1 degree model this seems quite large.

L1142. You note that the AMOC strength is close to observations, but why is there no mention of the northward heat transport, which is as or more important for the climate state. Some mention of how this compares to observations would be welcome.

L1196: You make no mention of the large range in power of the different ensemble members. For example, does the member with the strongest ENSO power have any other climatological differences, is such as range understandable?

L1213: Are there any hypotheses for the reason for the improvement in the ENSO-NAO that could inform other models?

Table 2: It is implied that the timestep in the ocean and atmosphere are the same, could this be stated explicitly.

Table 3: The variable "Sensitivity of non solar heat flux" may need more explanation.

Table 12: I'm not sure what is meant by "Mem.B. is the division between the theoretical memory of a memory and the real one".

Table 13: I'm slightly concerned that the web links used in the table here will not be persistent years hence, references to the datasets themselves used may be better (or in addition).

Fig. 2c: I hope the quality of this will improve, the black dots are difficult to see.

Fig. 4: would be nice to have this split into land and ocean to better understand.

Fig. 6: The bias in pr might be more instructive than just the full field – as you have done for the other variables.

Fig. 10: In Fig. 10b have EC-Earth#19 on it?

Fig. 11: I assume the quality of this figure will improve in the final version.

Fig. 15: I confess I don't find this figure very instructive as it is, as a rather bland mean with no orography shown, in z-space. I think it would be much more useful if you showed (either or both) of the overturning in density space, and/or some measure of where the variance between ensemble members occurs.

Fig. 17: big range

**Technical corrections**

Line 82: Need an open bracket (e.g. Koenigk

Line 219: testwise – is this a typo?

L241: I think sought may be better than searched.

L264: I think you mean "... interactive vegetation (using LPJ-GUESS)..."

L311: Repeat of the title of the next section

L455: typo "and. Is"

L1197: "impact on"

L1352: "though n"

Table 6: 4.2e-5  $\rightarrow$  4.2E-5 for consistency.

Table 11: the formatting of the table is slightly off for each row.

Fig. 18 caption: Regression of Nino3.4 SST index onto...

---

## Author Comment (AC1)

**Reply to reviewer CC1**

We are grateful for the reviewers' insightful and constructive comments and have addressed all issues as described below. The original reviewer questions and comments are colored in blue.

**Line 83: Are google hits a good measure, since they will vary over time.**

**Reply:**

We acknowledge the issue of results varying with time and we accept the criticism and omit the complete sentence and Google scholar is not mentioned any more in the article.

**Line 124: I realise that you do this later, but maybe the references for the different model components should be here as well or instead of later?**

Reply: We now have included references already here.

**New text:**

"EC-Earth3 comprises model components for various physical domains and system components describing atmosphere, ocean, sea ice, land surface, dynamic vegetation, atmospheric composition, ocean biogeochemistry and the Greenland ice sheet. The component models are described in section 3. The atmosphere and land domains are covered by ECMWF's IFS cycle 36r4 (based on IFS system 4,

https://www.ecmwf.int/sites/default/files/elibrary/2011/11209-new-ecmwf-seasonal-forecast -system-system-4.pdf), which is supplemented with a coupling interface to allow boundary data exchange with other components (ocean, dynamic vegetation, aerosols and atmospheric chemistry, etc). The NEMO3.6 (Madec 2008, Madec et al., 2015) and LIM3 (Vancoppenolle et al., 2009; Rousset et al, 2015) models are the ocean and sea-ice components, respectively. Biogeochemical processes in the ocean are simulated by the PISCES model (Aumont et al. 2015). Both LIM3 and PISCES are code-wise integrated in NEMO. Dynamical vegetation, land use and terrestrial biogeochemistry are provided by LPJ-GUESS (Smith et al., 2014, Lindeskog et al., 2013). Aerosols and chemical processes in the atmosphere are described by TM5. The ice sheet model PISM (Bueler and Brown, 2009, and Winkelmann et al., 2011, The PISM Team, 2019) is optionally utilized to model the Greenland ice sheet."

Line 179: Do different configurations of the model have different E-P imbalances, and hence is this flux corrector changed? Also how do future projections work, might having this corrector affect how the future change in runoff is simulated?

**Reply:**

The compensating flux by the corrector is calculated separately for different resolutions, since different resolutions give different results. The effects are described both here and in the section "Low resolution configuration". Correctors are derived for observed climate and applied throughout future scenario periods without change. Sensitivity experiments concerning the effects on future runoff have not been carried out. Sea level variables in ESMs such as EC-Earth are generally not used directly for estimates of expected future

sea level rise. These are rather derived indirectly from different types of model and observations.

We add additional text:"The compensating flux by the corrector is calculated separately for different resolutions, since different resolutions give different results. The effects are also described in the section "Low resolution configuration". Correctors are derived for observed climate and applied throughout future scenario periods without change. Sensitivity experiments concerning the effects on future runoff have so far not been carried out."

L219: can you state which timestep is used in the final model (perhaps refer to the table)

Reply:

We added new timestep information to the text, and table 2 has been extended: "In final model configurations timesteps ranging from 900s (high resolution) to 3600 s (low resolution) have been used; see table 2."

L239: I think this sentence would be clearer if rewritten, e.g.: The goal was to maintain the same atmospheric tuning as much as possible, and only modify the ocean and sea-ice parameters...

Reply:

The text has been modified accordingly.

Modified text:

"The goal was to maintain the same atmospheric tuning as much as possible, and only modify the ocean and sea-ice parameters"

L386: Do I understand that this is not a dynamic ice sheet (i.e. it cannot grow or shrink). It may be worth noting this just for clarity, or if I misunderstand then clarifying what the ice sheet can do.

Reply:

PISM is a dynamic ice sheet model. It handles the ice sheet dynamical and thermodynamical processes, including ice flow, subglacial hydrology, bed deformation, as well as the basal ice melt. The text has been updated accordingly

New added text:

"GrIS handles the ice sheet dynamical and thermodynamical processes, including ice flow, subglacial hydrology, bed deformation, as well as the basal ice melt."

L516: was the closure of Bechtold included in the model (it is implied but not said).

Reply:

Yes, the Bechtold et al. 2014 closure is implemented in the model.

We modified the sentence to "A closure described by Bechtold et al. (2014) **improving** the diurnal cycle of convection **has been implemented** in EC-Earth3.

L516: for what reason was the Rayleigh friction included - the other changes have reasons why they were included.

**Reply:**

We modified the text:

"...Rayleigh friction was activated in EC-Earth IFS for all resolutions to avoid unphysically large wind speeds at higher resolution.

L566: it might be useful to briefly mention what processes are missed by the MACv2-SP scheme, such as natural aerosol variability.

**Reply:**

We added text:

"As EC-Earth3 uses MACv2-SP in combination with a pre-industrial aerosol climatology, natural aerosol variability is only accounted for via the prescribed seasonal cycle of the climatology. Furthermore, MACv2-SP only captures the seasonal cycle and long-term changes in the optical properties and derived CDNC impact factor of anthropogenic aerosols. Diurnal variability in aerosol amounts or properties is not explicitly described. Day-to-day variability is only included to the extent captured by the seasonal cycles of the pre-industrial climatology and MACv2-SP. Of the interannual variability in the amount and properties of anthropogenic aerosols, only the long-term changes in plume strengths, which are assumed to covary with the 11-year averaged emissions of SOx plus NH3 in the associated countries, are accounted for. Changes in the spectral distribution of the optical properties, the single-scattering albedo and asymmetry factor of anthropogenic aerosols due to long-term changes in their size distribution and composition are ignored by MACv2-SP."

L1025: Can you say any more about the Southern Ocean warm bias? For a 1 degree model this seems quite large.

**Reply:**

The text has been updated with more information:

"Most coupled climate models suffer from a warm southern ocean (SO) bias (Hyder et al 2018).

In EC-Earth3 configurations, the warm bias is found in all seasons. Large parts of the bias have been attributed to biases in short wave cloud radiative effects. Modifications in the cloud scheme and the representation of supercooled liquid water made in more recent versions of IFS, including cycle 45r1 (Forbes and Ahlgrimm, 2014; Forbes et al., 2016), together with the introduction of the new ecRad radiation scheme in cycle 43r3 (Hogan et al., 2017) have been shown to substantially reduce these biases."

We also added new references:

Hyder, P., Edwards, J.M., Allan, R.P., Hewitt, H.T., Bracegirdle, T.J., Gregory, J.M., Wood, R.A., Meijers, A.J., Mulcahy, J., Field, P., Furtado, K., Bodas-Salcedo, A., Williams, K. D., Copsey, D., Josey, S. A., Liu, C., Roberts, C.D., Sanchez, C., Ridley, J., Thorpe, L., Hardiman, S. C., Mayer, M., Berry, D. I., ansd Belcher, S. E.: Critical Southern Ocean

climate model biases traced to atmospheric model cloud errors, Nat. Commun., 9, 3625, https://doi.org/10.1038/s41467-018-05634-2, 2018.

Forbes, R. M., and Ahlgrimm, M.: On the representation of high-latitude boundary layer mixed-phase cloud in the ECMWF global model, Mon. Wea. Rev., 142, 3425–3445, https://doi.org/10.1175/MWR-D-13-00325.1, 2014.

Forbes, R., Geer, A., Lonitz, K., and Ahlgrimm, M.: Reducing systematic error in cold-air outbreaks, ECMWF Newsletter No. 146, 17–22, 2016.

Hogan, R., Ahlgrimm, M., Balsamo, G., Beljaars, A., Berrisford, P., Bozzo, A., Di Giuseppe, F., Forbes, R. M., Haiden, T., Lang, S., Mayer, M., Polichtchouk, I., Sandu, I., Vitart, F., and Wedi, N.: Radiation in numerical weather prediction, ECMWF Technical Memorandum No. 816, 49 pp., https://doi.org/10.21957/2bd5dkj8x, 2017.

L1142. You note that the AMOC strength is close to observations, but why is there no mention of the northward heat transport, which is as or more important for the climate state. Some mention of how this compares to observations would be welcome.

Reply:

We now mention the northward heat transport.

**Added text:**

"The ocean heat transport (Figure 16) is related to the AMO. North of 200 N it shows values slightly lower than observation estimates from Trenberth et al. (2019), that covers the period from 2000-2014."

We also added a new figure illustrating the northward heat transport (new Figure 16) and a new reference:

Trenberth, K. E., Zhang, Y., Fasullo, J. T., & Cheng, L. (2019). Observation-Based Estimates of Global and Basin Ocean Meridional Heat Transport Time Series, Journal of Climate, 32(14), 4567-4583. Retrieved May 20, 2021, from

https://journals.ametsoc.org/view/journals/clim/32/14/jcli-d-18-0872.1.xml

L1196: You make no mention of the large range in power of the different ensemble members. For example, does the member with the strongest ENSO power have any other climatological differences, is such as range understandable?

Reply:

We added background information to the text.

The range among the different members spectra is considerable. Climatologically, most ensemble members show only small differences over the tropics when compared with the whole ensemble mean. Although the members with the most energetic ENSO share some

climatological features (cold Arctic and Labrador seas) compared to the ensemble mean, the reason why they have developed a more energetic ENSO remains unclear.

L1213: Are there any hypotheses for the reason for the improvement in the ENSO-NAO that could inform other models?

Reply: We added a hypothesis to the text:

"Previous research has linked La Niña/El Niño events to the positive/negative NAO patterns (Fereday at al. 2020). Although this link is relatively weak due to the fact that internal atmospheric variability is large in the North Atlantic European (NAE) region (Brönnimann, 2007), it depends on ENSO strength (Jiménez-Esteve & Domeisen, 2019; Toniazzo & Scaife, 2006). Therefore a more energetic ENSO (more comparable in scale with observations) such as in the current EC-Earth3 could impact on the intensity and sign of NAO."

We also added references:

Brönnimann, S. (2007). Impact of El Niño–Southern Oscillation on European climate. Reviews of Geophysics, 45, RG3003. https://doi.org/10.1029/2006RG000199

Fereday, D., Maidens, A., Arribas, A., Scaife, A., & Knight, J. (2012). Seasonal forecasts of Northern Hemisphere winter 2009/10. Environmental Research Letters, 7(3), 034031.

Jiménez-Esteve, B., & Domeisen, D. (2019). Nonlinearity in the North Pacific atmospheric response to a linear ENSO forcing. Geophysical Research Letters, 46, 2271–2281.

Toniazzo, T., & Scaife, A. (2006). The effect of non-linearity on winter ENSO teleconnections over Europe. Geophysical Research Letters, 33, L24704. https://doi.org/10.1029/2006GL027881

Table 2: It is implied that the timestep in the ocean and atmosphere are the same, could this be stated explicitly.

Reply: Table 2 is now updated with more time step information.

|                                                           | Resolution atmosphere | Resolution ocean     | Timestep                                                  |
|-----------------------------------------------------------|-----------------------|----------------------|-----------------------------------------------------------|
| Standard resolution                                       | T255L91 (~80 km)      | ORCA1L75 (1 deg)     | 2700 s (atm)
2700 s (oce)
2700 s (coupling)  |
| Low resolution EC-
Earth3-LR and EC-
Earth3-Veg-LR  | T159L62 (~125 km)     | ORCA1L75 (1 deg)     | 3600 s (atm)
2700 s (oce)
10800 s (coupling) |
| High resolution (EC-
Earth3P-HR and EC-
Earth3-HR)) | T511L91 (~40 km)      | ORCA025L75 (0.25deg) | 900 s (atm)
900 (oce)
2700 s (coupling)      |

 Table 2: Commonly used resolutions for CMIP6. The suffixes LR and HR are added to the name of the model configuration

 where applicable (e.g. EC-Earth3-Veg-LR)

**Table 3: The variable "Sensitivity of non solar heat flux" may need more explanation.**

Reply:

÷

We now explain the variable in the text:

"The "Sensitivity of non solar heat flux" refers to the sensitivity with respect to sea ice surface temperature. The variable is used by the sea ice model to distribute the non-solar heat fluxes over different ice categories."

Table 12: I'm not sure what is meant by "Mem.B. is the division between the theoretical memory of a memory and the real one".

Reply: "Mem. B." means Memory Bloat. It represents the division between the size (bytes) of the total number of prognostics variables used in the code and calculated manually from the code of the application (which is called theoretical memory) and the memory consumed and instrumented (using top command for example) during the execution of the application (which is called real).

We added the term "Memory Bloat" to the caption.

Table 13: I'm slightly concerned that the web links used in the table here will not be persistent years hence, references to the datasets themselves used may be better (or in addition).

**Reply:**

We use complete references for the CMIP forcing whenever possible. Unfortunately not all of those exist as of today. We are afraid we need to stick with at least 2 web links. One of the links was replaced by three references:

Hurtt, G. C., Chini, L., Sahajpal, R., Frolking, S., Bodirsky, B. L., Calvin, K., Doelman, J. C., Fisk, J., Fujimori, S., Klein Goldewijk, K., Hasegawa, T., Havlik, P., Heinimann, A., Humpenöder, F., Jungclaus, J., Kaplan, J. O., Kennedy, J., Krisztin, T., Lawrence, D., Lawrence, P., Ma, L., Mertz, O., Pongratz, J., Popp, A., Poulter, B., Riahi, K., Shevliakova, E., Stehfest, E., Thornton, P., Tubiello, F. N., van Vuuren, D. P., and Zhang, X.: Harmonization of global land use change and management for the period 850–2100 (LUH2) for CMIP6, Geosci. Model Dev., 13, 5425–5464, https://doi.org/10.5194/gmd-13-5425-2020, 2020

Hurtt, G. C., Chini, L., Sahajpal, R., Frolking, S., Bodirsky, B. L., Calvin, K., Doelman, J., Fisk, J., Fujimori, S., Goldewijk, K. K., Hasegawa, T., Havlik, P., Heinimann, A., Humpenöder, F., Jungclaus, J., Kaplan, J., Krisztin, T., Lawrence, D., Lawrence, P., Mertz, O., Pongratz, J., Popp, A., Riahi, K., Shevliakova, E., Stehfest, E., Thornton, P., van Vuuren, D., Zhang, X. (2019). Harmonization of Global Land Use Change and Management for the Period 850-2015. Version 20190529. Earth System Grid Federation. https://doi.org/10.22033/ESGF/input4MIPs.10454

Hurtt, G. C., Chini, L., Sahajpal, R., Frolking, S., Bodirsky, B. L., Calvin, K., Doelman, J., Fisk, J., Fujimori, S., Goldewijk, K. K., Hasegawa, T., Havlik, P., Heinimann, A., Humpenöder, F., Jungclaus, J., Kaplan, J., Krisztin, T., Lawrence, D., Lawrence, P., Mertz, O., Pongratz, J., Popp, A., Riahi, K., Shevliakova, E., Stehfest, E., Thornton, P., van Vuuren, D., Zhang, X. (2019). Harmonization of Global Land Use Change and Management for the Period 2015-2300. Version 20190529. Earth System Grid Federation. <a href="https://doi.org/10.22033/ESGF/input4MIPs.10468">https://doi.org/10.22033/ESGF/input4MIPs.10468</a>

Fig. 2c: I hope the quality of this will improve, the black dots are difficult to see.

Reply:

We have replaced the figure with improved dot density.

Fig. 4: would be nice to have this split into land and ocean to better understand.

Reply:

The figure has been split up in three parts: global, land-only and ocean-only. The caption has been adjusted, and the text has been modified with a few words.

"The bias in the EC-Earth3 global mean TAS is mainly due to a warmer ocean and especially due to a strong warm bias in the Southern Ocean as we will show below."

---

## Author Comment (AC2)

**Reply to reviewer RC1**

**General Comments:**

This is a nice paper documenting the new EC-Earth model. It covers a lot of the material relevant for the community to understand this model. I particularly appreciated the detailed sections on model tuning, and replication. I think the paper could be improved by streamlining the introduction and conclusions, reducing repition and addressing the specific questions raised below regarding specific details about the modelling system. Once that is done, I support publication.

**Reply:**

We are grateful for the constructive and insightful comments and we have addressed all the specific issues raised by the reviewer. The original reviewer comments are colored in bue. Concerning a possible streamlining of the introduction and conclusions, we think that both need to be readable even for a quick reader who has not studied the paper in detail. Therefore we would prefer to keep the introduction and summary roughly as is.

**By line:**

45 "largely improved physical performance". In the following sections, various biases are documented, some of which have improved (e.g. winds) and others which have degraded (e.g. tas). No overall assessment of the model skill / improvement is made. This could be useful (also see comment for conclusions).

**Reply:**

We find it important to go through the individual biases and issues to allow for understanding in more detail than what overall assessments can provide. We agree that a Reichler-Kim-type plot for example could give a good idea about overall improvements. However, it appears difficult within the time and staff resources given to produce proper plots. We therefore suggest leaving more overarching comparisons to forthcoming publications in conjunction with intercomparisons of CMIP5/6 data. An example is the recent publication by Brands (2021), who analyses circulation performance for a number of models including EC-Earth2 and 3.

Brands, S. (2021). A circulation-based performance atlas of the CMIP5 and 6 models for regional climate studies in the northern hemisphere. *Geoscientific Model Development Discussions*, 1-48.

52: Possible citation is: Flato, G.M. (2011), Earth system models: an overview. WIREs Clim Change, 2: 783-800. https://doi.org/10.1002/wcc.148

**Reply:**

We have added the reference:

Flato, G. M. (2011). Earth system models: an overview. Wiley Interdisciplinary Reviews: Climate Change, 2(6), 783-800.

54-61: Some references would be appropriate.

Reply: We added a reference.

**New text:**

"The Paris Climate accord is calling for limiting climate change "well below 2°C and to pursue efforts to limit the increase to 1.5°C". ESMs represent our most relevant tools available for exploring the emission pathways necessary for achieving this goal (Kawamiya et al, 2020), as well as for understanding the consequences of not making this target. The Paris agreement requires firm measures of mitigation, including carbon dioxide removal. Given the complexity of the climate system, alternative emission pathways towards this goal can be carefully explored only with Earth System Models (ESMs) which describe the most relevant feedback mechanisms, and provide methods for assessments of uncertainty. ESMs are the primary source of information for understanding the Earth's climate feedbacks, for attributing changes to specific drivers, for future climate projections and predictions, and for the development of mitigation policies."

Kawamiya, M., Hajima, T., Tachiiri, K., Watanabe, S., & Yokohata, T. (2020). Two decades of Earth system modeling with an emphasis on Model for Interdisciplinary Research on Climate (MIROC). Progress in Earth and Planetary Science, 7(1), 1-13.

83-85: These web statistic citations probably need to be more robust given the journal style guide. Normally the date of the access and url is requried. It is relevant here as these statistics will change in time. I also note that this refers to a search, not a url, and Google customizes web searches to users, so different users will get different results. Personally, I cannot verify these numbers, and I get completely different results than reported here when searching cesm "climate model" (7900) and "ec earth climate model" (234). Or for example "community earth system model" (11200) or "community earth system model (cesm)" (5140). It's not clear therefore that this is really a robust metric.

Reply:

We accept the criticism and omit the complete sentence and Google scholar is not mentioned any more in the article.

**87 "development has started in" -> "development started in"**

Reply: This is now corrected.

**102, 104: dynamical Greenland but not Antarctica? What is it physically justified including one major icesheet but not the other?**

Reply:

We argue that a configuration with the Greenland ice sheet is useful for addressing regional interaction with the atmosphere and ocean, and long term developments influenced by the Greenland ice sheet. An ice sheet model for Antarctica is an option for forthcoming model configurations.

119-120: "and it is used in its version 3.3 for CMIP6" -> "and version 3.3 is used for CMIP6"

Reply: This is now corrected

130: TM5 - > reference?

Reply:

We added a reference to TM5 and also for other component models already in this part of the manuscript.

150-151: Is this initialization and forcing data publicly available? Is there a reference or a link to it?

Reply:

The forcing data is publicly available; see table 13. The initial fields are not publicly available, but can be made available on request. A reference to table 13 is added to the text.

165-166: Is this conservation assumed or is it verified? If it is verified, how? A comparison of 3D ocean heat content with TOA fluxes and surface fluxes in a long piControl can be used to verify to first order.

Reply:

The paragraph in questions and the subsequent paragraphs have been rewritten, to improve understanding of the conservation methods applied.

We agree that a comparison of ocean heat content changes with TOA and ocean-atmosphere fluxes would be illustrative, but we cannot mobilize resources for that additional study at this time.

The new text reads:

At the atmosphere-ocean interface, we follow the principle that the ocean provides state variables and the atmosphere sends fluxes (Table 3). Flux formulations correspond to the documentation of IFS CY36R1, section 3, at

https://www.ecmwf.int/en/publications/ifs-documentation. Atmosphere fluxes are remapped onto the ocean grid by a nearest-neighbour distance-based Gauss-weighted interpolation. The energy (solar and non-solar radiation) and mass (evaporation and precipitation) fluxes are treated with a conservation post-processing method during coupling, in which the residual (target minus source grid integrals) is distributed over the target grid, proportional to the original, interpolated value. This does not constitute a locally conservative method, but it does conserve mass and energy of the coupling fields.

The freshwater runoff from land to ocean is derived from a runoff mapper (Table. 4). It uses OASIS3-MCT to interpolate local runoff and ice-shelf calving (from Greenland and Antarctica) to the ocean. The runoff and calving received from the atmosphere and from the surface model HTESSEL are interpolated onto 66 hydrological drainage basins,

remapped onto an intermediate grid by the same method and same post-processing as described above for the mass flux. The resulting runoff to the ocean is evenly and instantaneously distributed along several ocean coastal points connected to each hydrological basin, in the vicinity of the major river outlet. The runoff is even distributed vertically. The distribution depths are taken (read in from a file) from an ocean-only simulation, using a feature of NEMO to save these depths when the NEMO input parameter In\_rnf\_depth\_ini is set to true in the namelist. For a detailed description of the method we refer to the NEMO documentation (https://www.nemo-ocean.eu/doc/node53.html).

In order to avoid a significant long-term sea-surface height reduction in coupled model runs due to a net precipitation - evaporation (P-E) imbalance in the EC-Earth3 atmosphere of about -0.016 mm/day in the historical period, the coupled model implements a runoff flux corrector, which amplifies river runoff by 7.95% in order to compensate for this effect.

171-175: It would be interesting to know more (or see) what the drainage basins look like, and what the distribution over coastal points looks like. Is runoff inserted into a single ocean grid cell at the river mouth? Is it spread more widely? Inserted at the surface? Does runoff have proporties (nutrient, temperature) or is it inserted at SST?

**Reply:**

Here are maps of the drainage basins and the points where the runoff is added as freshwater to the ocean:

The treatment of the runoff is an update of the method that has been developed for EC-Earth v2. In ECE2 the separate maps of drainage basins were used for different model resolutions while in ECE3 we only use one map and let the coupler do the remapping of runoff from the atmosphere to the runoff-mapper, and from the runoff-mapper to the ocean. The freshwater from the runoff in a drainage basin is distributed over several points along the coast in the vicinity of the major river outlet. The freshwater is assumed to enter the ocean with the same temperature as SST.

The runoff is even distributed vertically. The distribution depths are taken (read in from a file) from an ocean-only simulation, using a feature of NEMO to save these depths when the NEMO input parameter In\_rnf\_depth\_ini is set to true in the namelist. For a detailed description of the method we refer to the NEMO documentation (https://www.nemo-ocean.eu/doc/node53.html).

We rewrote the respective paragraph, so that it reads now:

The freshwater runoff from land to ocean is derived from a runoff mapper (Table. 4). It uses OASIS3-MCT to interpolate local runoff and ice-shelf calving (from Greenland and Antarctica) to the ocean. The runoff and calving received from the atmosphere and from the surface model HTESSEL are interpolated onto 66 hydrological drainage basins, remapped onto an intermediate grid by the same method and same post-processing as described above for the mass flux. The resulting runoff to the ocean is evenly and instantaneously distributed along several ocean coastal points connected to each hydrological basin, in the vicinity of the major river outlet. The runoff is even distributed vertically. The distribution depths are taken (read in from a file) from an ocean-only simulation, using a feature of NEMO to save these depths when the NEMO input parameter In\_rnf\_depth\_ini is set to true in the namelist. For a detailed description of the method we refer to the NEMO documentation (https://www.nemo-ocean.eu/doc/node53.html).

176-180: So moisture is not conserved within IFS? Why correct runoff to compensate and not E-P directly? The E-P correction could be distributed with the tendencies. Also, why is the correction diagnosed over the transient historical period as opposed to in a balanced piControl? Is there any evidence that this imbalance/ correction is constant in time, or could it vary? What about snowfall into the ocean? Is this being thermodynamically accounted for (i.e. latent heat to melt snow and bring it to SST)?

**Reply:**

In EC-Earth3 we do not couple E-P but evaporation and precipitation separately (and even distinguish between solid and liquid precipitation) and therefore it would be an extra challenge to put a correction on E-P. The runoff flux correction has been developed early in the model development process when we did experiments with constant present day forcing. This setting was later checked during the piControl-spinup to make sure that there is no drift in the SSH which was the case and no adjustment was necessary. Comparing the P-E imbalance in CMIP6 historical runs with 4xCO2 experiments, we find that the P-E imbalance does not change significantly.

We add new text, also motivated by other reviewer requests:

"The compensating flux by the corrector is calculated separately for different resolutions, since different resolutions give different results. The effects are also described in the section "Low resolution configuration". Correctors are derived for observed climate and applied throughout future scenario periods without change. Comparing the P-E imbalance in CMIP6 historical runs with 4xCO2 experiments, we find that the P-E imbalance does not change significantly."

**189-192: Is this duality lead to unphysical behaviour, e.g. with opposite moisture tendencies in the two components? How does this affect conservation of moisture?**

**Reply:**

Conservation of moisture in the climate system is not affected by coupling to LPJ-GUESS. As we write in L691-696 (original manuscript), LPJ-GUESS sends daily updates to 6 fields (i.e. LAI, cover fraction and cover type for each high and low vegetation tile) for H-TESSEL to use in its biophysical and hydrological calculations of gridcell-averaged soil temperatures and moisture, affecting albedo, latent and sensible heat exchange, runoff and momentum exchange.

However, since, as we write on L615 (original manuscript), "the discretization in HTESSEL is such that coexistence in each grid point of more than one type of low and high vegetation, respectively, is not allowed" we are forced to aggregate the heterogeneity of vegetation simulated by LPJ-GUESS to match the fields used in H-TESSEL, e.g. the high and low types used in the GLCC global map used in the standard HTESSEL configuration.

In LPJ-GUESS, vegetation dynamics are simulated with a daily timestep on six stand types in the land portion of the gridcell; five stands having dynamic gridcell fractions consistent with the LUH2 dataset, namely Natural, Pasture, Urban, Crop, and Irrigated Crop, and one, Peatland, having a fixed gridcell fraction derived from the GLCC global map. For each simulated patch in each stand and gridcell, LPJ-GUESS must simulate its own soil moisture dynamics for the internal vegetation dynamics (especially important following a disturbance event) and coupled conductance-photosynthesis scheme (Smith et al. 2014) to be consistent. In order to maintain this consistency, and to keep LPJ-GUESS vegetation, carbon and nitrogen dynamics as similar as possible to its offline versions, we chose not to use the gridcell-average soil moisture calculated in H-TESSEL uniformly across the LPJ-GUESS patches and stands.

However, we did try to ensure more consistency by a) using H-TESSEL gridcell-average soil temperature in LPJ-GUESS calculations (e.g. for heterotrophic and root respiration), recognising the advantages of the more mechanistic snow and soil thermal dynamics in H-TESSEL, and b) we used the same soil texture in LPJ-GUESS as is used in H-TESSEL (Balsamo et al. 2009), governing hydrological properties in the LPJ-GUESS soil column, including wilting point and field capacity.

Full physical consistency will require a major update to both model components. Surface energy and hydrology calculations will need to be made for each sub-daily timestep, and for each patch. This will require significant development, testing and recalibration, which will likely require the replacement of H-TESSEL with a version of LPJ-GUESS developed to include a full, vertically-resolved surface energy balance coupled to its conductance-photosynthesis scheme (Smith et al. 2014). Steps in the direction of a tighter, more consistent coupling are planned for EC-Earth version 4. However, to date we have experienced only minor inconsistencies. For example, when replacing trees with grasses in tropical regions in global deforestation experiments (Boysen et al. 2020 - now added to the reference list), the high LAI of ungrazed C4 grasses freed from competition with trees can, for that special case, lead to differences in evaporation rates in the two models, with wet soils, low roughness and high wind speeds combining to keep latent heat fluxes high in H-TESSEL, even in the absence of trees with deeper roots.

As we write on L187-189 (original manuscript), our loose coupling for CMIP6 application "... ensures that EC-Earth makes best use of both the advanced biophysics in the HTESSEL land-surface model and of the state-of-the-art vegetation dynamics, land use functionality and terrestrial biogeochemistry (carbon and nitrogen) in LPJ-GUESS." (Boysen, L. R., Brovkin, V., Pongratz, J., Lawrence, D. M., Lawrence, P., Vuichard, N., Peylin, P., Liddicoat, S., Hajima, T., Zhang, Y., Rocher, M., Delire, C., Séférian, R., Arora, V. K., Nieradzik, L., Anthoni, P., Thiery, W., Laguë, M. M., Lawrence, D., and Lo, M.-H.: Global climate response to idealized deforestation in CMIP6 models, Biogeosciences, 17, 5615–5638, https://doi.org/10.5194/bg-17-5615-2020, 2020.)

We think the reviewer's question is largely covered by the original text and more in-depth explanations in the manuscript would bear the risk of duplication. We added the Boysen reference and one sentence to the text in this section ("Configurations"): "However, the conservation of moisture in the climate system is not affected by coupling to LPJ-GUESS"

**195-205: tuning these parameters almost certainly has a significant influence on the ECS of the model. Probably worth noting.**

**Reply:**

We provide background information below, but feel that the topic should not be included in the manuscript, because it would open the new field for ECS dependencies, which is not subject of this paper.

EC-Earth did not tune 20th century climate sensitivity or ECS. Still, it is true that different tuning configurations could significantly affect the evolution of temperatures during the 20th century (Golaz et al. 2013) and possibly significantly affect ECS (Hourdin et al 2017). Experiments which we have conducted recently with EC-Earth3, identifying alternative tuning configurations characterised by the same TOA equilibrium during the preindustrial phase, in which the two tuning parameters RPRCON and ENTRORG have been reduced by 30% and 20% respectively (beyond the limits explored during EC-Earth tuning) lead to an ECS increase of up to 0.35K in coupled 4xCO2 experiments.

Golaz, J.-C., Horowitz, L. W., and Levy, H. (2013), Cloud tuning in a coupled climate model: Impact on 20th century warming, *Geophys. Res. Lett.*, 40, 2246–2251, doi:10.1002/grl.50232.

Hourdin, F., Mauritsen, T., Gettelman, A., Golaz, J., Balaji, V., Duan, Q., Folini, D., Ji, D., Klocke, D., Qian, Y., Rauser, F., Rio, C., Tomassini, L., Watanabe, M., & Williamson, D. (2017). The Art and Science of Climate Model Tuning, Bulletin of the American Meteorological Society, 98(3), 589-602.

**208: "allowing to constuct" -> "allowing construction of"**

Reply: This is now corrected

265: The resolution of the standard "non-LR" ocean was not noted above as far as I see, but here ORCA1 is noted for the "LR" version.

Reply:

The low and standard resolution configurations share the same ocean resolution ORCA1 Resolutions are listed in table 2.

298: How is 100 years of spinup selected? In my experience, this is not enough time, and the model is likely still drifting after 100 years. 500 years, and often 1000+ years is required for the long time scales of the ocean and deep soils to equilibrate in models of this nature.

**Reply:**

The development of the ECE3 model was a continuous process that spanned over several years. We started a long run as soon as we had a promising candidate and evaluated the climate. Then we applied corrections (e.g. updated tuning parameters) and started a new run (or often a set of candidates), but used the restart states from the end of the old run assuming the changes to the model have only an incremental effect. This restart-stop-evaluation cycle was repeated several times before we had a final spun-up version that allowed us to start the piControl experiment. The entire length of all runs that eventually have led to the initial state for the piControl experiment is 1100 years, with the last chunk - done with the same model configuration as the CMIP6 experiments - stretching over 250 years.

We updated the text in the manuscript:

"The spinup of the coupled model prior to the final tuning was a continuous process during the years of development. Long runs were started as soon as a promising candidate version was available. After updating tuning parameters new runs were continuing from the end of the previous run, assuming the changes to the model have only an incremental effect. This restart-stop-evaluation cycle was repeated before a final spun-up version was available that allowed to start the piControl experiment. The entire length of all simulations is 1100 years, with the last chunk - done with the same model configuration as the CMIP6 experiments - stretching over 250 years."

**311 : redundant title repeated twice**

Reply: This is now corrected

320 "has been" -> "had been"

Reply: This is now corrected

399: "In this case, there is no coupling to the ice sheet model" - this is confusing, as this whole section is about coupling to the ice sheet model for PMIP. Perhaps this should say "For other resolutions"?

Reply: This is corrected now.

The new text is: "For other resolutions, there is no coupling to the ice sheet model."

455: incomplete sentence.

**Reply:**

The paragraph has been reformulated:

"It is important to note that the workflow of these experiments comprises different steps, with dependencies between them. This is especially true when the storage is a constraint and simulation steps need data from prior steps before post processing. In such cases, the way these dependencies are handled may have an impact on the overall throughput."

483: Some of these version details have been mentioned above. Perhaps only specify them here to help with length.

**Reply:**

We have considered the question and conclude that version information on the component models should be available both in the description of configurations and here in the description of component models. If we would need to cut, that we would rather shorten in the configuration section; however, another reviewer requests reference information already in the configuration section. The details only cover very few lines, and thus appear to be acceptable at both places in the manuscript to us.

518-523: This implies that snow albedo over land and sea-ice is computed differently. One might imagine a kind of non-physical line, where albedo changes as you move from "land snow" to "sea-ice snow" due to different parameterizations used in the sub-component models. A comment on the consistency of snow albedo and other proporties across the land-sea-ice boundary is perhaps warranted.

**Reply:**

We agree with the reviewer that snow albedo over land and sea ice is computed differently, which is a long-standing and difficult problem to solve for practical reasons, and probably shared by many modelling groups. However, the present paragraph is not about this issue but rather explains that the specification of sea-ice albedo is different in atmosphere-only and fully coupled models, so we suggest not to mention the snow continuity problem there.

595-600: a bit of repetition on the orbital parameters. I do not see any discussion here about how the land mask for the atmosphere is derived. Is a fractional or binary land mask being used? How does this related to the land mask in the ocean model? Is there tiling, and how are fluxes from ocean/land/ice combined?

**Reply:**

The orbital parameters are briefly mentioned in the configuration section 2.2, and described in more detail here in the model component section 3.1.

Concerning the masks, the following text has been added to section 2.2:

The land mask for the atmosphere is binary and is derived from GTOPO30 (see sections 11.2 and 11.4 in the IFS documentation

(https://www.ecmwf.int/en/publications/ifs-documentation). The ocean mask is binary as well and a remapping of coupling fields between atmosphere and ocean grid is carried out by the coupler OASIS\_MCT.

500-700: I do not see any explicit mention of how lakes and other inland water bodies are handled. This is important for their regional impact, and also how they are treated with respect to conservation of the global water cycle.

Reply:

A new section 3.9 has been introduced, about Lake treatment.

"EC-Earth3 has no explicit lake model. All gridpoints in the atmosphere model that are covered with less than 50% water in the land-sea mask are assumed to be land and use the HTESSEL scheme as described above. Gridpoints with more than 50% water are considered water, and the water temperature, sea ice cover and sea ice temperature in these points are updated by the coupler. This process is straightforward for gridpoints over the ocean, but for inland water bodies such as large lakes this implies an extrapolation of temperature etc. from the nearest unmasked gridpoint in the ocean model. This method potentially leads to problems in lakes where the closest ocean point is much further north such as the Great Lakes for which the closest ocean point is the Hudson Bay. This constellation implies colder lakes and longer sea ice cover which has an impact on the local climate (e.g. lake breeze) around these lakes. "

**773: Is this regular ORCA1 or the eORCA1 configuration?**

Reply:

It is the regular ORCA1 configuration

The last sentence in that paragraph has been extended: "The CMIP6 version of the EC-Earth model uses NEMO3.6 (revision r9466) in combination with the ORCA1 shared configuration (regular ORCA1, not eORCA1)."

Box: The protocol for testing replicability: Why use a single ocean restart with a small perturbation? Basically, larger state variations in the ocean are being excluded here, but would contribute to internal variability. 20 year simulations would not be long enough for the oceans to diverge significantly within the ensemble simulations themselves.

Reply:

The reviewer is right that this minimal method for perturbation does not allow sampling the entire distribution of climatic states, especially for the slower components like the ocean. However, we have noticed that the 5-member ensemble has diverged enough after a couple of years so that oceanic surface properties (SST, SSS) have a spread (standard deviation) that is commensurate with that of interannual variability.

We take the reviewers point that there is a risk that our method "misses" cases of replicability, i.e., when the large differences in oceanic states are accentuated by difference in computing environments, and have added the following sentence:

"The method of perturbation does not allow sampling the entire distribution of climatic states, as simulations are only 20 years long. If differences in computing environments cause differences in model output that are further enhanced by differences in climatic states, then there is a chance that our approach misses to detect cases of replicability. However, differences in surface properties like SSS and SST develop rather rapidly in the ensemble, with inter-model spread comparable to interannual variability well before 20 years."

Box: It's not clear whether the statistical testing (6) is on the standard metrics (5) only or also on the raw fields? The raw fields are shown in figure 2. If the testing were only on the standard metrics, it could lead to false negatives, because simulations on two platforms could have similar global-level biases with completely different underlying structure.

**Reply:**

The statistical testing is done primarily on the standard metrics. When an inconsistency is detected, the same statistical testing can be applied to output fields (spatial SST, for example) for further investigation in order to identify where the differences come from. The reviewer is right that the method, because it works by analyzing global numbers, can hide different realities and that there is always a risk that differences go undetected. We are developing multiple tests on multiple variables, and we are looking at maps, to minimize this possibility as much as possible.

We have now acknowledged in the text box:

"Because the tests are done on global metrics, there is a possibility of false negatives when the same performance is obtained from spatially varying biases. However, the use of multiple variables for the assessment of maps aims at minimizing this issue"

980-981: That a difference was detected and corrected proves the test can be useful, but it does not indicate instances when it might have failed to detect a difference. The test, especially if it is as above, definitely has a significant chance of failing to detect real differences. e.g. also see Baker, A. H., Hammerling, D. M., Levy, M. N., Xu, H., Dennis, J. M., Eaton, B. E., Edwards, J., Hannay, C., Mickelson, S. A., Neale, R. B., Nychka, D., Shollenberger, J., Tribbia, J., Vertenstein, M., and Williamson, D.: A new ensemble-based consistency test for the Community Earth System Model (pyCECT v1.0), Geosci. Model Dev., 8, 2829–2840, https://doi.org/10.5194/gmd-8-2829-2015, 2015.

**Reply:**

Yes, there is a non negligible probability (>20%) that small differences are not detected by the test. This is inherent to the limited statistical power due to the small ensemble size (itself due to limited computational resources). We believe that this is the case of any statistical test, where a tradeoff has to be found between minimizing errors of type-I (false positives), errors of type-II (false negatives) and available CPU resources. This is why we treat the test, at best, as a tool that can alert us on the existence of non-replicability. We never interpret a negative outcome of the test as conclusive evidence that the model is fully replicable.

Section 5: In terms of the CMIP6 protocol, the DECK simulations should be submitted for each model configuration. Please discuss the status of this, as it seems piControl runs have only been done for a limited number of configurations.

Reply:

Here we focus on the base configurations EC-Earth3 and EC-Earth3-veg only. We expect additional papers on the remaining configurations, that also discuss corresponding piControl runs. In the meantime most configurations have participated in CMIP6-MIPS and went through required DECK simulations. Resulting data will be published on the ESGF.

We note now in the text:

"PiControl simulations for other model configurations have been carried out, but will not be used in this paper."

**1017-1018: Can you say the response is too high? What about internal variability?**

Reply:

The reviewer is right. We omit the phrase ", indicating a too strong sensitivity of the model to the observed forcing"

1024-1030: The warm bias in the south is significant, and represents a large deterioration relative to the previous model version. This seems a bit inconsistent with the statements made in the abstract. Some discussion of the source of the new large bias would be appropriate.

Reply:

Yes, the warm bias in the southern ocean is significant and it is not unusual for ESMs.

We have added text that discusses possible reasons, based on existing literature:

"Most coupled climate models suffer from a warm southern ocean (SO) bias (Hyder et al 2018). In EC-Earth3 configurations, the warm bias is found in all seasons. Large parts of the bias have been attributed to biases in short wave cloud radiative effects. Modifications in the cloud scheme and the representation of supercooled liquid water made in more recent versions of IFS, including cycle 45r1 (Forbes and Ahlgrimm, 2014; Forbes et al., 2016), together with the introduction of the new ecRad radiation scheme in cycle 43r3 (Hogan et al., 2017) have been shown to substantially reduce these biases."

Conclusions: I don't think it is necessary to summarize the result over every bias again here. Perhaps replace this with a shorter, more general overview of conclusions of the validation.

Reply:

For a quick reader it could be helpful to summarize even the biases in this section. Therefore we prefer to keep it.

**Table 1: What about ocean BGC?**

Reply:

A column for ocean biogeochemistry has been added.

Table 2+: timestep in the atmosphere or ocean or both? Presumably not the same. What about the coupling interval? That is noted in some of the subsequent tables (7,8) but not all of them (3-5, 10).

Reply:

Table 2 has been updated with timesteps for atmosphere, ocean and for the coupling.

The caption of tables 5 has been updated with coupling frequencies: "...Variables exchanged between the atmosphere and the vegetation model, with a coupling frequency of 1 day (in standard and low resolution)."

Table 10 captions have been updated with coupling frequency information: "...Information is exchanged once a year with monthly variations"

**Figure 10: Showing both hemispheres on the same seasonal panel makes the changes very hard to see.**

Figure 10 has been replaced with a 2 panel figure. The caption has been adjusted: "Figure 10: Time series of Arctic (left column) and Antarctic (right column) sea ice area for both EC-Earth3 (ensemble mean as thick solid lines) and satellite observations (OSI SAF as blue dash dotted lines and NSIDC as dashed lines). The EC-Earth3 ensemble minimum up to maximum value is represented by the shading around the ensemble mean."

**Figure 12: Would be clearer in two panels, one for Sep and one for Mar.**

Figure 12 has been replaced by a 2-panel version. The caption has been adjusted: "Time series of September (orange, upper panel) and March (blue, lower panel) Arctic sea ice volume for EC-Earth3 (thin solid lines representing the ensemble mean), EC-Earth3-Veg (dashed lines representing the ensemble mean), the CMIP5 version of EC-Earth (dotted lines) and PIOMAS reanalysis (thick solid lines). The EC-Earth3 and EC-Earth3-Veg ensemble minimum and maximum are represented by the same linestyle as their means, but with transparent shading added around the ensemble means."

---

## Author Response (AR2)

Dear Editor,

Compared to the previous version, I have edited purely formal aspects as described below.

- I changed a word in the titel, "Coupled" replaced "Climate", because it the the actual meaning in "CMIP". If that should be not possible anymore, keep the original title.

- I changes few letters in names that were wrongly spelled.

- I added two lines at the end of the acknowledgements

A co-author, Benjamin Smith cannot link his co-authorship of this paper to his Copernicus author profile with user-id #194784. How can he solve that?

Is it possible to add a second affiliation to the co-auther Matthias Gröger?:
Leibniz Institute for Baltic Sea Research Warnemünde, Rostock, Germany

Best Regards
Ralf